# How Classifier Features Transfer to Downstream: An Asymptotic Analysis in a Two-Layer Model

**HEE BIN YOO**
Interdisciplinary Program in
Artificial Intelligence
Seoul National University
yooheebin@snu.ac.kr

**Sungyoon Lee**
Department of Computer Science
Hanyang University
sungyoonlee@hanyang.ac.kr

**Cheongjae Jang**
Department of Computer Science
Hanyang University
jchastro@gmail.com

**Dong-Sig Han**
Department of Computing
Imperial College London
d.han@imperial.ac.uk

**Jaein Kim**
Interdisciplinary Program in
Neuroscience
Seoul National University
qpwodlsqp@snu.ac.kr

**Seunghyeon Lim**
Artificial Intelligence Institute
Seoul National University
shlim@bi.snu.ac.kr

**Byoung-Tak Zhang**[*]
Artificial Intelligence Institute
Seoul National University
btzhang@snu.ac.kr

## Abstract

Neural networks learn effective feature representations, which can be transferred to new tasks without additional training. While larger datasets are known to improve feature transfer, the theoretical conditions for the success of such transfer remain unclear. This work investigates feature transfer in networks trained for classification to identify the conditions that enable effective clustering in unseen classes. We first reveal that higher similarity between training and unseen distributions leads to improved Cohesion and Separability. We then show that feature expressiveness is enhanced when inputs are similar to the training classes, while the features of irrelevant inputs remain indistinguishable. We validate our analysis on synthetic and benchmark datasets, including CAR, CUB, SOP, ISC, and ImageNet. Our analysis highlights the importance of the similarity between training classes and the input distribution for successful feature transfer.

## 1  Introduction

Neural networks have demonstrated remarkable success through their ability to learn hidden representations, which are crucial for generalization [Damian et al., 2022]. These learned features, especially those extracted from the penultimate layers, are semantically meaningful and transferable across tasks [Yosinski et al., 2014, Kornblith et al., 2019]. A wide range of techniques, from open-set clustering [Roth et al., 2020] to vision-language models [Li et al., 2023] and language models [Kojima et al., 2023], leverage feature transfer for downstream tasks. Recent successes in feature transfer have been largely attributed to the increasing volume of available data [Brown et al., 2020, Radford et al., 2021, An et al., 2023]. However, the underlying conditions under which features can be effectively transferred remain underexplored.

---

[*]Corresponding author

39th Conference on Neural Information Processing Systems (NeurIPS 2025).

Among various applications, classification-based visual open-set clustering [Musgrave et al., 2020] is a fundamental benchmark for evaluating feature transferability on unseen data. The task typically involves training a classifier on one set of classes and then testing it on a disjoint set of unseen classes to assess whether the extracted features form cohesive and separable class-wise clusters in unseen data [Wang et al., 2018, Seidenschwarz et al., 2021, Deng et al., 2022]. Given this context, we aim to investigate feature transfer through open-set clustering with the following research questions:

> Can we **capture the presence of feature learning in classification** and **identify the conditions under which features cluster effectively** in new distributions?

To address this, we consider a two-layer nonlinear network trained with a single gradient descent step on a mean-squared classification loss in the *proportional regime* (in Section 2). Then, we show that the dominant part of the trained feature function consists of random initialization and a *spike component* (Def. 3.4) associated with the training data (in Section 3). Finally, leveraging the dominant features, we identify conditions for effective clustering in unseen distributions (in Section 4).

First, we assess the intra-class *Cohesion* and inter-class *Separability* of trained features through numerical and analytical analysis, as standard measures of clustering performance [Clémençon, 2011, Papa et al., 2015, Liu et al., 2017, Li and Liu, 2021]. As a result, we find that *Cohesion* increases as the *train-unseen similarity* grows. Meanwhile, for *Separability*, if classes are *assigned* to different training classes, it also increases with the *train-unseen similarity*; otherwise, it decreases, as shown in Figure 1.

Figure 1: Mapping input data (left) to learned feature space (right); Training classes are solid balls, unseen classes $a,b,p,n$ are dashed lines. *Cohesion*: Strong *Cohesion* occurs for $a,p,n$, which have high similarity to the training data compared to $b$. *Separability* of $a,n$: *assigned* to different training class, they show high *Separability*. *Separability* of $a,p$: *assigned* to the same training class, they exhibit low *Separability*.

Second, we analyze the "spike component," the predominant element of the feature function in multi-class settings. We show that the inner product between the *spike component* and a new input governs the feature's expressiveness. Specifically, the *spike component* maps new inputs using a linear combination of randomly initialized classifier-head weights. Therefore, we find that the *spike direction* contributes to feature extraction only when it aligns with the input data, as illustrated in Figure 3.

Empirically, we evaluate *train-unseen similarity*, *Cohesion*, *Separability* under our theoretical assumptions using synthetic datasets. As predicted by our analysis, *Cohesion* and *Separability* follow the expected trends with respect to *train-unseen similarity*. We further study open-set clustering in realistic settings and demonstrate the practical implications of our theoretical results. In most cases, clustering performance is higher when the unseen classes belong to the same semantic domain as the training classes (Expr. IV). Moreover, adding dissimilar classes does not enhance performance (Expr. V), whereas incorporating semantically related training classes does (Expr. VI)—consistent with our theoretical claims.

This work provides new insights showing that effective feature transfer depends on the *train-unseen similarity*, further implying that features can be learned in a data-efficient manner without relying on large-scale data. Our contributions are summarized as follows:

- We analyze the classifier's feature representation, providing insights into how feature extractors operate:
    - Higher *train-unseen similarity* increases *Cohesion*.
    - Higher *train-unseen similarity* increases *Separability* between data *assigned* to different classes but reduces it otherwise.
    - Feature expressiveness improves as more *spike directions* become non-orthogonal to the input.
- We validate the theoretical results through experiments and show that they hold in practical settings.
- We generalize the distribution assumption of prior works to non-centered sub-Gaussian distributions and present novel proof techniques for classifier analysis (refer to Assumption 2.2).

## 1.1 Related Works

**Deep Metric Learning and Open-Set Clustering**   Metric learning has been proposed to cluster visually similar unseen classes using classification or triplet losses [Movshovitz-Attias et al., 2017, Zhai and Wu, 2019, Boudiaf et al., 2021]. Several recent approaches have focused on increasing the number of training classes to improve clustering performance. One line of work introduces virtual classes [Chen et al., 2018, Qian et al., 2020, Gu et al., 2021]. Another approach leverages a large number of classes induced from the dataset of Schuhmann et al. [2021] to achieve state-of-the-art performance [An et al., 2023]. This empirical trend aligns with our analysis, which suggests that performance improves as the number of relevant classes for clustering increases. However, to the best of our knowledge, theoretical analyses of feature transfer—particularly within metric-learning frameworks without additional training—remain unexplored.

**Neural Collapse (NC) and the Unconstrained Layer-Peeled Model (ULPM)**   Recent studies have introduced the concept of Neural Collapse [Papyan et al., 2020] to explain the emergence of intra-class feature compactness and feature-weight alignment in trained neural networks. To analyze these phenomena, several studies propose the ULPM which treats understand the training dynamics of NC treating features and classifier weights as unconstrained free variables [Fang et al., 2021, Zhu et al., 2021, Ji et al., 2022, Tirer and Bruna, 2022]. Unlike the two-layer network model adopted in our study, ULPM assumes free variable features, which limits the analyzability of input distribution and, consequently, prevents the study of feature transferability.

**Feature Learning in Two-Layer Networks**   The Conjugate Kernel (CK) has been widely used to analyze feature learning in two-layer networks [Louart et al., 2017, Goldt et al., 2020, Hu and Lu, 2022]. More recently, Ba et al. [2022], Moniri et al. [2024] have shown that feature learning can reduce population risk in teacher-student regression settings. In contrast, our work investigates feature transfer in classifier-trained networks, particularly in scenarios resembling metric learning, which remain theoretically underexplored. To this end, we extend CK-based regression analyses to accommodate non-centered sub-Gaussian inputs with Hermite-expandable activations (Sections I, H), enabling classifier-based training with arbitrary labels and distributions. Moreover, Table 1 in Appendix C compares our framework with contemporary studies on feature learning in shallow networks.

Additional related work on shallow-network feature learning, feature transferability, and classifier-based feature analysis is provided in Appendix C.

## 2   Problem Statement

**Notations**   We denote by $\|\cdot\|$ either the $L^2$ or operator norm. Let $\odot$ be the Hadamard product and $A^{\circ k}$ be the Hadamard power. Constants $C, c > 0$ and $\kappa \in \mathbb{R}$ may vary from line to line. Define $[\![d]\!] \triangleq \{1, 2, \cdots, d\}$. For $o, O, \Theta$ notations regarding complexity, we follow Moniri et al. [2024].

**Training Data**   We consider a one-vs-one classification problem with K classes that constructs $\mathrm{P} \triangleq \binom{\mathrm{K}}{2}$ problems. Let $e_1, \ldots, e_\mathrm{K}$ denote the class-conditional distributions of the training data, and the training dataset be $\mathcal{D} = (X, Y)$, where $X \in \mathbb{R}^{\mathbf{n} \times \mathbf{d}}$ and $Y \in [\![\mathrm{K}]\!]^{\mathbf{n}}$. Here, $X = \bigcup_{k=1}^{\mathrm{K}} \left(\{x \sim e_k\} \times m\right)$, where $m$ is the number of instances per class and $\mathrm{K}m = \mathbf{n}$ [^2].

**Network Structure in the Proportional Regime**   We consider two-layer networks in the proportional regime [Ba et al., 2022], which represents a scenario where the network width and the size of the dataset are of similar scales. Let $\mathbf{n}, \mathbf{d}$, and $\mathbf{N}$ be sample size, data, and feature dimension, respectively. We perform our analysis under $\mathbf{d}/\mathbf{n}, \mathbf{N}/\mathbf{n} \to c$ as $\mathbf{n}, \mathbf{d}, \mathbf{N} \to \infty$. Concretely, the initial weights of the first layer and the second layer are $W_0 \in \mathbb{R}^{\mathbf{d} \times \mathbf{N}}$ and $a_{ij} \in \mathbb{R}^{\mathbf{N}}$ for $i, j \in [\![\mathrm{K}]\!]$ s.t. $i < j$, respectively. These are initialized as $W_0[i] \sim Unif(\mathbb{S}^{\mathbf{d}-1})$ for $i \in [\![\mathbf{N}]\!]$ and $a_{ij}$ follows a Gaussian distribution $\mathcal{N}(0, \frac{1}{\mathbf{N}}I)$. Denote $W$ as the first-layer weights after a single gradient step. Accordingly, we define $F_0(x) \triangleq \sigma(W_0^\top x)$ and $F(x) \triangleq \sigma(W^\top x)$ as the initialized and trained features, respectively.

---

[^2]: Our theorem also holds in imbalanced settings, but we use balanced settings here for simplicity.

**Optimization Problem** Denote the set of all network parameters as $\theta = \{W, a_{12}, \cdots, a_{K-1,K}\}$. Let $X_{ij}$ be a matrix in $\mathbb{R}^{2m \times \mathbf{d}}$, where the first $m$ rows contain samples $x \sim c_i$ and the last $m$ rows contain samples $x \sim c_j$. Let $y \triangleq [1, 1, \ldots, 1, -1, \ldots, -1]^\top \in \mathbb{R}^{2m}$ be a vector consisting of $m$ ones followed by $m$ negative ones. To classify the given data, we use the Mean Squared Error,

$$L(x, y; \theta) = \frac{1}{2n} \sum_{i<j} \|y - \sigma(X_{ij}W)a_{ij}\|^2. \tag{1}$$

The weight update for the first layer is given by $W = W_0 + G$, where $G \triangleq -\frac{\partial L}{\partial w} = \sum_{i<j} G_{ij}$, s.t.

$$G_{ij} = -\frac{1}{n}\Big[X_{ij}^\top[(\sigma(X_{ij}W)a_{ij} - y)a_{ij}^\top \odot \sigma'(X_{ij}W)]\Big]. \tag{2}$$

We then introduce the assumptions for the theoretical analysis. We omit the learning rate for simplicity, but it may be explicitly included as $\eta = \Theta(1)$ without affecting the proof.

**Assumption 2.1** (Activation Function). Let $\sigma(x)$ be an element-wise activation s.t. $\sigma', \sigma'', \sigma'''$ is bounded by $\lambda_\sigma$ almost surely. It admits a Hermite decomposition i.e. $\sigma(z) = \sum_{k=0}^\infty c_k H_k(z)$, where $c_k = \frac{1}{k!}\mathbb{E}[\sigma(z)H_k(z)]$ for a standard Gaussian $z$. We assume $c_0 = 0, c_1 > 0$ and $c_k^2 k! \leq Ck^{-3/2-w}$, for constants $C, w > 0$ e.g., the Shifted ReLU, defined as $\max(x, 0) - \frac{1}{\sqrt{2\pi}}$.

**Assumption 2.2** (Training Data). We assume the class-conditional distributions $c_i$ are non-centered Sub-Gaussians [Vershynin, 2018, Cao et al., 2021, Cole and Lu, 2024], a necessary generalization for classification tasks involving non-identical distributions with bounded support. This extends common assumptions such as Gaussians and Gaussian mixtures (see Appendix Table 1 for comparison).

To accommodate non-centered Sub-Gaussian distributions, we generalize Hermite expectation techniques from centered Gaussians to non-centered Sub-Gaussians (Lemma F.5, Appendix H), extending prior work on centered Gaussian cases [O'Donnell, 2021, Moniri et al., 2024]. We also adapt key results for centered Sub-Gaussians [Vershynin, 2010, 2018] to the non-centered setting (Appendix I). Please refer to Appendix D for preliminary information on Hermite polynomials and for additional notations used in the proofs.

## 3 Feature Decomposition

This section analyzes the one-step trained feature extractor in the proportional regime. As a first step, we linearize the gradient $G$ in the proportional regime to approximate the trained feature.

**Theorem 3.1** (Gradient Approximation). *Let $\mathbb{A}_{ij} \triangleq \frac{c_1}{n}X_{ij}^\top ya_{ij}^\top$, a rank-one structure for each $ij$-th classification problem. Under Assumptions 2.1 and 2.2, and for sufficiently large $\mathbf{n}$,*

$$\Big\|G - \sum_{i<j} \mathbb{A}_{ij}\Big\| \leq \kappa\frac{\log^2 \mathbf{n}}{\mathbf{n}}, \quad w.p. \ 1 - o(1). \tag{3}$$

For the proof, please refer to Appendix E. Theorem 3.1 shows that the gradient $G$ with respect to $W_0$ exhibits an almost Rank-P structure and can be expressed as $\sum_{i<j} \mathbb{A}_{ij}$. Next, leveraging $\sum_{i<j} \mathbb{A}_{ij}$ and the Hermite decomposition of the activation $\sigma$, we deterministically decompose the one-step trained feature $F(x) = \sigma((W_0 + G)^\top x)$ into its dominant and residual components, in order to analyze the feature structure for new inputs based on its deterministic and dominant terms.

**Definition 3.2** (Spike Direction). The *spike direction* is defined as $\beta_{ij} = \frac{1}{n}X_{ij}^\top y \in \mathbb{R}^{\mathbf{d}}$. If the distinction between $i$ and $j$ is not essential, we omit the subscripts.

**Theorem 3.3** (Feature Decomposition). *Under Assumptions 2.1 and 2.2, let $\tilde{X} \in \mathbb{R}^{\mathbf{n} \times \mathbf{d}}$ follow Sub-Gaussian, $L \triangleq \log n$, $F_{0,L} \triangleq \sum_{k=1}^L c_k H_k(\tilde{X}W_0)$, and, $s_L \triangleq \sum_{k=1}^L c_1^k c_k(\tilde{X}\sum_{i<j}\beta_{ij}a_{ij}^\top)^{ok}$.*

$$F = F_{0,L} + s_L + \Delta, \quad w.p. \ 1 - o(1), \tag{4}$$

*where $\|s_L\| = \Omega(\sqrt{\mathbf{n}})$, $\|F_{0,L}\| = \Theta(\sqrt{\mathbf{n}})$, and $\|\Delta\| = o(\sqrt{\mathbf{n}})$.*

Please refer to Appendix F for the proof. Theorem 3.3 represents that the learned features are mainly composed of the *spike component*—which maps new inputs via inner products with the data-label covariance $\beta_{ij}$—and $F_{0,L}$, inherited from the random initialization. Under Theorems 3.1 and 3.3, it follows that for sufficiently large $\mathbf{n}$, the one-step trained feature is primarily described by the term $F_{0,L} + s_L$. Therefore, we introduce the deterministic Dominant Feature $F_L(x)$ below, which serves as the foundation for our subsequent analysis.

**Definition 3.4** (Dominant Feature $F_L(x)$)**.** For sufficiently large $\mathbf{n}$, w.h.p and retaining only components of $F$ larger than $o(\sqrt{\mathbf{n}})$, we define the dominant feature $F_L$ as follows.

$$F_L(x) \triangleq \underbrace{\sum_{k=1}^{L} c_k H_k(W_0^\top x)}_{F_{0,L}(x)} + \underbrace{\sum_{k=1}^{L} c_1^k c_k (\sum_{i<j} (\beta_{ij}^\top x) a_{ij}^\top)^{\circ k}}_{s_L(x)}, \tag{5}$$

where $F_{0,L}$ is the approximated randomly initialized feature, and $s_L$ is the trained *Spike Component*.

$F_L(x)$ defined in Definition 3.4 represents the dominant part of the feature decomposition using the Hermite expansion for sufficiently large $\mathbf{n}$. This approximation is expressed through linear operations and polynomials, facilitating an explicit closed-form representation of the clustering criteria.

## 4 Feature Analysis

### 4.1 Clustering Criteria Analysis

We analyze two key criteria—*Cohesion* and *Separability*—which capture intra-class compactness and inter-class distinction of features to identify the conditions under which features from unseen classes form coherent and separable clusters. We suggest that these quantities are governed by the *train ($\beta$)-unseen ($\mu$) similarity* under Condition 4.2 in binary classification. Specifically, we express both *Cohesion* and *Separability* (Definition 4.3) as closed-form polynomials of $\beta^\top \mu$.

**Definition 4.1** (*Train-Unseen Similarity*)**.** Given the *spike direction* $\beta$ in Definition 3.2 and the mean of the unseen distribution $\mu$, the *Train-Unseen Similarity* is defined as $\beta^\top \mu$.

*Condition* 4.2. Suppose $\mathbf{n}, \mathbf{d}, \mathbf{N}$ are fixed and sufficiently large. Under Assumptions 2.1, 2.2, let $c_i = \mathcal{N}(\mu_i, I_\mathbf{d})$ for $i \in [\![2]\!]$ be the class conditional distributions. Define $\rho_{k,k',r,r'} > 0$ to depend only on $\mathbf{N}$ and $\mathbf{d}$, and to be independent of $\mu$ and $\beta$, as in Definition G.1.

**Definition 4.3** (*Cohesion* and *Separability*)**.** We define two clustering performance criteria as functionals based on the inner product similarity between features, for given any feature $G$:

$$\mathcal{C}(G) \triangleq \mathbb{E}_\theta[\mathbb{E}_{x \sim c} G(x)^\top \mathbb{E}_{x' \sim c} G(x')], \qquad \mathcal{S}(G) \triangleq -\mathbb{E}_\theta[\mathbb{E}_{x \sim c_1} G(x)^\top \mathbb{E}_{x' \sim c_2} G(x')].$$

*Cohesion* $\mathcal{C}(G)$ quantifies the expected inner product similarity between i.i.d. features belonging to the same class $c$, with the expectation taken over network parameters $\theta$. *Separability* $\mathcal{S}(G)$ quantifies the expected dissimilarity between independent features from different classes $c_1$ and $c_2$, again with the expectation taken over $\theta$.

Since $\|F_{0,L}\|/\|s_L\| \to 0$ as $\mathbf{n} \to \infty$ by Theorem 3.3, we focus on presenting $\mathcal{C}(s_L)$ and $\mathcal{S}(s_L)$[3].

**Proposition 4.4** (Cohesion of $s_L$)**.** *Following Condition 4.2, the Cohesion $\mathcal{C}(s_L)$ for $c$, is given by:*

$$\mathcal{C}(s_L) = \sum_{\substack{k=1 \\ k'=1}}^{L} c_k c_{k'} \left[ \sum_{r=0}^{k} \sum_{r'=0}^{k'} \rho_{k,k',r,r'} |\beta^\top \mu|^{k+k'-r-r'} \|\beta\|^{r+r'} \right]. \tag{6}$$

**Proposition 4.5** (Separability of $s_L$)**.** *Under Condition 4.2, the Separability $\mathcal{S}(s_L)$ for $c_1, c_2$ yields:*

$$\mathcal{S}(s_L) = -\sum_{\substack{k=1 \\ k'=1}}^{L} c_k c_{k'} \left[ \sum_{r=0}^{k} \sum_{r'=0}^{k'} \rho_{k,k',r,r'} (\beta^\top \mu_1)^{k-r} (\beta^\top \mu_2)^{k'-r'} \|\beta\|^{r+r'} \right]. \tag{7}$$

The proofs of Propositions 4.4 and 4.5 are in Appendix G. In the above Propositions, $\mathcal{C}(s_L)$ and $\mathcal{S}(s_L)$ are computed in closed-form expressions and expressed as polynomials of *Train-Unseen Similarity*. Also, $\mathcal{C}(F_L)$ and $\mathcal{S}(F_L)$, which include the $F_{0,L}$ term, are polynomials of the *train-unseen similarity*, as shown in Propositions G.2 and G.3. With the polynomial from, to identify conditions that improve the clustering criteria, we numerically evaluate $\mathcal{C}(F_L)$ and $\mathcal{S}(F_L)$ by varying $\mu_1^\top \beta$ and $\mu_2^\top \beta$.

---

[3]Empirical support for the dominance of $s_L$ in $\mathcal{C}(F_L)$ and $\mathcal{S}(F_L)$ is presented in Appendix K.

As illustrated in Figure 2, we find that *Cohesion* increases with $|\mu^\top\beta|$, i.e., as the "unseen" data becomes more aligned with the *Spike Direction* $\beta$. Also, *Separability* rises when the magnitudes of $\mu_1^\top\beta$ and $\mu_2^\top\beta$ grow with opposite signs and decreases when they grow with the same sign i.e., it increases when the two "unseen" classes are *assigned* to different classes as explained below:

*Note* 4.6 (Explanation of *assignment* and $\beta^\top\mu$). $\beta$ represents the normal vector of the linear decision boundary in binary classification within the data space before training, i.e. the direction that determines class assignment based on the sign of its inner product with the data. Therefore, the sign of $\beta^\top\mu$ indicates the class *assignment* of unseen data with mean $\mu$.

These two phenomena can be interpreted based on the property of the leading term $s_L$ in $F_L$. The phenomena follow the expressions inside the brackets of Eqs. (6) and (7). Furthermore, if the Shifted ReLU is used as the activation function, the analysis in Appendix D.1 shows that the first Hermite coefficient $c_1$ takes a significantly positive value, while the subsequent coefficients oscillate and gradually diminish in magnitude. This suggests that the positive contribution may dominate the summation $\sum c_k c_{k'}$, which is consistent with our empirical observations. However, further investigation is required to validate this hypothesis rigorously. Additionally, in Appendix K, we show that our theory tends to hold over a wider range as $\mathbf{n}$ increases, and in Note G.6, we discuss the $\|\beta\|$ term.

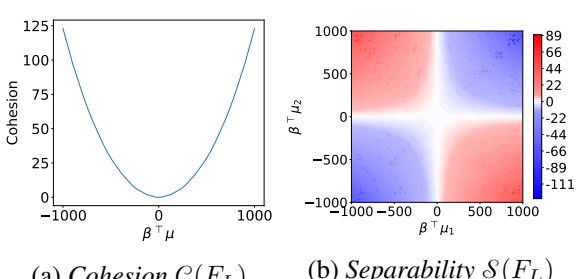

(a) *Cohesion* $\mathcal{C}(F_L)$    (b) *Separability* $\mathcal{S}(F_L)$

Figure 2: F by adjusting $\beta^\top\mu_1$ and $\beta^\top\mu_2$. We set $\mathbf{n} = 320000$, $\mu_1 = -\mu_2 \in \mathbb{R}^{\mathbf{n}}$ and $\|\mu_1\|, \|\mu_2\|, \|\beta\| = 1$.

## 4.2 Spike Component Analysis

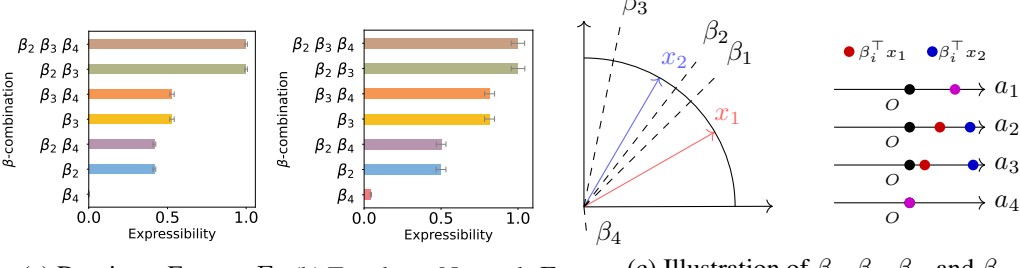

(a) Dominant Feature $F_L$ (b) Two-layer Network $F$    (c) Illustration of $\beta_1$, $\beta_2$, $\beta_3$, and $\beta_4$

Figure 3: *Expressibility* of $F_L$ and $F$ trained with combinations of $\beta_2$, $\beta_3$, and $\beta_4$ (a, b): While including $\beta_2$ and $\beta_3$ allow $F_L$ and $F$ to extract distinct features as $x_1$ and $x_2$ vary, $\beta_4$ yields less informative representations. (c): Including $\beta_1$ and $\beta_4$ does not contribute to distinguish $x_1$ and $x_2$.

This section examines the impact of the *spike component* $s_L$ on $F_L = F_{0,L} + s_L$, specifically how the learned component $s_L$ contributes to feature extraction for unseen inputs.

**Corollary 4.7** (Corollary of Theorem 3.3). *Let the inputs $x_1, x_2 \in \mathbb{R}^{\mathbf{d}}$ and remark Dominant Feature $F_L(x) = F_{0,L}(x) + s_L(x)$ in Definition 3.4. If $|\beta^\top x_1 - \beta^\top x_2| \to 0$ (e.g., when $x_1, x_2 \perp \beta$). Then,*

$$\|s_L(x_1) - s_L(x_2)\| \to 0 \quad and \quad \|F_L(x_1) - F_L(x_2)\| \to \|F_{0,L}(x_1) - F_{0,L}(x_2)\|.$$

Corollary 4.7 shows how the dominant feature $F_L$ behaves under different conditions. Specifically, if the direction of $x$ is not orthogonal to $\beta_{ij}$, then spike of $\beta_{ij}$ involve feature extraction. Conversely, when $x$ is orthogonal to $\beta_{ij}$, the impact of spike $\beta_{ij}$ is eliminated, indicating that the training becomes irrelevant and the feature extractor relies solely on the random features.

To demonstrate this, we define four $\beta$ directions based on two test inputs $x_1, x_2 \in \mathbb{S}^{d-1}(\sqrt{\mathbf{d}})$ as Midpoint $\beta_1 = \frac{x_1+x_2}{2}$, Interpolation $\beta_2 = \frac{x_1+3x_2}{4}$, Extrapolation $\beta_3 = \frac{-x_1+5x_2}{4}$, and Orthogonal $\beta_4$, a random unit vector orthogonal to both $x_1$ and $x_2$ s.t. $\|\beta_i\| = \sqrt{\mathbf{d}}, \forall i \in [\![4]\!]$. By construction, $\beta_1$ and $\beta_4$ cannot contribute to distinguishing the two inputs via inner product, while $\beta_2$ and $\beta_3$ can distinguish the two inputs. See Figure 3 (c) for an illustration.

We evaluate both the approximated features $F_L$ and the trained two-layer neural network $F$ on four binary classification tasks, where each task involves Gaussian distributions with means $\beta_i$ and $-\beta_i$ to form given $\beta_i$, and covariance $0.1I$ with $\mathbf{n} = \mathbf{d} = \mathbf{N} = 2^{11}$. For all non-empty combinations of $\beta_i$, the $L_2$ distances between $F(x_1)$ and $F(x_2)$, and between $F_L(x_1)$ and $F_L(x_2)$, are measured by varying the angle between $x_1$ and $x_2$ from 0 to $\frac{\pi}{2}$ radians. The results are summarized using *Expressibility*, defined as the maximum $L_2$ distance between two feature vectors for a given class combination, normalized by the global maximum distance observed across all combinations. This metric quantifies how well a representation distinguishes structural variations in the input space.

As shown in Figure 3 (a,b), features derived from $\beta_4$ exhibit limited *Expressibility*. In contrast, features from $\beta_2$ and $\beta_3$ are more responsive to input variations, capturing meaningful structural differences. When multiple expressive $\beta$s (e.g., $\beta_2$ and $\beta_3$) are combined, the resulting features are even more sensitive to input changes, reflecting improved representational capacity. These results suggest that training with structurally relevant training classes enhances feature *Expressibility*, while incorporating unrelated classes as $\beta_4$ has minimal impact.

Additional experimental results are presented in Appendix L, including analyses of $\beta_1$ and experiments isolating the contributions of the $F_{0,L}$ and $s_L$ components. We also provide the original data used in the *Expressibility* computation of Figure 3. In summary, Consistent with $\beta_4$, $\beta_1$ also shows limited expressiveness. In addition, the $s_L$ component from $\beta_1$ and $\beta_4$ maps $x_1$ and $x_2$ to nearly identical feature vectors. In contrast, the $F_{0,L}$ component introduces some separability, suggesting that $F_L$ and $F$ calculated from less expressive $\beta$ can still extract weakly distinguishable features.

# 5 Experiments

We conduct seven experimental setups to validate our analysis and explore practical implications of the above analysis for open-set clustering, using feasible clustering criteria *Recall@1* (Remark 5.1).

*Remark* 5.1. *Recall@1* $\triangleq \mathbb{E}_{x_i, y_i} \mathbf{1}_{\mathrm{y_i = \hat{y}_{i,1\text{-NN}}}}$. $\hat{y}_{i,1\text{-NN}}$ denotes the class of the nearest feature to $x_i$. This serves as a practical metric for evaluating whether new unseen test classes form coherent clusters.

The choice of recall@1 is motivated by its ability to reflect clustering quality: when both Cohesion and Separability are high, the nearest neighbor is more likely to belong to the same class, leading to a higher recall@1. First, in Experiments I, II and III, we examine the relationship between *train-unseen similarity* (i.e. $\beta^\top \mu$) and *Cohesion*, *Separability*—as discussed in Section 4.1—and *Recall@1* within theoretical assumptions. Second, to demonstrate how our theoretical explanations can provide intuition in practical settings, we conduct the Experiments IV, V, VI, and VII. For this purpose, we employ the open-set clustering problem using fine-grained real image datasets. For all experiments, we repeat each experiment with three repetitions, except for VII, where we perform five runs.

## 5.1 Setup for Theory Vaildation: Expr. I, II, III

**Synthetic Data Design**  Based on Assumptions 2.1 and 2.2, and Condition 4.2, we construct three types of non-centered sub-Gaussian training datasets—uniform ball, truncated Gaussian, and uniform sphere (Data 1, 2, and 3). All distributions are origin-symmetric. For evaluation, we introduce two Gaussian test distributions—Eval 1 and Eval 2—parameterized by a translation vector $e$ and a rotation matrix $R \in \mathrm{SO}(n)$ which control the *train-unseen similarity*. Specifically, as $e$ increases from 0 to 1, $\beta^\top \mu$ increases due to mean shift; similarly, as $R$ approaches the identity matrix $I$, $\beta^\top \mu$ also increases due to rotational alignment. Refer to Figure 24 and Appendix M.1 for illustrations and details.

**Experiment Design**  For each experiment, we utilize Data 1, 2, and 3, with each dataset paired with a distinct evaluation dataset. Expr. I uses two Eval 1 distributions with translation parameter $e_1 \in [-0.9, 0.9]$ and $e_2 = -e_1$, so they are *assigned* to opposite training classes. Experiments II and III are based on two Eval 2 distributions, each parameterized by a small-angle random rotation matrix $R$ generated as described in Appendix P. In Expr. II, considering the case where the datasets are *assigned* to opposite classes, the first distribution uses $R$, and the second distribution uses $-R$. In Expr. III, considering the situation where the datasets are *assigned* to the same class, the first distribution uses $R$ and the second uses $R^\top$.

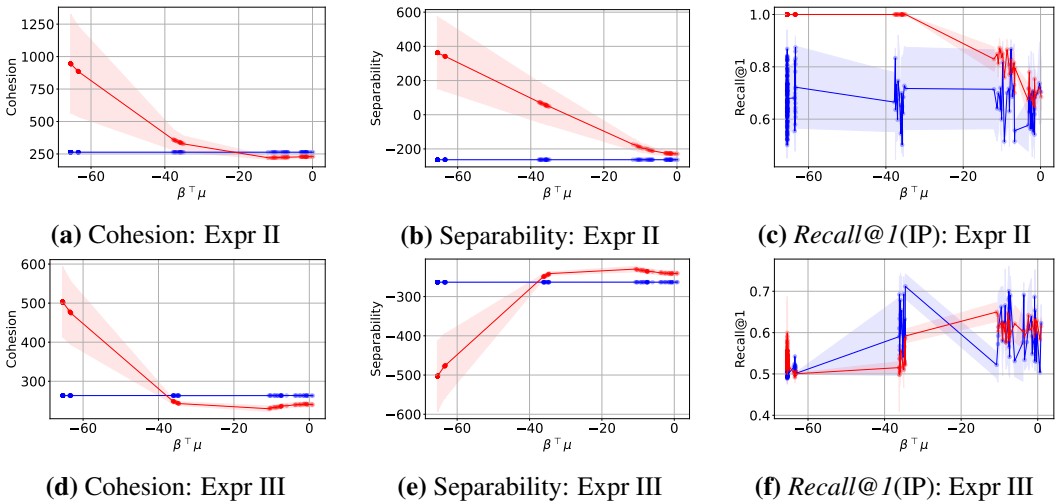

**(a)** Cohesion: Expr II     **(b)** Separability: Expr II     **(c)** *Recall@1*(IP): Expr II

**(d)** Cohesion: Expr III     **(e)** Separability: Expr III     **(f)** *Recall@1*(IP): Expr III

Figure 4: Upper row: In Expr II, all metrics increase as $|\beta^\top \mu|$ increases. Lower row: In Expr III, where two test classes are *assigned* to a single train class, *Recall@1* and *Separability* tend to decrease as $|\beta^\top \mu|$ increases. This aligns with our predictions. In all cases, results using Data 1 are reported. The red line — represents the values after one step of training, i.e., $F$. The blue line — represents the values from initialization, i.e., $F_0$.

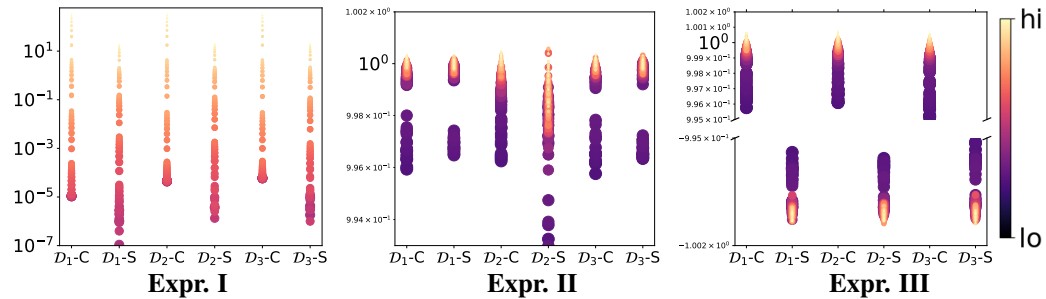

Figure 5: Summary of the synthetic data experiments: The large and dark circles represent low *train-unseen similarity*, while the small and light circles indicate high *train-unseen similarity*. The datasets $D_1$, $D_2$, and $D_3$ correspond to synthetic Data 1, 2, and 3, respectively. C denotes *Cohesion*, and S denotes *Separability*. We scale all measurements using the absolute value at the 85th percentile.

## 5.2 Results of Theory Vaildation: Expr. I, II, III

In Figure 4, we present the results of Expr. II and III using Data 1; uniform ball. These results show that as the *train-unseen similarity* increases, the *Cohesion* consistently increases. On the case of *Separability*, in Expr. II, the two unseen classes are *assigned* to different training classes. As the *train-unseen similarity* increases, the *Separability* also increases accordingly. In contrast, in Expr. III, where the two unseen classes are *assigned* to the same training class, we observe that *Separability* decreases as the *train–unseen similarity* increases. A similar trend is observed for *Recall@1*, which follows the behavior of *Separability*.

Figure 5 summarizes the results across all cases. For individual plots, refer to Appendix M.2. Except for the *Separability* in Expr. III, both *Cohesion* and *Separability* generally increase with *train-unseen similarity* (from big dark points to small light ones), with the drop in Expr. III attributed to duplicated training class assignment. These observations align with our analysis. On top of that, Appendix M.3 provides empirical evidence that the relationship between the clustering criteria and *train-unseen similarity* holds in both multi-step and multi-layer settings. These findings suggest that our theoretical framework remains valid, at least to some extent, beyond the two-layer, one-step setting.

## 5.3 Setup for Practical Vaildation: Expr. IV, V, VI, VII

Building on our analysis, which identified conditions for effective feature clustering under unseen distributions, we conduct experiments to examine whether the theoretical insights hold in practical settings. First, we examine if the positive relationship between *train-unseen similarity* and clustering criteria also holds for semantic similarity and *Recall@1* (Expr. IV). Second, we evaluate whether the theoretical result—that feature *expressiveness* improves only with a larger number of non-orthogonal *spike directions*—holds in practice, by examining whether increasing the number of semantically similar training classes enhances clustering performance (Expr. V, VI). Finally, we test if removing redundantly *assigned* unseen classes improves performance more than random removal (Expr. VII).

We use the datasets CAR (Vehicle, Krause et al. [2013]), CUB (Bird, Wah et al. [2011]), SOP (Product, Song et al. [2015]), and ISC (Clothing, Liu et al. [2016]), referred as *Domain*. We define ImageNet [Deng et al., 2009] subsets for the Vehicle, Bird, Product, and Clothing domains, denoted as I(V), I(B), I(P), and I(C), and refer to them as *sub In1k*. The classes are manually selected and listed in Appendix O. We also define a dataset called *subsampled whole In1k*, which includes all classes. To balance the number of samples per class with those in *Domain* data, we subsample 82, 58, 5, and 6 samples per class in ImageNet variants, respectively. Most experimental configurations follow the approach of Zhai and Wu [2019], a seminal baseline. We utilize ResNet-18 and 50 [He et al., 2015] with random initialization and ImageNet pre-trained weights.

## 5.4 Results of Practical Vaildation: Expr. IV, V, VI, VII

For **Expr. IV**, we train with each *Domain* dataset (CAR, CUB, SOP, and ISC train) and *Domain+sub In1k* dataset (CAR+I(V), CUB+I(B), SOP+I(P), and ISC+I(C)), and then measured how well each model clusters across all test datasets (CAR, CUB, SOP, and ISC test). Figure 6 demonstrates that clustering test data semantically related to training classes is more effective than using classes from unrelated domains. This observation aligns with our theoretical prediction that clustering performance increases as *train-unseen similarity* increases, provided under limited redundant *assignments*. The trend held in 29 out of 32 experiments. Refer to Appendix N.1 for comprehensive results, which also include additional experiments demonstrating the correspondence between *train-unseen similarity* and semantic similarity, as well as between clustering criteria and *Recall@1*.

In **Expr. V**, after learning the *Domain*, *Domain+sub In1k*, and *Domain+subsampled whole In1k*, we measure *Recall@1* for corresponding testsets. Including all ImageNet classes to *Domain* offers no clear advantage over including only related ImageNet classes (Figure 7a). Meanwhile, clustering on CAR and CUB with random initialization improved with *Domain+sub In1k* compared to *Domain*. This supports our results that feature extraction is driven by similar training classes with the input forming non-orthogonal *spike directions*. See Figure 35 and Table 9 for additional results.

In **Expr. VI**, we investigate how gradually increasing the number of semantically related training classes affects clustering performance on each *Domain* test dataset. Each *Domain* training datasets is divided into four stages, introducing 25%, 50%, 75%, and 100% of its classes at Steps 0 through 3, respectively. The classes are randomly sampled. As shown in Figure 7b, increasing the number of semantically related training classes improves performance. Refer to Figure 36 for more results.

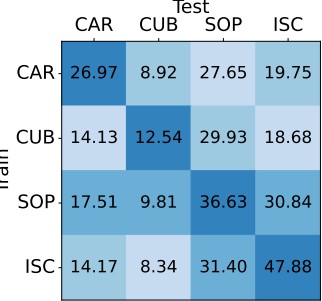

ResNet18 (init)

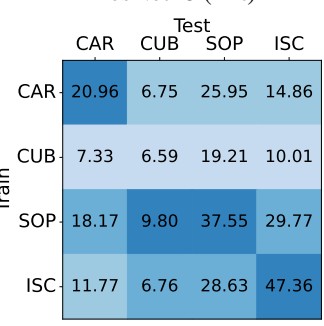

ResNet50 (init)

Figure 6: Expr. IV, *Recall@1* measurements. In most cases, the highest performance is observed when the domains of the training and test datasets correspond.

For **Expr. VII**, during the evaluation phase, removing redundantly *assigned* unseen classes resulted in a 2.31 percentage point improvement in *Recall@1* compared to random removal of the same number of unseen classes, with a maximum improvement of 12.71 percentage points, a maximum drop of -6.04 percentage points, and a success rate of 82.81%. This suggests that duplicate *assignments* hinder clustering, corroborating our analysis of *Separability*. We refer to more details in Appendix N.2.

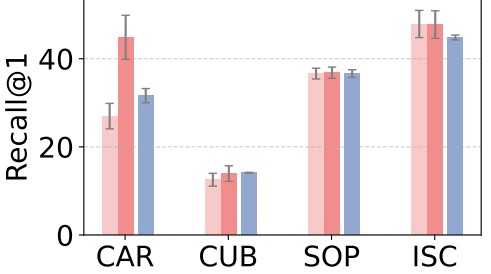
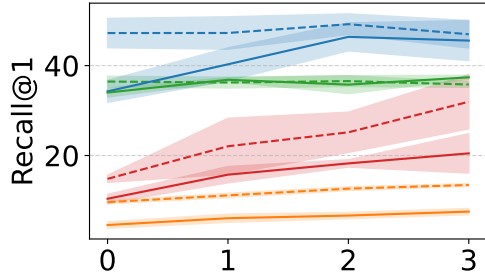

(a) Expr V in ResNet18(init). The pink ▬, red ▬, and blue ▬ represent *Domain*, *Domain+sub In1k*, *Domain+subsampled whole In1k*, respectively.

(b) Expr VI, *Recall@1* values for the CAR, CUB, SOP, and ISC datasets are shown with dashed lines ‑ ‑ for ResNet18 and solid lines ▬ for ResNet50.

Figure 7: Key results of (a) Expr. V and (b) Expr. VI. (a) Adding all ImageNet classes (*Domain + subsampled whole In1k*) yields similar performance gains to adding only domain-relevant classes (*Domain + sub In1k*). (b) Using more classes within the *Domain* improves performance.

## 6  Conclusion

In this study, we investigate the feature learning dynamics of a two-layer classifier in the proportional regime by extending the conventional regression setting and introducing novel technical lemmas, thereby providing a deeper understanding of the mechanisms underlying feature transferability. To our knowledge, this is the first work to quantitatively connect train-unseen alignment with open-set clustering performance, rather than relying solely on hypothesis-level arguments. Our analyses reveal that *Cohesion* increases with higher similarity between training and unseen data, while *Separability* depends not only on this similarity but also on avoiding redundant class *assignments*. Furthermore, our framework characterizes how feature *expressiveness* for unseen classes depends on the number of semantically related training classes, showing that only *spike directions* non-orthogonal to the input contribute meaningfully to feature extraction. Beyond validation on synthetic datasets, our theoretical insights also hold for real-world datasets, where we observe improved clustering performance when training classes are semantically aligned with the target domain.

### 6.1  Limitations and Future Works

While our study provides valuable insights into feature learning and transferability, several important directions remain for future research. *First*, while the Hermite approximation was helpful for our feature analysis, it introduced challenges due to discrepancies between nonlinear neural networks and the polynomial functions used to approximate them. Specifically, we observed that our theoretical predictions tend to align better as the dimensionality increases (Figure 2). In finite dimensions, while a similar trend could be identified in the unnormalized results shown in Figure 3, a precise scaling alignment was lacking. Moreover, in Section 4.1, we hypothesized that the influence of Hermite approximation coefficients would be positive and confirmed this experimentally, although a theoretical verification remains an open avenue for future work. These observations suggest promising directions for future research exploring alternative approximation techniques or more refined analytical methods. *Second*, an important direction for future research is to extend the concepts of Cohesion and Separability to multi-step, multi-layer and multi-class softmax, integrating techniques like normalization and temperature scaling. This could help align our analysis more closely with practical settings, particularly in the context of Neural Collapse research under arbitrary input distributions. *Finally*, in contrast to prior work that emphasizes large-scale datasets on broad domains [Brown et al., 2020, An et al., 2023], our arguments highlight the importance of domain-relevant datasets for effective feature transfer. This perspective aligns with recent research on constructing efficient training coresets [Mirzasoleiman et al., 2020, Paul et al., 2023, Xia et al., 2023, Zhang et al., 2023, Jain et al., 2022] and suggests a potential new direction for representation-aware coreset selection. For instance, reducing the number of training classes or excluding training data that do not contribute to relevant directions can lower computational complexity while still enabling successful clustering and feature transfer.

## Acknowledgements

We are deeply grateful to Inwon Lee for her invaluable support and encouragement. We are also grateful to Won-Seok Choi for insightful scholarly discussions.

This work was partly supported by the IITP ( RS-2021-II212068-AIHub/10%, RS-2021-II211343-GSAI/10%, RS-2022-II220951-LBA/10%, RS-2022-II220953-PICA/10%, RS-2020-II201373-Artificial Intelligence Graduate School Program (Hanyang University)/5%, RS-2023-002206284-Artificial intelligence for prediction of structure-based protein interaction reflecting physicochemical principles/5%), NRF ( RS-2024-00353991-SPARC/9%, RS-2023-00274280-HEI/9%, RS-2023-00244896-Implicit bias of optimization algorithms for robust generalization of deep learning/4%, the BK21 FOUR (Fostering Outstanding Universities for Research) project/3%, NRF-2024S1A5C3A02043653-Socio-Technological Solutions for Bridging the AI Divide: A Blockchain and Federated Learning-Based AI Training Data Platform/3%, RS-2023-00249714/3%, RS-2025-25427337/3%), KEIT (RS-2024-00423940/8%), and Gwangju Metropolitan City (Artificial intelligence industrial convergence cluster development project/8%) grant funded by the Korean government.

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

## A  Computational Resources

For all experiments, we use one NVIDIA GeForce RTX 3090, Intel(R) Xeon(R) Gold 5218 CPU @ 2.30GHz and 128GB CPU memory.

## B  Broader impacts

This paper presents work aimed at advancing the field of Machine Learning. In this research, we analyze the potential for clustering performance improvement through the classification training of many classes. Such an approach may reduce the level of personal data masking required for fine-grained data differentiation, which could trigger new ethical discussions regarding privacy protection. Additionally, to effectively implement this approach, there may be a tendency to collect more data, which can have significant implications for the scale and scope of data collection and data management practices.

## C  Additional Related Works

Table 1: Comparison of Data and Target Settings Between Our Work and Other Feature Learning Studies: We use a non-centered Sub-Gaussian distribution as the data distribution, encompassing all existing studies below. Moreover, the classification problem requires multiple non-identical distributions, so we adopt a non-centered assumption. Unlike these works that assume a teacher structure and study whether the feature extractor can learn it, we analyze what structure the feature extractor learns when trained on arbitrarily labeled targets. This enables us to characterize the criteria when transferring features to a clustering task, making our setting more aligned with real-world applications.

| | Data Distribution | Key Target Function | Miscellaneous |
|---|---|---|---|
| Ba et al. [2022] | $x \sim \mathcal{N}(0, I)$ | $f(w^\top x) + \epsilon$ | $\epsilon \sim \mathcal{N}(0, \sigma^2)$ |
| Dandi et al. [2023] | $x \sim \mathcal{N}(0, I)$ | $f(w_1^\top x, \cdots, w_r^\top x)$ | |
| Cui et al. [2024] | $x \sim \mathcal{N}(0, I)$ | $f(\frac{w^\top x}{\sqrt{d}})$ | |
| Dandi et al. [2024] | $x \sim \mathcal{N}(0, I)$ | $f(w^\top x)$ | |
| Moniri et al. [2024] | $x \sim \mathcal{N}(0, I)$ | $f(w^\top x)$ | |
| Demir and Dogan [2025] | $x \sim \sum \rho \mathcal{N}(\mu, \Sigma)$ | $f(w^\top x \vert c)$ | c is GMM component assign |
| Ba et al. [2023] | $x \sim \mathcal{N}(0, I + d^\beta \mu\mu^\top)$ | $f(\frac{1}{\sqrt{1+d^\beta}} w^\top x)$ | $\beta \in [0, 1)$ |
| Mousavi-Hosseini et al. [2023] | $x \sim \mathcal{N}(0, \frac{I + \kappa\theta\theta^\top}{\kappa+1})$ | $f(\frac{w^\top x}{\|\Sigma^{\frac{1}{2}} w\|}) + \epsilon$ | $\|\theta\| = 1, \epsilon \sim \mathcal{N}(0, \sigma^2)$ |
| Damian et al. [2022] | $x \sim \mathcal{N}(0, I)$ | $f(Ax)$ | |
| Nichani et al. [2023] | $x \sim$ centered $SG$ | $f(x^\top A x)$ | $SG$ denotes Sub-Gaussian |
| Wang et al. [2023] | $x \sim \mathcal{N}(0, I)$ | $f(x^\top A x)$ | |
| Fu et al. [2025] | $x \sim Unif(\mathbb{S}^{d-1})$ | $f(x^\top A_1 x, \cdots, x^\top A_r x)$ | |
| ours | $x \sim$ non-centered $SG$ | $\{1, -1\}$ | The target is an arbitrary label |

**Additional Works on Feature Learning in Shallow Networks**  Dandi et al. [2023] studied how neural networks can learn feature structure step-by-step during gradient descent. To demonstrate that neural networks can effectively learn low-dimensional structures, Ba et al. [2023] and Mousavi-Hosseini et al. [2023] analyzed the dynamics of learning when both the teacher and the data contain low-dimensional structures such as spikes.

Demir and Dogan [2025] assumed a Gaussian mixture model for the data, proposed Conditional Gaussian Equivalence, and stochastically approximated a 2-layer network. While this approach might appear similar to our Theorem 3.3, our proof does not require conditional approximations or constructing a stochastically equivalent model, as we rely solely on the Sub-Gaussian property. This provides a more straightforward approach to analyzing the features of new data.

Building on the work of Damian et al. [2022], several studies analyze the sample complexity of neural networks in learning internal representations while performing regression tasks. These studies consider progressively more complex forms of teacher, e.g., $g(x^\top Ax)$, where the data comes from a centered distribution, as discussed in Nichani et al. [2023], Wang et al. [2023], and Fu et al. [2025]. These works primarily focus on regression problems with a teacher, whereas we assume arbitrarily assigned classification labels without using a teacher function. Furthermore, the transfer learning aspect in these studies examines sample complexity when the function head, e.g., $g$, is changed, but the internal structure, e.g., $x^\top x$, is preserved. In contrast, our work investigates feature transfer in classification settings where new input distributions are introduced without additional learning.

Extending the heuristic results of Cui et al. [2024], Dandi et al. [2024] analyzed the learned feature extractor using an equivalent model to characterize the test error in regression problems and analyze the spectral tail behavior of the feature extractor's covariance matrix. Their study explored the phenomenon of spikes in the spectrum, resulting in heavy-tailed distributions. Similarly, we derive an equivalent model suited to our classification setup and examine the characteristics of clustering errors for unseen distributions (see Section 4.1). We then investigate the behavior of the spike term in the feature extractor (see Section 4.2).

**Feature Transferability in Deep Metric Learning**   The explanation for how Deep Metric Learning learns transferable features towards unseen data remains insufficient. Chopra et al. [2005] suggested that CNNs' robustness to geometric distortions enables the creation of generalizable features. This explanation has been replaced in transformer-based research by the idea that, without the inductive biases of CNNs, transformers are less constrained and thus capable of extracting generalizable features [El-Nouby et al., 2021, Caron et al., 2021].

Following the manifold hypothesis [Chang et al., 2003, Lee et al., 2003, Talwalkar et al., 2008, Goodfellow et al., 2016], Liu et al. [2018], Ermolov et al. [2022] explained that normalized softmax for metric learning works well because the hyperspherical/hyperbolic feature space and the data lie on a manifold. However, these studies do not adequately analyze how features are learned and transferred through classification.

There are empirical studies on classifier transfer learning (not feature transfer like ours, which doesn't require learning) (Li et al. [2024]) and theoretical studies (Lampinen and Ganguli [2019], Tripuraneni et al. [2020]).

**Neural Collapse (NC) and Features Learned by Classifiers**   Studies exist exploring Neural Collapse (NC) and features learned by classifiers under the free variable assumption. Hui et al. [2022] argue that NC does not manifest on test data. Sohoni et al. [2020], Yang et al. [2023] claim that even on training data, NC is not fully realized, with critical fine-grained structures concealed. Notably, Yang et al. [2023] utilized a two-layer network to analyze training data features. Regarding NC on novel data, Galanti et al. [2022] statistically analyzes NC in transfer learning, suggesting that NC generalizes to new samples within training classes and unseen classes with empirical observations. However, their analysis is constrained by focusing on general function spaces rather than specific neural network architectures.

Also, there are papers addressing the alignment phenomenon between the network and the training data in neural network classification tasks (Min et al. [2024]), studies on the increased separability and cohesion of training data features in classification settings (Zarka et al. [2021], Das and Chaudhuri [2019]), and research showing that classifier networks outperform linear classifiers through feature learning (Shi et al. [2022], Frei et al. [2023], Refinetti et al. [2021]).

Furthermore, there are theoretical papers studying optimization properties like implicit margin maximization (Lyu et al. [2021], Wu et al. [2023]) and interpolation (Chatterji et al. [2021], George et al. [2023]).

**MSE for Classification**   Utilizing MSE in classification is as well-established as using softmax-cross entropy, especially in theoretical analyses of classification problems [Han et al., 2022, Zhou et al., 2022].

**Generalization Bound for Metric Learning** Research on the generalization bounds of metric learning related to the U-process we use is also ongoing [Bellet and Habrard, 2015, Huai et al., 2019, Zhou et al., 2024]. However, these studies do not analyze the exact feature learning structure.

# D   Additional Notations

The operator $\mathrm{diag}(\cdot)$ creates a matrix with the input vector elements placed along the diagonal. Let $\mathbf{1}_{\text{condition}}$ be 1 if the condition is true and 0 otherwise. Let $m!$ be factorials of $m$. Let $n!!$ be double factorial. We define $(-1)!! = 0!! = 1$. For $o_{\mathbb{P}}, O_{\mathbb{P}}, \Theta_{\mathbb{P}}$ notations, we follow Moniri et al. [2024] $\|\cdot\|_F$ is the Frobenius norm. $\|\cdot\|_{\infty}$ is the infinity norm. $\|\cdot\|_{\psi_2}$ is orlicz-2 norm $e^{(i)}$ Standard basis vector with 1 at position $i$. $\lfloor n/2 \rfloor$ denotes the floor of $n/2$. $\Gamma(z)$ is the Gamma function.

**Additional information on Hermite Polynomials** We employ the probabilist's Hermite polynomials [Szegő, 1975, Bienstman, 2023, Moniri et al., 2024]. We denote $H_k(x)$ as $k$-th Hermite polynomial.

The $n$-th Hermite polynomials, $H_n(\cdot)$, are defined by the recurrence relation: $H_{n+1}(x) = xH_n(x) - nH_{n-1}(x)$, for $n \geq 1$, with the initial conditions $H_0(x) = 1, H_1(x) = x$. Using this recurrence, we have $H_2(x) = x^2 - 1, H_3(x) = x^3 - 3x, \cdots$.

Hermite polynomials can be represented in the following explicit form:

$$H_n(x) = (-1)^n e^{\frac{x^2}{2}} \frac{d^n}{dx^n} e^{-\frac{x^2}{2}}.$$

for $n \in \mathbb{N}_0$. Lastly, there is another expression:

$$H_n(x) = n! \sum_{m=0}^{\lfloor \frac{n}{2} \rfloor} \frac{(-1)^m}{m!(n-2m)!} \frac{x^{n-2m}}{2^m} \tag{8}$$

The probabilist's Hermite polynomials form an orthogonal set with respect to the standard normal weight function $\phi(x) = \frac{1}{\sqrt{2\pi}} e^{-\frac{x^2}{2}}$ on the interval $(-\infty, \infty)$. Their orthogonality condition is given by:

$$\int_{-\infty}^{\infty} H_m(x) H_n(x) \frac{1}{\sqrt{2\pi}} e^{-\frac{x^2}{2}} \, dx = n! \mathbf{1}_{\text{m=n}}.$$

## D.1   Hermite Coefficients of shifted ReLU

One of the activation functions that satisfy our condition 2.1 is shifted ReLU,

$$\sigma(x) = \max(0, x) - \frac{1}{\sqrt{2\pi}}. \tag{9}$$

This enables a Hermite decomposition with coefficients given by

$$c_n = \frac{1}{n!} \mathbb{E}_z[\sigma(z) H_n(z)]. \tag{10}$$

Then the zeroth coefficient is calculated as

$$c_0 = \mathbb{E}_z[\sigma(z) \times 1] = \mathbb{E}_z[\max(0, x)] - \frac{1}{\sqrt{2\pi}}$$
$$= \int_0^{\infty} x\phi(x)dx - \frac{1}{\sqrt{2\pi}} = 0 \tag{11}$$

By the way, if $n \neq 0$, $\mathbb{E}[\frac{1}{\sqrt{2\pi}} \times H_n] = \frac{1}{\sqrt{2\pi}} \mathbb{E}[1 \times H_n] = \frac{1}{\sqrt{2\pi}} \mathbb{E}[H_0 \times H_n] = 0$ by orthogonality. Thus, the shift only affects $n = 0$.

The coefficient $c_n$ of the expansion of Shifted-ReLU is defined as:

$$c_n = \begin{cases} 0, & \text{if } n = 0, \\ \sum_{m=0}^{\lfloor n/2 \rfloor} \frac{(-1)^m \cdot 2^{\frac{n-2m}{2} - m} \cdot \Gamma\left(\frac{n-2m+2}{2}\right)}{m! \cdot (n-2m)! \cdot \sqrt{2\pi}}, & \text{otherwise.} \end{cases} \quad (12)$$

We directly calculate Equation (12) and obtain the following result in Figure 8.

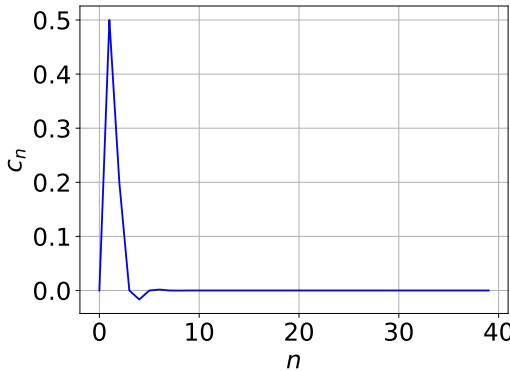

Figure 8: Hermite coefficient of shifted ReLU

# E  Proof of Theorem 3.1

This section builds upon the proof structure introduced by Ba et al. [2022], adapting and extending their technique to our classification learning framework. Unlike their assumption of centered Gaussian training data, we consider non-centered Sub-Gaussian data distributions. In this process, we apply a novel approach involving the concentration of the operator norm on a random matrix. Also, since our framework is not in a teacher-student setting, we use class labels instead of a teacher function.

**Gradient Decomposition**   We decompose the gradient (Equation (2)) using Hermite decomposition, which allows us to extract the essential rank-one matrix structure for each $ij$-th classification problem. Denote $\sigma' = c_1 + \sigma'_\perp$ for the Hermite decomposition result.

$$G_{ij} = \frac{c_1}{n} X_{ij}^\top y a_{ij}^\top + \frac{1}{n} X_{ij}^\top y a_{ij}^\top \odot \sigma'_\perp(X_{ij}W_0) - \frac{1}{n} X_{ij}^\top \sigma(X_{ij}W_0)(a_{ij}a_{ij}^\top) \odot \sigma'(X_{ij}W_0). \quad (13)$$

Denote

$$\begin{aligned} \mathbb{A}_{ij} &\triangleq \frac{c_1}{n} X_{ij}^\top y a_{ij}^\top \\ \mathbb{B}_{ij} &\triangleq \frac{1}{n} X_{ij}^\top y a_{ij}^\top \odot \sigma'_\perp(X_{ij}W_0) \\ \mathbb{C}_{ij} &\triangleq -\frac{1}{n} X_{ij}^\top \sigma(X_{ij}W_0)(a_{ij}a_{ij}^\top) \odot \sigma'(X_{ij}W_0). \end{aligned} \quad (14)$$

We derive the norm bound for the terms $\mathbb{A}_{ij}$, $\mathbb{B}_{ij}$, and $\mathbb{C}_{ij}$ in Lemma E.1. We establish the following Theorem 3.1 using these bounds.

We will omit the subscript $ij$ since it does not cause any confusion in Lemma E.1, Remark E.2, and Proof of Lemma E.1. The following statements hold for $\forall ij$. For the aforementioned $\mathbb{A}$, $\mathbb{B}$, and $\mathbb{C}$, we obtain bounds for each operator norm as follows.

**Lemma E.1** (Probabilistic Bounds on Gradient Decomposition Components).

$$\mathbb{P}\bigg(\|\mathbb{A}\| \le C\big(\frac{1}{\sqrt{\mathbf{N}}} - C\frac{\sqrt{\mathbf{d}}}{\sqrt{\mathbf{nN}}}\big)\bigg) \le 2\big(e^{-c\mathbf{N}} + e^{-c\mathbf{n}}\big)$$

$$\mathbb{P}\bigg(\|\mathbb{B}\| \ge \frac{C}{\mathbf{n}\sqrt{\mathbf{Nd}}}(\sqrt{\mathbf{n}} + \sqrt{\mathbf{d}})(\sqrt{\mathbf{n}} + \sqrt{\mathbf{N}})\log\mathbf{N}\bigg) \le C\big(e^{-c\mathbf{N}} + e^{-c\mathbf{d}} + \mathbf{N}e^{-c\log^2\mathbf{n}} + e^{-(\sqrt{\mathbf{n}}+\sqrt{\mathbf{d}})^2}\big)$$

$$\mathbb{P}\bigg(\|\mathbb{C}\| \ge \frac{C}{\sqrt{\mathbf{nN}}}(2\sqrt{\mathbf{d}} + \sqrt{\mathbf{n}})\log\mathbf{n}\log\mathbf{N}\bigg) \le 2\big(\mathbf{n}e^{-c\mathbf{d}} + \mathbf{n}e^{-c\log^2\mathbf{n}} + \mathbf{N}e^{-c\log^2\mathbf{n}}\big). \tag{15}$$

*Remark* E.2 (Reformulated Lemma E.1). In the proportional regime, these quantities can be interchanged to a constant as $\mathbf{n}, \mathbf{d}, \mathbf{N} \to \infty$. Thus, Lemma E.1 is reformulated as follows

$$\mathbb{P}(\|\mathbb{A}\| \le \kappa/\sqrt{\mathbf{n}}) \le Ce^{-c\mathbf{n}})$$
$$\mathbb{P}\bigg(\|\mathbb{B}\| \ge \frac{C\log\mathbf{N}}{\mathbf{n}}\bigg) \le C(e^{-c\mathbf{n}} + \mathbf{n}e^{-c\log^2\mathbf{n}})$$
$$\mathbb{P}\bigg(\|\mathbb{C}\| \ge \frac{C\log^2\mathbf{N}}{\mathbf{n}}\bigg) \le C(\mathbf{n}e^{-c\mathbf{n}} + \mathbf{n}e^{-c\log^2\mathbf{n}}) \tag{16}$$

Also, for the gradient, we have

$$\|G\| = \|\mathbb{A} + \mathbb{B} + \mathbb{C}\| \le \|\mathbb{A}\| + \|\mathbb{B}\| + \|\mathbb{C}\| = O_{\mathbb{P}}(\frac{1}{\sqrt{\mathbf{n}}} + \frac{\log\mathbf{n}}{\mathbf{n}} + \frac{\log^2\mathbf{n}}{\mathbf{n}}) = O_{\mathbb{P}}(\frac{1}{\sqrt{\mathbf{n}}}) \tag{17}$$

*Proof of Theorem 3.1.* Using $\|G_{ij} - \mathbb{A}_{ij}\| = \|\mathbb{B}_{ij} + \mathbb{C}_{ij}\| \le \|\mathbb{B}_{ij}\| + \|\mathbb{C}_{ij}\|$ and Remark E.2

$$\mathbb{P}\bigg(\|G_{ij} - \mathbb{A}_{ij}\| \ge C\frac{\log^2\mathbf{n}}{\mathbf{n}}\bigg) \le \mathbb{P}\bigg(\|G_{ij} - \mathbb{A}_{ij}\| \ge C(\frac{\log\mathbf{n}}{\mathbf{n}} + \frac{\log^2\mathbf{n}}{\mathbf{n}})\bigg) \le C\mathbf{n}e^{-c\log^2\mathbf{n}}. \tag{18}$$

Therefore, almost surely, in the proportional limit,

$$\|G_{ij} - \mathbb{A}_{ij}\| \le C\frac{\log^2\mathbf{n}}{\mathbf{n}} = \frac{\kappa}{\sqrt{\mathbf{n}}}\frac{C}{\kappa}\frac{\log^2\mathbf{n}}{\sqrt{\mathbf{n}}} \le \|\mathbb{A}_{ij}\|\frac{C}{\kappa}\frac{\log^2\mathbf{n}}{\sqrt{\mathbf{n}}} \le \kappa'\frac{\log^2\mathbf{n}}{\sqrt{\mathbf{n}}}\big(\|G_{ij}\| + \|G_{ij} - \mathbb{A}_{ij}\|\big). \tag{19}$$

We get $(1 - \kappa'\frac{log^2\mathbf{n}}{\sqrt{\mathbf{n}}})\|G_{ij} - \mathbb{A}\| \le \kappa'\frac{log^2\mathbf{n}}{\sqrt{\mathbf{n}}}\|G_{ij}\|$. For large enough $\mathbf{n}$ for $1 - \kappa'\frac{log^2\mathbf{n}}{\sqrt{\mathbf{n}}} \ge \frac{1}{2}$,

$$\|G_{ij} - \mathbb{A}_{ij}\| \le \kappa'\frac{\log^2\mathbf{n}}{\sqrt{\mathbf{n}}}\|G_{ij}\| \le C\frac{\log^2\mathbf{n}}{\mathbf{n}} \tag{20}$$

Sum up for $\forall ij$,

$$\|G - \sum_{i<j}\mathbb{A}_{ij}\| = \|\sum_{i<j}G_{ij} - \mathbb{A}_{ij}\| \le \sum_{i<j}\|G_{ij} - \mathbb{A}_{ij}\| \le C\frac{\log^2\mathbf{n}}{\mathbf{n}} \tag{21}$$

with probability $1 - C(\mathbf{n}e^{-c\log^2\mathbf{n}} + e^{-c\mathbf{n}})$. $\qquad\square$

## E.1 Proof of Lemma E.1

Use the norm bound of sub-Gaussian vectors and the spectral bound of sub-Gaussian matrices as the main idea of the proof.

*Proof of Lemma E.1 ($\mathbb{A}$).* We obtain

$$\mathbb{A} = \frac{c_1}{\mathbf{n}\sqrt{\mathbf{N}}}X^\top ya^\top. \tag{22}$$

Then, we can find an explicit notation of the norm as

$$\|\mathbb{A}\| = \frac{c_1}{\mathbf{n}\sqrt{\mathbf{N}}}\|X^\top ya^\top\| = \frac{c_1}{\mathbf{n}\sqrt{\mathbf{N}}}\|X^\top y\|_2\|a\|_2 = \frac{c_1}{\mathbf{n}\sqrt{\mathbf{N}}}\big(y^\top XX^\top y\big)^{1/2}\|a\|_2 \tag{23}$$

We probabilistically bound the two components of Equation (23) in the following two paragraphs, respectively.

$\|a\|_2$ **study** By definition, $a \sim \mathbf{N}(0, \frac{1}{\mathbf{N}})$, so $\sqrt{\mathbf{N}}a[i]$ is a Sub-Gaussian. Use Thm 3.1.1 in Vershynin [2018],

$$\mathbb{P}\left(\left|\|\sqrt{\mathbf{N}}a\| - \sqrt{\mathbf{N}}\right| \geq t\right) \leq 2e^{-ct^2} \quad \text{let } t = \sqrt{\mathbf{N}} \tag{24}$$

$$\mathbb{P}(\|a\|_2 \leq 1) \leq 2e^{-c\mathbf{N}}$$

$\left(y^\top X X^\top y\right)^{1/2}$ **study** Note that the $U, V$ matrices resulting from the SVD belong to the $O$-group, so there is no length transformation.

$$y^\top X X^\top y = \|X^\top y\|_2^2 = \|U\Sigma V^\top y\|_2^2 = \|\Sigma V^\top y_1\|$$
$$= \sum_i \sigma_i^2 |V^\top y_i|^2 \geq \sigma_{\min}^2 \sum_i |V^\top y_i|^2 = \sigma_{\min}^2 \|y\|_2^2 = \mathbf{n}\sigma_{\min}^2 \tag{25}$$

We get $\left(y^\top X X^\top y\right)^{1/2} \geq \sqrt{\mathbf{n}}\sigma_{\min}$. $\sigma_{\min}$ is the singular value of $X$ which is an anistropic Sub-Gaussian matrix. With the result of Remark 1.2 in Liaw et al. [2016],

$$\mathbb{P}\sigma_{\min} \leq (\sqrt{\mathbf{n}} - c\sqrt{\mathbf{d}})) \leq e^{-\mathbf{n}}. \tag{26}$$

Therefore, $\mathbb{P}(\|\mathbb{A}\| \leq C(\frac{1}{\sqrt{\mathbf{N}}} - C\frac{\sqrt{d}}{\sqrt{\mathbf{nN}}})) \leq 2(e^{-c\mathbf{N}} + e^{-c\mathbf{n}})$. $\qquad\square$

*Proof of Lemma E.1 ($\mathbb{B}$).* Remark on the definition of $\mathbb{B}$.

$$\mathbb{B} = \frac{1}{\mathbf{n}\sqrt{\mathbf{N}}}X^\top y a^\top \odot \sigma'_\perp(XW_0). \tag{27}$$

$$\|\mathbb{B}\| \leq \frac{1}{\mathbf{n}\sqrt{\mathbf{N}}}\|X\| \, \|y a^\top \odot \sigma'_\perp(XW_0)\| \quad \text{(by Property of Operator Norm)}$$
$$\leq \frac{1}{\mathbf{n}\sqrt{\mathbf{N}}}\|X\|\|y\|_\infty \, \|\sigma'_\perp(XW_0)\| \, \|a\|_\infty \quad \text{(by Fact J.2)} \tag{28}$$
$$= \frac{1}{\mathbf{n}\sqrt{\mathbf{N}}}\|X\| \, \|\sigma'_\perp(XW_0)\|\|a\|_\infty \quad (\|y\|_\infty = 1)$$

We probabilistically bound the three components in the last line of Equation (28) in the following paragraphs, respectively.

$\|\sigma'_\perp(XW_0)\|$ **study** Use the result of D.4 in Fan and Wang [2020], which is held for orthogonal columns. $X$ is sampled from continuous support distribution $c_1, c_2$. The first vector is linearly independent with probability one due to the continuous support of its distribution. For the second vector, which is drawn independently, the probability that it lies in the span of the first vector is 0, as it also has a continuous density. This reasoning extends to $\mathbf{n}$ vectors, implying that, with high probability, they are orthogonal or nearly orthogonal because no vector falls into the span of the others. Thus, $\forall B > 0$, the following holds.

$$\mathbb{P}(\{\|\sigma'_\perp\| \geq C(\sqrt{\mathbf{n}} + \sqrt{\mathbf{N}})\lambda_\sigma B\}, \mathscr{A}_B) \leq 2e^{-c\mathbf{N}}$$
$$\mathscr{A}_B = \{\{\|W_0\| \leq B\} \cup \{\sum_{i=1}^{\mathbf{N}}(\|W_{0,i}\|^2 - 1)^2 \leq B^2\}\}. \tag{29}$$

Therefore,

$$\mathbb{P}(\|\sigma'_\perp\| \geq C(\sqrt{\mathbf{n}} + \sqrt{\mathbf{N}})\lambda_\sigma B) \leq 2e^{-c\mathbf{N}} + \mathbb{P}(\mathscr{A}_B^c) \tag{30}$$

$\mathbb{P}(\mathscr{A}_B)$ **study** We choose $t = C\sqrt{\frac{\mathbf{d}}{\mathbf{N}}}$, $B = C\sqrt{\frac{\mathbf{d}}{\mathbf{N}}}$.

**case of** $\|W_{0,i}\| \leq B$ By Lemma I.3,

$$\mathbb{P}(\|\sqrt{\mathbf{N}}W_0\| \geq 2\sqrt{\mathbf{N}} + \sqrt{\mathbf{d}}) \leq 2e^{-c\mathbf{N}} \Rightarrow \quad \mathbb{P}(\|W_0\| \geq C\sqrt{\frac{\mathbf{d}}{\mathbf{N}}}) \leq 2e^{-c\mathbf{N}} \tag{31}$$

Therefore, $\|W_0\| \leq B$ at least w.p. $1 - 2e^{-c\mathbf{N}}$

**case of** $\sum_{i=1}^{\mathbf{N}}(\|W_{0,i}\|^2-1)^2 \le B^2$   By definition, $\|W_{0,i}\|^2 = 1$, so $0 \le B^2$, trivially.
We know $\mathbb{P}(\mathscr{A}_B^c) \le 2e^{-c\mathbf{N}}$.

$$\mathbb{P}(\|\sigma'_\perp\| \ge C(\sqrt{\mathbf{n}}+\sqrt{\mathbf{N}})\sqrt{\frac{\mathbf{d}}{\mathbf{N}}}) \le 2e^{-c\mathbf{N}} \tag{32}$$

**Studies of the remaining components**   Use Lemma J.6, and I.3,

$$\|\sigma'_\perp\| \le C\left(\sqrt{\frac{\mathbf{nN}}{\mathbf{d}}} + \sqrt{\frac{\mathbf{N}^2}{\mathbf{d}}}\right) \qquad \text{w.p. } 1 - C(e^{-c\mathbf{N}} + e^{-c\mathbf{d}}) \tag{33}$$

$$\|a\|_\infty \le \frac{t}{\sqrt{\mathbf{N}}} \qquad \text{w.p. } 1 - 2\mathbf{N}e^{-ct^2} \tag{34}$$

$$\|X\| \le \sqrt{\mathbf{n}} + \sqrt{\mathbf{d}} + t' \qquad \text{w.p. } 1 - 2e^{-ct'^2}. \tag{35}$$

In summary, we get

$$\|\mathbb{B}\| \le \frac{C}{\mathbf{n}\sqrt{\mathbf{N}}}(\sqrt{\mathbf{n}}+\sqrt{\mathbf{d}}+t')\left(\sqrt{\frac{\mathbf{nN}}{\mathbf{d}}}+\sqrt{\frac{\mathbf{N}^2}{\mathbf{d}}}\right)\frac{t}{\sqrt{\mathbf{N}}}$$

let $t = \log \mathbf{n}$, $t' = \sqrt{\mathbf{n}}+\sqrt{\mathbf{d}}$

$$\mathbb{P}(\|\mathbb{B}\| \ge \frac{C}{\mathbf{n}\sqrt{\mathbf{Nd}}}(\sqrt{\mathbf{n}}+\sqrt{\mathbf{d}})(\sqrt{\mathbf{n}}+\sqrt{\mathbf{N}})\log N) \le C\big(e^{-c\mathbf{N}} + e^{-c\mathbf{d}} + \mathbf{N}e^{-c\log^2\mathbf{n}} + e^{-(\sqrt{\mathbf{n}}+\sqrt{\mathbf{d}})^2}\big). \tag{36}$$

$\square$

*Proof of Lemma E.1 ($\mathbb{C}$).*  Remark on the definition of $\mathbb{C}$,

$$\mathbb{C} = -\frac{1}{\mathbf{nN}}X^\top\sigma(XW_0)\big(aa^\top\big)\odot\sigma'(XW_0). \tag{37}$$

We can bound the norm as follows,

$$\begin{aligned}
\|\mathbb{C}\| &\le \frac{1}{\mathbf{nN}}\|X\|\|\sigma aa^\top \odot \sigma'\| \quad \text{(by Property of Operator Norm)} \\
&\le \frac{1}{\mathbf{nN}}\|X\|\|\sigma a\|_\infty\|a\|_\infty\|\sigma'\|_F \quad \text{(by Fact J.2)} \\
&\le \frac{\lambda_\sigma}{\sqrt{\mathbf{nN}}}\|X\|\|\sigma a\|_\infty\|a\|_\infty \quad (\sigma' \text{ is bounded, so } \|\sigma'\|_F \le \lambda_\sigma\sqrt{\mathbf{nN}}).
\end{aligned} \tag{38}$$

We probabilistically bound the $\|\sigma a\|_\infty$ to finish the proof.

**Control of** $\|\sigma a\|_\infty$   Let $t = \sqrt{\mathbf{d}}$. Given $X$ s.t. $\mathbb{P}(|X_i - \sqrt{\mathbf{d}}| \ge \sqrt{\mathbf{d}}) \le 2e^{-ct^2}$, consider one element $\sigma\big(X_j^\top W_0\big)a = \sum_i^{\mathbf{N}}a_i\sigma\big(X_j^\top W_0[i]\big)$.

We know $a_i, \sqrt{\mathbf{n}}W_{0,i}$ is an independent centered Sub-Gaussian, and use Fact J.3, then $\sigma\big(\frac{X_j^\top}{\sqrt{\mathbf{N}}}\sqrt{\mathbf{N}}W_0\big)a$ is sub-exponential and mean is zero, since $\|a\sigma(x_j^\top W_{0,i})\|_{\psi_1} \le \|a\|_{\psi_2}\|\sigma(x_j^\top W_{0,i})\|_{\psi_2} < \infty$. Apply the Bernstein inequality for the sub-exponential,

$$\mathbb{P}(|\sigma(X_j^\top a)| \ge \log \mathbf{n} \text{ given } \{|X_j - \sqrt{\mathbf{d}}| \ge \sqrt{\mathbf{d}}\}) \le 2e^{-c\log^2\mathbf{n}}. \tag{39}$$

For every element $\|\sigma(XW_0)a\|_\infty \le \log \mathbf{n}$ w.p. $1 - [2\mathbf{n}e^{-c\log^2\mathbf{n}+2\mathbf{n}e^{-c\mathbf{d}}}]$

By Lemma J.6 $\mathbb{P}(\|a\|_\infty \le t/\sqrt{\mathbf{N}}) \ge 1 - 2\mathbf{N}e^{-ct^2}$, and Lemma I.3 with $t = \sqrt{\mathbf{d}}$

$$\mathbb{P}\left(\|\mathbb{C}\| \ge \frac{C}{\sqrt{\mathbf{nN}}}(2\sqrt{\mathbf{d}}+\sqrt{\mathbf{n}})\log\mathbf{n}\log\mathbf{N}\right) \le 2\big(\mathbf{n}e^{-c\mathbf{d}} + \mathbf{n}e^{-c\log^2\mathbf{n}} + \mathbf{N}e^{-c\log^2\mathbf{n}}\big). \tag{40}$$

$\square$

# F  Proof of Theorem 3.3

**Corollary F.1** (Corollary of Theorem 3.1). *By Lemma J.4, we have w.p.* $1 - o(1)$,

$$\|XG - c_1 X \sum_{i<j} \beta_{ij} a_{ij}^\top\| = O(\frac{\log^2 \mathbf{n}}{\mathbf{n}} \cdot \sqrt{\mathbf{n}}) = O(\frac{\log^2 \mathbf{n}}{\sqrt{\mathbf{n}}}) \tag{41}$$

*Remark* F.2 (Updata Rule). $W_1 = W_0 + G$, so $\tilde{X}W_1 = \tilde{X}W_0 + \tilde{X}G$. $\tilde{X} \in \mathbb{R}^{\mathbf{n} \times \mathbf{d}}$ is sub-Gaussian random matrix

We generalize Corollary F.1, which provides a monomial approximation of the data-gradient product in polynomial form, into Lemma F.3. This result approximates the Hadamard power of the data-gradient product, enabling the decomposition of the dominant feature.

**Lemma F.3** (Polynomial Approximation of Data-Gradient Product). *For any $k \in \mathbb{N}$, sufficiently large $\mathbf{n}$, and w.p. 1 - o(1),*

$$\|(\tilde{X}G)^{ok} - c_1^k (\tilde{X} \sum_{i<j} \beta_{ij} a_{ij}^\top)^{ok}\| = O(\mathbf{n}^{-\frac{k}{2}} \log^{2k} \mathbf{n}) \tag{42}$$

Lemma F.4 below provides norm bounds for a single Hermite polynomial, enabling the Hermite decomposition of the activation function to be bounded in the proof of Theorem 3.3.

**Lemma F.4** (Operator Norm Bound of $H(\tilde{X}W_0)$). *Following condition in Section 2, Assume event* $\Omega = \sup_{k \geq 1} \|(W_0 W_0^\top)^{ok}\|_{op} \leq C$ *occurs, the following statement holds.*

$$\|H_j(\tilde{X}W_0)\|_{op} = O_{\mathbb{P}}(\sqrt{\mathbf{n}} \log^{\frac{3}{2}} \mathbf{n} \sqrt{j!}) \tag{43}$$

We introduce the following lemma to prove Lemma F.4.

**Lemma F.5** (Bound on $\mathbb{E}[\|H(\tilde{X}W_0)H(\tilde{X}W_0)^\top\|]$). *Following condition of Lemma F.4,*

$$\mathbb{E}\|H_j(\tilde{X}W_0)H_j(\tilde{X}W_0)^\top\|_{op} \leq Cj! \tag{44}$$

The proofs of the lemmas above can be found in Section F.1.

Finally, we prove Theorem 3.3.

*Proof of Theorem 3.3.* Let $L = O(\log \mathbf{n})$.

Denote $\sigma_L(z) = \sum_{k=1}^L c_k H_k(z)$, $F_L = \sigma_L(\tilde{X}W)$ and $F_{L,0} = \sigma_L(\tilde{X}W_0)$, then,

$$F = F_L + (\sigma - \sigma_L)(\tilde{X}W). \tag{45}$$

**Bound** $(\sigma - \sigma_L)(\tilde{X}W)$ **term**   Using Lemma F.5, $w$ in assumption 2.1, w.p. $1 - o(1)$

$$\|E[(\sigma - \sigma_L)(\tilde{X}W_0)(\sigma - \sigma_L)(\tilde{X}W_0)^\top]\|$$
$$\leq C \sum_{k=L+1}^\infty k! c_k^2 \leq C \sum_{k=L+1}^\infty k^{-3-w} \leq C \int_L^\infty k^{-\frac{3}{2}-w} dk \leq CL^{-2-w}. \tag{46}$$

Therefore, following same proof technique as Lemma F.4, F.5, and J.7 one can obtain follows:

$$\|(\sigma - \sigma_L)(\tilde{X}W_0)\|_{op} = o_{\mathbb{P}}(\sqrt{\mathbf{n} \log^3 \mathbf{n}} \cdot L^{-2-w}) = o_{\mathbb{P}}(\sqrt{\mathbf{n}}) \tag{47}$$

Also, because $\|W\|_{op} = O(1)$,

$$\|(\sigma - \sigma_L)(\tilde{X}W)\|_{op} = o(\sqrt{\mathbf{n} \log^3 \mathbf{n}} \cdot L^{-2-w}) = o_{\mathbb{P}}(\sqrt{\mathbf{n}}) \tag{48}$$

**Bound $F^L$ term**  We write $F_L + F_{0,L} = F^L + F_0^L$, then

$$F_L = F_{L,0} + \sum_{k=1}^{L} c_k (H_k(\tilde{X}W) - H_k(\tilde{X}W_0)). \tag{49}$$

Thus, We have to study $H_k(\tilde{X}W) - H_k(\tilde{X}W_0)$ term.

$$
\begin{aligned}
&H_k(\tilde{X}W) - H_k(\tilde{X}W_0) \\
&= H_k(\tilde{X}W_0^\top + \tilde{X}G^\top) - H_k(\tilde{X}W_0) \\
&= (\tilde{X}G)^{ok} + \sum_{j=1}^{k-1} \binom{k}{j} H_{k-j}(\tilde{X}W_0) \odot (\tilde{X}G)^{oj}. \quad \text{(by Lemma J.1)}
\end{aligned}
\tag{50}
$$

Substituting Equation (50) into Equation (49), we expand the resulting expression and complete the proof by bounding $\Delta_1$, $\Delta_2$, and $\Delta_3$, making use of the facts that $\|F_{0,L}\| = \Theta(\sqrt{\mathbf{n}})$ (as shown by Moniri et al. [2024]) and that $\left\| \sum_{k=1}^{L} c_1^k c_k \left( \tilde{X} \sum_{i<j} \beta_{ij} a_{ij}^\top \right)^{ok} \right\| = \Omega(\sqrt{\mathbf{n}})$.

$$
\begin{aligned}
F^L &= F_0^L + \sum_{k=1}^{L} c_k (\tilde{X}G)^{ok} + \sum_{k=1}^{L}\sum_{j=1}^{k-1} c_k \binom{k}{j} H_{k-j}(\tilde{X}W_0) \circ (\tilde{X}G)^{oj} \\
&= F_0^L + \sum_{k=1}^{L} c_1^k c_k (\tilde{X} \sum_{i<j} \beta_{ij} a_{ij}^\top)^{ok} \\
\Delta_1 &\left[ 
\begin{aligned}
&- \sum_{k=1}^{L} c_1^k c_k (\tilde{X} \sum_{i<j} \beta_{ij} a_{ij}^\top)^{ok} \\
&+ \sum_{k=1}^{L} c_k (\tilde{X}G)^{ok}
\end{aligned}
\right. \\
\Delta_2 &\left[
\begin{aligned}
&+ \sum_{k=1}^{L}\sum_{j=1}^{k-1} c_k \binom{k}{j} H_{k-j}(\tilde{X}W_0) \odot (\tilde{X}G)^{oj} \\
&- \sum_{k=1}^{L}\sum_{j=1}^{k-1} c_1^j c_k \binom{k}{j} H_{k-j}(\tilde{X}W_0) \odot [\tilde{X} \sum_{i<j} \beta_{ij} a_{ij}^\top]^{oj}
\end{aligned}
\right. \\
\Delta_3 &\left[ + \sum_{k=1}^{L}\sum_{j=1}^{k-1} c_1^j c_k \binom{k}{j} H_{k-j}(\tilde{X}W_0) \odot [\tilde{X} \sum_{i<j} \beta_{ij} a_{ij}^\top]^{oj} \right.
\end{aligned}
\tag{51}
$$

For $\Delta_1, \Delta_2, \Delta_3$, it is derived as follows:

$$
\begin{aligned}
\|\Delta_1\| &\leq \sum_{k=1}^{L} c_k \|(\tilde{X}G)^{ok} - c_1^k (\tilde{X} \sum_{i<j} \beta_{ij} a_{ij}^\top)^{ok}\| \quad \text{(by Triangle Inequality)} \\
&\leq C \sum_{K=1}^{L} \log^{2k} \mathbf{n} \cdot \mathbf{n}^{-\frac{k}{2}} = O(\frac{\log^2 \mathbf{n}}{\sqrt{\mathbf{n}}}) = o(1) \quad \text{(by Lemma F.3)}
\end{aligned}
\tag{52}
$$

$$\|\Delta_2\| \le \sum_{k=1}^{L} \sum_{j=1}^{k-1} c_k \binom{k}{j} \|H_{k-j}(\tilde{X} W_0^\top) \odot [(\tilde{X} G^\top)^{\circ j} - c_1^j [\tilde{X} \sum_{i<j} \beta_{ij} a_{ij}^\top]^{\circ j}] \| \quad \text{(by Triangle Inequality)}$$

$$\le C \sum_{k=1}^{L} \sum_{j=1}^{k-1} \|H_{k-j}(\tilde{X} W_0^\top)\| \|(\tilde{X} G^\top)^{\circ j} - c_1^j (\tilde{X} \sum_{i<j} \beta_{ij} a_{ij}^\top)^{\circ j}\| \quad \text{(by Fact J.5)}$$

$$\le C \sum_{k=1}^{L} \sum_{j=1}^{k-1} \sqrt{\mathbf{n}} \log^{\frac{3}{2}} \mathbf{n} \sqrt{j!} \cdot \mathbf{n}^{-\frac{j}{2}} \log^{2j} \mathbf{n} \quad \text{(by Lemma F.4 and Lemma F.3)}$$

$$\le C \sum_{k=1}^{L} \sum_{j=1}^{k-1} \frac{\sqrt{\mathbf{n}} \sqrt{j!} \log^{\frac{3}{2}+2j} \mathbf{n}}{\sqrt{\mathbf{n}^j}} = O(\log^{\frac{7}{2}} \mathbf{n})$$

(53)

$$\|\Delta_3\| \le C \sum_{k=1}^{L} \sum_{j=1}^{k-1} \|H_{k-j}(\tilde{X} W_0) \odot [\tilde{X} \sum_{i<j} \beta_{ij} a_{ij}^\top]^{\circ j}\| \quad \text{(by Triangle Inequality)}$$

$$\le C \sum_{k=1}^{L} \sum_{j=1}^{k-1} \|\text{diag}(\tilde{X}\beta)^{\circ j}\| \|H_{k-j}(\tilde{X} W_0)\| \|\text{diag}(a)^{\circ j}\| \quad \text{(by Fact J.2)}$$

$$\le C \sum_{k=1}^{L} \sum_{j=1}^{k-1} (M_a M_b)^j \|H_{k-j}(\tilde{X} W_0)\| \quad \text{(by Lemma J.4)}$$

$$\le C \sum_{k=1}^{L} \sum_{j=1}^{k-1} \mathbf{n}^{-\frac{1}{2}j} \log^j \mathbf{n} \sqrt{\mathbf{n}} \log^{\frac{3}{2}} = O(\log^{\frac{5}{2}} \mathbf{n}) \quad \text{(by Lemma F.4)}$$

(54)

Therefore, we conclude the proof. $\qquad\square$

### F.1 Proofs of Lemma F.3, F.4, and F.5.

*Proof of Lemma F.3.* $k = 1$ is trivial Corollary F.1. For $k \ge 2$, we need to show $\exists C > 0$, w.p. 1-o(1),

$$\|(\tilde{X} G)^{ok} - c_1^k (\tilde{X} \sum_{i<j} \beta_{ij} a_{ij}^\top)^{ok}\| \le C \mathbf{n}^{-\frac{k}{2}} \log^{2k} \mathbf{n}. \tag{55}$$

We use Binomial Expansion.

$$(\tilde{X} G)^{ok} = (\tilde{X} G - c_1 \tilde{X} \sum_{i<j} \beta_{ij} a_{ij}^\top + c_1 \tilde{X} \sum_{i<j} \beta_{ij} a_{ij}^\top)^{ok}$$

$$= \sum_{j=1}^{k} (\binom{k}{j}) (\tilde{X} G - c_1 \tilde{X} \sum_{i<j} \beta_{ij} a_{ij}^\top)^{oj} \odot (c_1 \tilde{X} \sum_{i<j} \beta_{ij} a_{ij}^\top)^{o(k-j)} + c_1^k (\tilde{X} \sum_{i<j} \beta_{ij} a_{ij}^\top)^{ok}$$

(56)

Thus,

$$(\tilde{X} G)^{ok} - c_1^k (\tilde{X} \sum_{i<j} \beta_{ij} a_{ij}^\top)^{ok}$$

$$= \sum_{j=1}^{k} \binom{k}{j} (\tilde{X} G - c_1 \tilde{X} \sum_{i<j} \beta_{ij} a_{ij}^\top)^{oj} \odot c_1^{k-j} (\sum_{i<j} (\tilde{X} \beta_{ij} a_{ij}^\top))^{o(k-j)}$$

(57)

Now we will show

$$\|(\tilde{X} G - c_1 \tilde{X} \sum_{i<j} \beta_{ij} a_{ij}^\top)^{oj} \odot c_1^{k-j} (\sum_{i<j} (\tilde{X} \beta_{ij} a_{ij}^\top))^{o(k-j)}\| = O_{\mathbb{P}}(\log^{k+j} \mathbf{n} \cdot \mathbf{n}^{-\frac{1}{2}k}). \tag{58}$$

$$\|(\tilde{X}G - c_1 \tilde{X} \sum_{i<j} \beta_{ij} a_{ij}^\top)^{oj} \odot c_1^{k-j} (\sum_{i<j} (\tilde{X} \beta_{ij} a_{ij}^\top))^{o(k-j)}\|$$

$$\leq C \|(\tilde{X}G - c_1 \tilde{X} \sum_{i<j} \beta_{ij} a^\top)^{oj} \odot (\tilde{X}\beta a^\top)^{o(k-j)}\| \quad \text{(by triangle inequality)}$$

$$\leq C \|\text{diag}(\tilde{X}\beta)^{ok-j}\|_{op} \|(\tilde{X}G^\top - c_1 \tilde{X} \sum_{i<j} \beta_{ij} a^\top)^{oj}\|_{op} \|\text{diag}(a)^{ok-j}\| \quad \text{(by Fact J.2)} \tag{59}$$

$$\leq C (M_a M_b)^{k-j} \|(\tilde{X}G - c_1 \tilde{X} \sum_{i<j} \beta_{ij} a^\top)^{oj}\|^j \quad \text{(by Lemma J.4)}$$

$$\leq C(n^{-\frac{1}{2}(k-j)} \log^{k-j} \mathbf{n}) \log^{2j} \mathbf{n} \cdot \mathbf{n}^{-\frac{1}{2}j} \quad \text{(by Lemma F.3)}$$

$$= O_{\mathbb{P}}(\mathbf{n}^{-\frac{1}{2}k} \log^{k+j} \mathbf{n})$$

Therefore,

$$\|(\tilde{X}G)^{ok} - c_1^k (\tilde{X} \sum_{i<j} \beta_{ij} a_{ij}^\top)^{ok}\| = O_{\mathbb{P}}(\mathbf{n}^{-\frac{k}{2}} \log^{2k} \mathbf{n}) \tag{60}$$

$\square$

*Proof of Lemma F.4.* Let $A = H_j(\tilde{X}W_0)$, then

$$\mathbb{P}(\|A\|_{op} \geq t) \leq \mathbb{P}\left(\|\frac{1}{\mathbf{n}} AA^\top - \mathbb{E}AA^\top\|_{op} \geq \frac{t^2}{\mathbf{n}} - \|\mathbb{E}AA^\top\|_{op}\right) \quad \text{(by Lemma J.7)}$$

$$\leq \frac{1}{\frac{t^2}{\mathbf{n}} - \|\mathbb{E}AA^\top\|_{op}} \mathbb{E}\left[\|\frac{1}{\mathbf{n}} AA^\top - \mathbb{E}AA^\top\|_{op}\right] \quad \text{(by Markov's inequality)}$$

$$\leq \left[\frac{t^2}{\mathbf{n}} - \mathbb{E}\left[\|AA^\top\|_{op}\right]\right]^{-1} \delta \max\left(\sqrt{\|\mathbb{E}AA^\top\|_{op}}, \delta\right) \quad \text{(by Theorem 5.48 in Vershynin [2010])}$$

$$\leq \left[\frac{t^2}{\mathbf{n}} - \mathbb{E}\left[\|AA^\top\|_{op}\right]\right]^{-1} \delta \max\left(\sqrt{\mathbb{E}\left[\|AA^\top\|_{op}\right]}, \delta\right) \quad \text{(by Jensen's inequality)}.$$

Let $M = E \max_i \|H_j(W_0 \tilde{X}_i)\|^2$ and $\delta = C\sqrt{\frac{M \log \mathbf{n}}{\mathbf{N}}}$. Moreover, we note that $\frac{\|\tilde{X}_i\|^{2j}}{\mathbf{N}}$ is a sub-Weibull random variable, and the bound of Kuchibhotla and Chakrabortty [2022] proposition A.6 can be applied.

Use property of $\frac{\|\tilde{X}_i\|^{2j}}{\mathbf{N}}$, $W_0$ and Hermite polynomials, we have

$$M \leq c_j \mathbb{E} \max_i \|(W_0 \tilde{X}_i)^{oj}\|_2^2 \leq c_j E \max_i \|\tilde{X}_i\|^{2j} \leq c_j \mathbf{N}(\log \mathbf{n})^{\frac{1}{2}}. \tag{61}$$

Therefore, $\delta \leq C \log n$. Let $t^2 = n \cdot Q_n \mathbb{E}\|AA^\top\|_{op}$ s.t. $Q_n$ is positive and increasing. Building on the result derived above, we can continue expanding the expression as follows:

$$\left[\frac{t^2}{\mathbf{n}} - \mathbb{E}\left[\|AA^\top\|_{op}\right]\right]^{-1} \delta \max\left(\sqrt{\mathbb{E}\left[\|AA^\top\|_{op}\right]}, \delta\right)$$

$$\leq \left[\frac{t^2}{\mathbf{n}} - \mathbb{E}\|AA^\top\|_{op}\right]^{-1} C \log \mathbf{n} \max(\sqrt{\mathbb{E}\|AA^\top\|_{op}}, \log \mathbf{n})$$

$$= \left[\mathbb{E}\|AA^\top\|_{op}(Q_{\mathbf{n}} - 1)\right]^{-1} C \log \mathbf{n} \max(\sqrt{\mathbb{E}\|AA^\top\|_{op}}, \log \mathbf{n}) \tag{62}$$

$$\leq C \frac{\log \mathbf{n} \max(\sqrt{\mathbb{E}\|AA^\top\|_{op}}, \log \mathbf{n})}{\mathbb{E}\|AA^\top\|_{op} Q_{\mathbf{n}}}$$

Choosing $Q_{\mathbf{n}} = \log^3 \mathbf{n}$, and using Lemma F.5, we conclude the proof. $\square$

*Proof of Lemma F.5.* For this proof, we apply the proof techniques from Lemma H.1 for Gaussian random variables to Sub-Gaussian random variables and extend the non-centered technique from Theorem H.7.

For non-centered Sub-Gaussian random variable $X$ with mean $\mu$, the following inequalities hold with $t \in \mathbb{R}$:

$$E(e^{(X-\mu)t}) \leq e^{\frac{k^2}{2}t^2} \tag{63}$$

Building on the inequalities above, we first prove the case where $\mu = 0$ with $s, t \in \mathbb{R}$. For centered Sub-Gaussian vector $g$, let $z = g^\top u$, $z' = g^\top v$, $\rho$-correlated. s.t. $\|u\|^2 = \|v\|^2 = 1$, $u^\top v = \rho$, then by Equation (63)

$$\mathbb{E}\exp(sz + tz') \leq \exp(\frac{k^2}{2}\|u\|^2 s^2 + k^2 u^\top v st + \frac{k^2}{2}\|v\|^2 t^2)$$

$$\leq \exp\big(\frac{k^2}{2}(s^2 + 2\rho st + t^2)\big)$$

Dividing by $\exp(\frac{k^2}{2}(s^2 + t^2))$, then

$$\mathbb{E}\big[\exp(sz - \frac{k^2}{2}s^2)\exp(tz' - \frac{k^2}{2}t^2)\big] \leq \exp(\rho st) = \sum_{j=0}^{\infty} \frac{\rho^j}{j!}s^j t^j$$

By applying proof techniques analogous to those in Lemma H.1, one can derive the following:

$$\mathbb{E}H_j(u^\top g)H_k(v^\top g) \leq j!\rho^j \mathbf{1}_{j=k} \tag{64}$$

For the $\mu \neq 0$ case, considering a non-centered Sub-Gaussian Random vector $g$ with mean $\mu$ and a centered Sub-Gaussian Random vector $\xi$ s.t. $g = \xi + \mu$. We apply proof techniques analogous to those in Theorem H.7.

Denote $\nu = \min(j, k)$. Considering $u^\top g, v^\top g$,

$$\mathbb{E}[H_j(u^\top \mu + u^\top \xi)H_k(v^\top \mu + v^\top \xi)]$$

$$= \mathbb{E}[\{\sum_{i=0}^{j} \binom{j}{i}(u^\top \mu)^i H_{j-i}(u^\top \xi)\} \cdot \{\sum_{h=0}^{k} \binom{k}{h}(v^\top \mu)^h H_{k-h}(v^\top \xi)\}]$$

$$= \mathbb{E}[\sum_{q=0}^{\nu} \binom{\nu}{q}^2 (u^\top \mu)^{j-q}(v^\top \mu)^{k-q} H_q(u^\top \xi)H_q(v^\top \xi)] \text{ by Equation (64)} \tag{65}$$

$$\leq \sum_{q=0}^{\nu} \binom{\nu}{q}^2 (u^\top \mu)^{j-q}(v^\top \mu)^{k-q} \cdot \nu!\rho^\nu$$

$$\leq C \min(j, k)!$$

$\square$

# G   Proof of Clustering Criteria Analysis

To reduce redundancy and complexity, we first introduce the following notation.

**Definition G.1.** Given fixed $N, d \in \mathbb{R}$, $e_1$ s.t. the first standard basis of euclidean space, $\mu_1, \mu_2 \in \mathbb{R}^d$, $\hat{\mu}$ is normalized vector to unit length and let

$$S_{d,k}^{(1)} = \mathbb{E}_{w \sim Unif(\mathbb{S}^{d-1})}[(w^\top e_1)^k] \in \mathbb{R}_+$$

$$S_{d,k,k'}^{(2)} = E_w[(w^\top \hat{\mu}_1)^k (w^\top \hat{\mu}_2)^{k'}]$$

$$\rho_{k,k'}^{(1)} = N S_{d,k+k'}^{(1)} \mathbf{1}_{\text{k+k' is even}} \in \mathbb{R}_+$$

$$\rho_{k,k'}^{(2)}(\cos(\mu_1, \mu_2)) = N S_{d,k,k'}^{(2)} \mathbf{1}_{\text{k+k' is even}} \in \mathbb{R}_+ \tag{66}$$

$$\rho_{k,k',r}^{(3)} = \frac{c_1^k S_{d,k'}^{(1)}}{N^{\frac{k}{2}-1}} \binom{k}{r} (r-1)!!(k-1)!! \mathbf{1}_{\text{k,k',r is even}} \in \mathbb{R}_+$$

$$\rho_{k,k',r,r'} = \frac{2c_1^{k+k'} S_{d,k}^{(1)}}{N^{\frac{k+k'}{2}-1}} \binom{k'}{r'} (r'-1)!!(k'-1)!! \mathbf{1}_{\text{k,k',r' is even}} \in \mathbb{R}_+$$

We note that $\rho_{k,k'}^{(1)}, \rho_{k,k',r}^{(3)} > 0$. For $S_{d,k,k'}^{(2)}$, it depends on $\cos(\mu_1, \mu_2)$. As $\cos(\mu_1, \mu_2)$ increases, $S_{d,k,k'}^{(2)}$ grows, while it decreases as $\cos(\mu_1, \mu_2)$ decreases. e.g. when $\mu_1 = \mu_2$, $S_{d,k,k'}^{(2)} = S_{d,k+k'}^{(1)}$, and when $\mu_1 = -\mu_2 = -S_{d,k+k'}^{(1)}$.

We introduce Propositions G.2 and G.3, which serve as extensions of the main results presented in Propositions 4.4 and 4.5. These extended versions incorporate the complete expansion of the dominant feature representation.

In the following, we provide detailed proofs of Propositions G.2 and G.3. Note that if we exclude the contribution from the component $F_{0,L}$ in the computation, the resulting expressions reduce to those in Propositions 4.4 and 4.5.

**Proposition G.2** (Cohesion of $F_L$, extension of Proposition 4.4). *Following condition 4.2, the Cohesion of $F_L$ for $c_i$, $i \in [\![2]\!]$ is given by:*

$$\mathcal{C}(F_L) = \sum_{\substack{k=1 \\ k'=1}}^{L} c_k c_{k'} \left[ \begin{array}{l} \rho_{k,k'}^{(1)} \|\mu\|^{k+k'} \\ +2\sum_{r'=0}^{k'} \rho_{k,k',r'}^{(3)} |\mu^\top \beta|^{k'-r'} \|\beta\|^{r'} \|\mu\|^k \\ +\sum_{r=0}^{k} \sum_{r'=0}^{k'} \rho_{k,k',r,r'} |\mu^\top \beta|^{k+k'-r-r'} \|\beta\|^{r+r'}. \end{array} \right] \tag{67}$$

**Proposition G.3** (Separability of $F_L$, extension of Proposition 4.5). *Following condition 4.2, the Separability of $F_L$ for $c_1, c_2$ is given by:*

$$\mathcal{S}(F_L) = -\sum_{\substack{k=1 \\ k'=1}}^{L} \left[ \begin{array}{l} \rho_{k,k'}^{(2)}(\cos(\mu_1, \mu_2)) \|\mu_1\|^k \|\mu_2\|^{k'} \\ +\sum_{r=0}^{k} \rho_{k,k',r}^{(3)} |\mu_1^\top \beta|^{k-r} \|\beta\|^{r'} \|\mu_2\|^{k'} \\ +\sum_{r'=0}^{k'} \rho_{k,k',r'}^{(3)} |\mu_2^\top \beta|^{k'-r'} \|\beta\|^{r'} \|\mu_1\|^k \\ +\sum_{r=0}^{k} \sum_{r'=0}^{k'} \rho_{k,k',r,r'} (\mu_1^\top \beta)^{k-r} (\mu_2^\top \beta)^{k'-r'} \|\beta\|^{r+r'}. \end{array} \right] \tag{68}$$

We use the following Lemma to prove Proposition G.2 and Proposition G.3. The role of each Lemma is as follows: Lemma G.4 is a foundational result in deriving the expected value associated with the initialized weights of the first layer and the corresponding input data distribution. Lemma G.5 forms the basis of the computation of the expected value of the *spike component* with the data distribution.

**Lemma G.4** (Expectation of the Inner Product Between Uniform Sphere Sample and Given Vector). *Let $C_{d,k} \triangleq \mathbb{E}_\omega[(\omega^\top e_1)^k]$ s.t. $\omega \sim Unif(\mathbb{S}^{d-1})$, then*

$$\mathbb{E}_\omega[(\omega^\top \mu)^k] = \|\mu\|^k S_{d,k}^{(1)} \mathbf{1}_{\text{k is even}} \tag{69}$$

*Proof of G.4.* The uniform distribution on the sphere is origin-symmetric. Therefore, when $k$ is odd, the Expectation is zero. In the other case, also use the isotropic property of a uniform sphere,

$$\mathbb{E}_\omega[(\omega^\top \mu)^k] = \|\mu\|^k E_\omega[(\omega^\top e_1)^k] = \|\mu\|^k S_{d,k}^{(1)} \tag{70}$$

$\square$

**Lemma G.5** (Moment of the Product of Gaussian Random Vectors). *Given vector* $a \in \mathbb{R}^{\mathbf{N}}$ $\beta \in \mathbb{R}^{\mathbf{d}}$ *and Gaussian Random vector* $x \sim \mathcal{N}(\mu, I)$. *Let* $b = x^\top \beta \sim \mathcal{N}(\mu^\top \beta, \|\beta\|^2)$, *then*

$$\mathbb{E}_x (x^\top \beta a^\top)^{\circ k} = \sum_{r=0}^{k} \binom{k}{r} (\mu^\top \beta)^{k-r} \|\beta\|^r (r-1)!! \mathbf{1}_{\text{r is even}} a^{\circ k \top}$$

$$\mathbb{E}_a a^{\circ k} = \frac{(k-1)!! \mathbf{1}_{\text{k is even}}}{\mathbf{N}^{\frac{k}{2}}} \mathbb{1} \tag{71}$$

$$\mathbb{E}_a a^{\circ k \top} a^{\circ k'} = \frac{(k+k'-1)!! \mathbf{1}_{\text{k+k' is even}}}{\mathbf{N}^{\frac{k+k'}{2}-1}}$$

*Proof.* This follows directly from Corollary J.11. □

In the proof below, we utilize the results of Corollary J.11, Corollary J.12, and Lemma G.4.

*Proof of Proposition G.2.* Let the *Cohesion* of the initialized feature be

$$\mathcal{C}(F_{0,L}) = \mathbb{E}_{W_0}[\mathbb{E}_{x \sim c_1} F_{0,L}(x)^\top \mathbb{E}_{x' \sim c_1} F_{0,L}(x')] \tag{72}$$

Let the *Cohesion* of the feature after training be

$$\mathcal{C}(F_L) = \mathbb{E}_{W_0,a}[\mathbb{E}_{x \sim c_1} F_L(x)^\top \mathbb{E}_{x' \sim c_1} F_L(x')] \tag{73}$$

**Calculate** $\mathcal{C}(F_{0,L})$  By Lemma G.4,

$$
\begin{aligned}
\mathcal{C}(F_{0,L}) &= \mathbb{E}_{W_0}[\mathbb{E}_{x \sim c_1}[\sum_{k=1}^{L} c_k H_k(W_0^\top x)]^\top \mathbb{E}_{x' \sim c_1}[\sum_{k'=1}^{L} c_{k'} H_{k'}(W_0^\top x)]] \\
&= \sum_{\substack{k=1 \\ k'=1}}^{L} c_k c_{k'} \mathbb{E}_{W_0}[\sum_{q=1}^{N} (W_0[q]^\top \mu_1)^{k+k'}] \\
&= \mathbf{N} \sum_{\substack{k=1 \\ k'=1}}^{L} c_k c_{k'} (\|\mu_1\|^{k+k'} S_{d,k+k'}^{(1)}) \mathbf{1}_{(k+k') \text{even}} \\
&= \sum_{\substack{k=1 \\ k'=1}}^{L} c_k c_{k'} \rho_{k,k'}^{(1)} \|\mu\|^{k+k'}
\end{aligned} \tag{74}
$$

**Calculate** $\mathcal{C}(F_L)$

$$\mathcal{C}(F_L) = \mathbb{E}_{W_0,a}[\mathbb{E}_{x\sim c_1}[\sum_{k=1}^{L}(c_k H_k(W_0^\top x) + c_k c_1^k(x^\top \beta a)^{\circ k}]^\top \mathbb{E}_{x'\sim c_1}[\sum_{k'=1}^{L}(c_{k'} H_{k'}(W_0^\top x) + c_1^k(x^\top \beta a)^{\circ k}]]$$

$$= \mathbb{E}_{W_0,a}[\sum_{\substack{k=1\\k'=1}}^{L} c_k c_{k'}[\mathbb{E}_x H_k(W_0^\top x)^\top \mathbb{E}_{x'} H_{k'}(W_0^\top x')$$

$$+ 2\mathbb{E}_x H_k(W_0^\top x)^\top \mathbb{E}_{x'} c_1^{k'}(x^\top \beta a)^{\circ k'} + c_1^{k+k'} \mathbb{E}_x(x^\top \beta a)^{\circ k\top} \mathbb{E}_{x'}(x'^\top \beta a)^{\circ k'}]]$$

$$= \mathcal{C}(F_{0,L}) + 2\sum_{\substack{k=1\\k'=1}}^{L} c_k c_{k'} c_1^{k'} \mathbb{E}_{W_0}\mathbb{E}_x H_k(W_0^\top x)^\top \mathbb{E}_a \mathbb{E}_{x'}(x'^\top \beta a)^{\circ k'}$$

$$+ \sum_{\substack{k=1\\k'=1}}^{L} c_k c_{k'} c_1^{k+k'} \mathbb{E}_a[\mathbb{E}_x(x^\top \beta a)^{\circ k\top} \mathbb{E}_{x'}(x'^\top \beta a)^{\circ k}]$$

$$= \mathcal{C}(F_{0,L}) + 2\mathbf{N}\sum_{k,k'=1}^{L} c_k c_{k'} c_1^{k'}(\|\mu_1\|^k S_{d,k}^{(1)})(\frac{1}{\mathbf{N}^{\frac{k'}{2}}}\sum_{r'=0}^{k'}\binom{k'}{r'}(\mu_1^\top\beta)^{k'-r'}\|\beta\|^{r'}(r'-1)!!(k'-1)!!\mathbf{1}_{\text{k,k',r' is even}}$$

$$+ \sum_{\substack{k=1\\k'=1}}^{L} \frac{c_k c_{k'} c_1^{k+k'}}{\mathbf{N}^{\frac{k+k'}{2}-1}}\sum_{r=0}^{k}\sum_{r'=0}^{k'}\binom{k}{r}\binom{k'}{r'}(\mu_1^\top\beta)^{k+k'-r-r'}\|\beta\|^{r+r'}(r-1)!!(r'-1)!!\mathbf{1}_{\text{k+k',r,r' is even}}$$

$\square$

*Proof of Proposition G.3.* Let the *Separability* of the initialized feature be

$$\mathcal{S}(F_{0,L}) = -\mathbb{E}_{W_0}[\mathbb{E}_{x\sim c_1} F_{0,L}(x)^\top \mathbb{E}_{x'\sim c_2} F_{0,L}(x')] \tag{75}$$

Let the *Separability* of the feature after training be

$$\mathcal{S}(F_L) = -\mathbb{E}_{W_0,a}[\mathbb{E}_{x\sim c_1} F_L(x)^\top \mathbb{E}_{x'\sim c_2} F_L(x')] \tag{76}$$

**Calculate** $\mathcal{S}(F_{0,L})$   By Lemma G.4,

$$\mathcal{S}(F_{0,L}) = -\sum_{\substack{k=1\\k'=1}}^{L} c_k c_{k'} \mathbb{E}_{W_0}[\sum_{q=1}^{N}(W_0[q]^\top \mu_1)^k(W_0[q]^\top \mu_2)^{k'}]$$

$$= -\mathbf{N}\sum_{\substack{k=1\\k'=1}}^{L} c_k c_{k'} \mathbb{E}_{w\sim Unif(\mathbb{S}^{d-1})}[(w^\top \mu_1)^k(w^\top \mu_2)^{k'}]$$

$$= -\mathbf{N}\sum_{\substack{k=1\\k'=1}}^{L} c_k c_{k'} \|\mu_1\|^k \|\mu_2\|^{k'} E_w[(w^\top \hat\mu_1)^k(w^\top \hat\mu_2)^{k'}] \tag{77}$$

$$= -\mathbf{N}\sum_{\substack{k=1\\k'=1}}^{L} c_k c_{k'} \|\mu_1\|^k \|\mu_2\|^{k'} S_{d,k,k'}^{(2)} \mathbf{1}_{\text{k+k' is even}}$$

$$= -\sum_{\substack{k=1\\k'=1}}^{L} c_k c_{k'} \|\mu_1\|^k \|\mu_2\|^{k'} \rho_{k,k'}^{(1)}$$

**Calculate** $\mathcal{S}(F_L)$

$\mathcal{S}(F_L)$

$$
= -\sum_{\substack{k=1\\k'=1}}^{L} c_k c_{k'} \mathbb{E}_{W_0,a}
\begin{bmatrix}
\mathbb{E}_{x\sim c_1} H_k(W_0^\top x)^\top \mathbb{E}_{x'\sim c_2} H_{k'}(W_0^\top x') \\
+ \mathbb{E}_{x\sim c_1} H_k(W_0^\top x)^\top \mathbb{E}_{x'\sim c_2} c_1^{k'}(x'^\top \beta a)^{\circ k'} \\
+ \mathbb{E}_{x\sim c_1} c_1^{k}(x^\top \beta a)^{ok\,\top} \mathbb{E}_{x'\sim c_2} H_{k'}(W_0^\top x) \\
+ c_1^{k+k'} \mathbb{E}_{x\sim c_1}(x^\top \beta a)^{ok\,\top} \mathbb{E}_{x'\sim c_2}(x'^\top \beta a)^{\circ k'}
\end{bmatrix}
$$

$$
= \mathcal{S}(F_{0,L}) - \sum_{\substack{k=1\\k'=1}}^{L} c_k c_{k'}
\begin{bmatrix}
c_1^{k'}(\|\mu_1\|^k S_{d,k}^{(1)}) \dfrac{1}{\mathbf{N}^{\frac{k'}{2}-1}} \sum_{r'=0}^{k'} \binom{k'}{r'}(\mu_2^\top \beta)^{k'-r'}\|\beta\|^{r'}(r'-1)!!(k'-1)!!\mathbf{1}_{k,k',r'\text{ is even}} \\[3mm]
+ c_1^{k}(\|\mu_2\|^{k'} S_{d,k'}^{(1)}) \dfrac{1}{\mathbf{N}^{\frac{k}{2}-1}} \sum_{r=0}^{k} \binom{k}{r}(\mu_1^\top \beta)^{k-r}\|\beta\|^{r}(r-1)!!(k-1)!!\mathbf{1}_{k,r,k'\text{ is even}} \\[3mm]
+ c_1^{k+k'} \sum_{r=0}^{k}\sum_{r'=0}^{k} \binom{k}{r}\binom{k'}{r'}(\mu_1^\top \beta)^{k-r}(\mu_2^\top \beta)^{k'-r'}\|\beta\|^{r+r'}(r-1)!!(r'-1)!! \\[3mm]
\hspace{4cm} \dfrac{1}{\mathbf{N}^{\frac{k+k'}{2}-1}}(k+k'-1)!!\mathbf{1}_{k+k',r,r'\text{ is even}}
\end{bmatrix}
$$

$\square$

*Note* G.6 (Discussion on the $\|\beta\|$ term). We note that when $\|\beta\|$ decreases, it reduces magnitude of term inside the brackets of Eqs. (6) and (7). This aligns with the intuitive notion that noisier training data leads to less transferable features. To illustrate, suppose the training data consists of two classes drawn from $c_1 \sim \mathcal{N}(\mu, I)$ and $c_2 \sim \mathcal{N}(R\mu, I)$, where $R$ is a rotation matrix. Then the spike direction converges (under large $n$) to $\beta \to \frac{1}{2}(\mu - R\mu)$ by law of large numbers

$$
\beta = \frac{1}{n}X^\top y = \frac{1}{n}X_i y_i = \frac{1}{2}\left(\frac{n}{2}\sum_{c_1} X_i - \frac{n}{2}\sum_{c_2} X_j\right) \to \frac{1}{2}(\mu - R\mu).
$$

In the extreme case where $R = I$, the classes are indistinguishable and $\beta = 0$, eliminating both Cohesion and Separability. When $R = -I$, $\beta = \mu$, which yields maximal separation.

# H  Additional Results of Expectation of Hermite Polynomials

The non-standard Gaussian expectation of the product of two Hermite polynomials is computed as follows. It is a generalization of the results of standard Gaussian distributions in O'Donnell [2021], Moniri et al. [2024] into a generalized multivariate Gaussian. These provide a useful analysis tool for Hermite polynomials and may offer a foundation for broader applications in future works involving nonlinear activations decomposable into Hermite polynomials under the assumption of a multivariate Gaussian distribution. We start with previously known facts and derive our generalized results.

## H.1  Expectation of a product of two Hermite polynomials

Here is the result of the expectation of the product of two Hermite polynomials, obtained by utilizing the orthogonality of Hermite polynomials with bivariate centered unit-variance correlated random variables.

**Lemma H.1** (Gaussian Expectation of the Product of Two Hermite Polynomials from Lemma C.1 Moniri et al. [2024]). *See also derivation in Chapter 11.2 O'Donnell [2021].*

*Let $(Z_1, Z_2)$ be jointly Gaussian with $\mathbb{E}[Z_1] = \mathbb{E}[Z_2] = 0$, $\mathbb{E}[Z_1^2] = \mathbb{E}[Z_2^2] = 1$, and $\mathbb{E}[Z_1 Z_2] = \rho$. Then for any $k_1, k_2 \in \{0, 1, \cdots, \}$*

$$
\mathbb{E}[H_{k_1}(Z_1)H_{k_2}(Z_2)] = k_1! \rho^{k_1} \mathbf{1}_{k_1=k_2} \tag{78}
$$

*In the other form, for $d \in \mathbb{N}$, $Z \sim \mathcal{N}(0, I_d)$, $a, b \in \mathbb{S}^{d-1}$,*

$$
\mathbb{E}[H_{k_1}(Z^\top a)H_{k_2}(Z^\top b)] = k_1!(a^\top b)^{k_1}\mathbf{1}_{k_1=k_2} \tag{79}
$$

We extend Lemma H.1 to vector form for application in multiple dimensions.

*Fact* H.2 (Vector Form of Lemma H.1). Let $W \in \mathbb{R}^{d \times N}$ s.t. $\forall i \; W[i] \in \mathbb{S}^{d-1}$. For $Z \sim \mathcal{N}(0, I)$,

$$\mathbb{E}_{Z \sim \mathcal{N}(0,1)}[H_j(W^\top Z) H_k(W^\top Z)^\top] = k!(W^\top W)^{\circ j} \mathbf{1}_{j=k} \tag{80}$$

$$\mathbb{E}_{Z \sim \mathcal{N}(0,1)}[H_j(W^\top Z)^\top H_k(W^\top Z)] = k! \sum \|W[i]\|^{2j} \mathbf{1}_{j=k} = k! N \mathbf{1}_{j=k} \tag{81}$$

*Proof.* We apply $H_j$ element-wise. We can acquire the above result by Lemma H.1. $\square$

The following remark presents a modified condition of Lemma H.1 for the case where $a, b \notin \mathbb{S}^{d-1}$ in Lemma H.1. In this case, the variances of $Z^\top a$ and $Z^\top b$ are not equal to 1, and the covariance may exceed the bounds $[-1, 1]$. Under this condition, we will compute the expectation of the product of two Hermite polynomials as in Lemma H.1.

*Remark* H.3 (the modified condition of Lemma H.1). For $d \in \mathbb{N}$, $u, v \in \mathbb{R}^d$, $Z \sim \mathcal{N}(0, I_d)$,

$Z_1 = \langle u, Z \rangle \sim \mathcal{N}(0, \|u\|_2^2)$, $Z_2 = \langle v, Z \rangle \sim \mathcal{N}(0, \|v\|_2^2)$.

Then, $Z_1, Z_2$ is $\rho =^\triangle \langle \frac{u}{\|u\|}, \frac{v}{\|v\|} \rangle$ - correlated

$$\text{corr}(Z_1, Z_2) = \frac{\mathbb{E}_Z \langle u, Z \rangle \langle v, Z \rangle}{\|u\| \, \|v\|} = \frac{\langle u, v \rangle}{\|u\| \, \|v\|} \tag{82}$$

Additionally,

$$\begin{pmatrix} Z_1 \\ Z_2 \end{pmatrix} \sim n\left( \begin{pmatrix} 0 \\ 0 \end{pmatrix}, \begin{pmatrix} \|u\|^2 & \langle u, v \rangle \\ \langle v, u \rangle & \|v\|^2 \end{pmatrix} \right) \tag{83}$$

Now, we generalize the unit variance distribution assumptions so that Lemma H.1 holds for arbitrary vectors as in Remark H.3. This could allow the networks' weights to become analyzable when they go beyond the assumption of lying on the unit spheres.

**Theorem H.4** (Generalization of Lemma H.1 for non unit variance Gaussian distribution as Remark H.3). *For $d \in \mathbb{N}$, $u, v \in \mathbb{R}^d$, $g \sim \mathcal{N}(0, I_d)$, $\langle u, g \rangle \sim \mathcal{N}(0, \|u\|_2^2)$, $\langle v, g \rangle \sim \mathcal{N}(0, \|v\|_2^2)$.*

$$\begin{aligned}
&\mathbb{E}_g[H_j(u^\top g) H_k(v^\top g)] \\
&= \frac{j! \langle u, v \rangle^j}{\|u\|^2 \|v\|^2} \mathbf{1}_{j=k} - \frac{(\|u\|^2 - 1)(\|v\|^2 - 1)}{\|u\|^2 \|v\|^2} \mathbb{E}_g[(v^\top g)^k (u^\top g)^j] \\
&\quad + \frac{(\|v\|^2 - 1)}{\|v\|^2} \mathbb{E}_g[H_j(u^\top g)(v^\top g)^k] + \frac{(\|u\|^2 - 1)}{\|u\|^2} \mathbb{E}_g[H_k(v^\top g)(u^\top g)^j]
\end{aligned} \tag{84}$$

*Remark* H.5 (Unit-variance case of Theorem H.4). The same results can be derived as in Lemma H.1 when the variance is 1 in Thm. H.4.

*Proof of Theorem H.4.* (Generalize Chapter 11.2 O'Donnell [2021]'s derivation to non-unit variance)

$\mathbb{E}_{z \sim \mathcal{N}(0, \sigma^2)}[e^{tz}]$ **study**

First, we study about $\mathbb{E}_{g \sim \mathcal{N}(0, \sigma^2)}[e^{tg}]$ in order to analysis non-unit variance case.

$$\begin{aligned}
\mathbb{E}_{g \sim \mathcal{N}(0, \sigma^2)}[e^{tg}] &= \frac{1}{\sqrt{2\pi}\sigma} \int e^{tg} e^{-\frac{g^2}{2\sigma^2}} \, dg \\
&= \frac{1}{\sqrt{2\pi}\sigma} e^{\frac{1}{2}t^2} \int \exp\left(-\frac{(g - \sigma^2 t)^2}{2\sigma^2}\right) \quad \text{complete square} \\
&= e^{\frac{1}{2}t^2}
\end{aligned} \tag{85}$$

$\mathbb{E}_{Z, Z'}[\exp(sZ + tZ')]$ **study**

Studying $\mathbb{E}_{Z, Z'}[\exp(sZ + tZ')]$, we can derive what we need to show.

$$\mathbb{E}_{Z,Z'}\left[\exp(sZ+tZ')\right]=\mathbb{E}_{g\sim n(0,I)}\left[\exp\left(s\langle u,g\rangle\right)+\exp\left(t\langle v,g\rangle\right)\right]$$

$$=\prod_i\mathbb{E}_{g\sim n(0,1)}\left[\exp\left((su_i+tv_i)g_i\right)\right]\quad\text{Use Equation (85)}$$

$$=\prod_i\exp\left(\frac{1}{2}(su_i+tv_i)^2\right)=\prod_i\exp\left(\frac{1}{2}s^2\|u\|^2+\langle u,v\rangle st+\frac{1}{2}t^2\|v\|^2\right)\tag{86}$$

Therefore,

$$\exp\left(\langle u,v\rangle st\right)=\mathbb{E}_g\left[\exp\left(su^\top g-\frac{1}{2}s^2\|u\|^2\right)\exp\left(tv^\top g-\frac{1}{2}t^2\|v\|^2\right)\right].\tag{87}$$

*Fact* H.6 (Facts for the Proof of Lemma H.4). One can verify the propositions below with simple calculations.
Let $P_j(z)+z^j=H_j(z)$, $C_u=\|u\|^2-1$, $a>0$.
Let $f(s)=\exp(sz-\frac{1}{2}s^2)$, $\bar{f}(s)=\exp(sz-\frac{1}{2}as^2)$, then

    A. By Taylor expansion, $\exp(\langle u,v\rangle st)=\sum_{j=0}^\infty\frac{1}{j!}\langle u,v\rangle^j s^j t^j$.

    B. By Taylor expansion, $\bar{f}(s)=\sum_{j=0}^\infty\frac{1}{j!}\bar{f}^{(n)}(0)s^j$

    C. $\bar{f}^{(n)}(0)=H_n(z)+C_uP_n(z)$

By using the fact that $\exp\left(\langle u,v\rangle st\right)=\mathbb{E}_g\left[\exp(su^\top g-\frac{1}{2}s^2\|u\|^2)\exp(tv^\top g-\frac{1}{2}t^2\|v\|^2)\right]$, we can eliminate the different orders of $s\,t$ by a Taylor expansion and equating all monomials of the resulting polynomials.

$$j!\langle u,v\rangle^j\mathbf{1}_{j=k}=\mathbb{E}_g\left[(H_j(u^\top g)+P_j(u^\top g)C_u)(H_j(v^\top g)+P_j(v^\top g)C_v)\right]$$

$$=\mathbb{E}_g\left[(H_j(u^\top g)+(H_j(u^\top g)-(u^\top g)^j)C_u)(H_j(v^\top g)+(H_j(v^\top g)-(v^\top g)^j)C_v)\right]$$

$$=\|u\|^2\|v\|^2\mathbb{E}_g\left[H_j(u^\top g)H_j(v^\top g)\right]+(\|u\|^2-1)(\|v\|^2-1)\mathbb{E}_g\left[(v^\top g)^j(u^\top g)^j\right]$$

$$-\|u\|^2(\|v\|^2-1)\mathbb{E}_g\left[H_j(u^\top g)(v^\top g)^j\right]-\|v\|^2(\|u\|^2-1)\mathbb{E}_g\left[H_j(v^\top g)(u^\top g)^j\right]\tag{88}$$

Therefore,

$$\mathbb{E}_g\left[H_j(u^\top g)H_j(v^\top g)\right]$$

$$=\frac{j!\langle u,v\rangle^j}{\|u\|^2\|v\|^2}\mathbf{1}_{j=k}-\frac{(\|u\|^2-1)(\|v\|^2-1)}{\|u\|^2\|v\|^2}\mathbb{E}_g\left[(v^\top g)^j(u^\top g)^j\right]$$

$$+\frac{(\|v\|^2-1)}{\|v\|^2}\mathbb{E}_g\left[H_j(u^\top g)(v^\top g)^j\right]+\frac{(\|u\|^2-1)}{\|u\|^2}\mathbb{E}_g\left[H_j(v^\top g)(u^\top g)^j\right]\tag{89}$$

Note that the result of Lemma J.9 can be applied for the concrete calculation of results in Theorem H.4 and conclude the proof. $\qquad\square$

## H.2 Expectation of a product of two Hermite polynomials—Generalization toward non-centered Gaussian

We will change Theorem H.4 and Lemma J.9 to adopt a generalized Gaussian assumption with non-centered mean.

**Theorem H.7** (Generalization of Thm. H.4 for any Gaussian distribution). *For $d \in \mathbb{N}$, $u, v \in \mathbb{R}^d$, $\xi \sim \mathcal{N}(0,1)$, $g \sim \mathcal{N}(\mu, \Sigma)$, $Z_1 = \langle u, g \rangle \sim \mathcal{N}(\mu^\top u, u^\top \Sigma u)$, $Z_2 = \langle v, g \rangle \sim \mathcal{N}(\mu^\top v, v^\top \Sigma v)$.*

$$
\mathbb{E}_g[H_j(Z_1) H_k(Z_2)]
$$

$$
= \sum_{\alpha=0}^{j} \sum_{\beta=0}^{k} \binom{j}{\alpha} \binom{k}{\beta} (u^\top \mu)^\alpha (v^\top \mu)^\beta
$$

$$
\times \left[ \frac{(j-\alpha)!(u^\top \Sigma v)^{j-\alpha}}{u^\top \Sigma u v^\top \Sigma v} \mathbf{1}_{j-\alpha = k-\beta} - \frac{(u^\top \Sigma u - 1)(v^\top \Sigma v - 1)}{u^\top \Sigma u v^\top \Sigma v} \mathbb{E}_g[(\sqrt{u^\top \Sigma u}\xi)^{j-\alpha}(\sqrt{v^\top \Sigma v}\xi)^{k-\beta}] \right.
$$

$$
\left. + \frac{(v^\top \Sigma v - 1)}{v^\top \Sigma v} \mathbb{E}_g[H_{j-\alpha}(\sqrt{u^\top \Sigma u}\xi)(\sqrt{v^\top \Sigma v}\xi)^{k-\beta}] + \frac{(u^\top \Sigma u - 1)}{u^\top \Sigma u} \mathbb{E}_g[(\sqrt{u^\top \Sigma u}\xi)^{j-\alpha} H_{k-\beta}(\sqrt{v^\top \Sigma v}\xi)] \right] \tag{90}
$$

*Proof of Theorem H.7.* By reparametrization i.e. $Z_1 = \sqrt{u^\top \Sigma u}\xi + u^\top \mu$, $Z_2 = \sqrt{v^\top \Sigma v}\xi + v^\top \mu$, and Lemma J.1,

$$
H_j(\sqrt{u^\top \Sigma u}\xi + u^\top \mu) = \sum_{\alpha=0}^{j} \binom{j}{\alpha} (u^\top \mu)^\alpha H_{j-\alpha}(\sqrt{\mu^\top \Sigma u}\xi). \tag{91}
$$

$$
\mathbb{E}_g[H_j(u^\top g) H_k(v^\top g)] = \mathbb{E}_\xi[H_j(\sqrt{u^\top \Sigma u}\xi + u^\top \mu) H_k(\sqrt{v^\top \Sigma v}\xi + v^\top \mu)]
$$

$$
= \mathbb{E}_\xi \left[ \sum_{\alpha=0}^{j} \binom{j}{\alpha} (u^\top \mu)^\alpha H_{j-\alpha}(\sqrt{\mu^\top \Sigma u}\xi) \right] \left[ \sum_{\beta=0}^{k} \binom{k}{\beta} (v^\top \mu)^\beta H_{k-\beta}(\sqrt{\mu^\top \Sigma v}\xi) \right] \tag{92}
$$

$$
= \sum_{\alpha=0}^{j} \sum_{\beta=0}^{k} \binom{j}{\alpha} \binom{k}{\beta} (u^\top \mu)^\alpha (v^\top \mu)^\beta \mathbb{E}_\xi[H_{j-\alpha}(\sqrt{\mu^\top \Sigma u}\xi) H_{k-\beta}(\sqrt{\mu^\top \Sigma v}\xi)]
$$

Use the same proof technique as Theorem H.4, with $\begin{pmatrix} \sqrt{u^\top \Sigma u}\xi \\ \sqrt{v^\top \Sigma v}\xi \end{pmatrix} \sim \mathcal{N}\left( \begin{pmatrix} 0 \\ 0 \end{pmatrix}, \begin{pmatrix} u^\top \Sigma u & u^\top \Sigma v \\ v^\top \Sigma u & v^\top \Sigma v \end{pmatrix} \right)$

$$
\mathbb{E}_\xi[H_{j-\alpha}(\sqrt{u^\top \Sigma u}\xi) H_{k-\beta}(\sqrt{v^\top \Sigma v}\xi)]
$$

$$
= \frac{(j-\alpha)!(u^\top \Sigma v)^{j-\alpha}}{u^\top \Sigma u v^\top \Sigma v} \mathbf{1}_{j-\alpha = k-\beta} - \frac{(u^\top \Sigma u - 1)(v^\top \Sigma v - 1)}{u^\top \Sigma u v^\top \Sigma v} \mathbb{E}_g[(\sqrt{u^\top \Sigma u}\xi)^{j-\alpha}(\sqrt{v^\top \Sigma v}\xi)^{k-\beta}]
$$

$$
+ \frac{(v^\top \Sigma v - 1)}{v^\top \Sigma v} \mathbb{E}_g[H_{j-\alpha}(\sqrt{u^\top \Sigma u}\xi)(\sqrt{v^\top \Sigma v}\xi)^{k-\beta}] + \frac{(u^\top \Sigma u - 1)}{u^\top \Sigma u} \mathbb{E}_g[(\sqrt{u^\top \Sigma u}\xi)^{j-\alpha} H_{k-\beta}(\sqrt{v^\top \Sigma v}\xi)] \tag{93}
$$

In summary,

$$
\mathbb{E}_g[H_j(u^\top g) H_k(v^\top g)]
$$

$$
= \sum_{\alpha=0}^{j} \sum_{\beta=0}^{k} \binom{j}{\alpha} \binom{k}{\beta} (u^\top \mu)^\alpha (v^\top \mu)^\beta
$$

$$
\times \left[ \frac{(j-\alpha)!(u^\top \Sigma v)^{j-\alpha}}{u^\top \Sigma u v^\top \Sigma v} \mathbf{1}_{j-\alpha = k-\beta} - \frac{(u^\top \Sigma u - 1)(v^\top \Sigma v - 1)}{u^\top \Sigma u v^\top \Sigma v} \mathbb{E}_\xi[(\sqrt{u^\top \Sigma u}\xi)^{j-\alpha}(\sqrt{v^\top \Sigma v}\xi)^{k-\beta}] \right.
$$

$$
\left. + \frac{(v^\top \Sigma v - 1)}{v^\top \Sigma v} \mathbb{E}_\xi[H_{j-\alpha}(\sqrt{u^\top \Sigma u}\xi)(\sqrt{v^\top \Sigma v}\xi)^{k-\beta}] + \frac{(u^\top \Sigma u - 1)}{u^\top \Sigma u} \mathbb{E}_\xi[(\sqrt{u^\top \Sigma u}\xi)^{j-\alpha} H_{k-\beta}(\sqrt{v^\top \Sigma v}\xi)] \right] \tag{94}
$$

$\square$

# I Additional Lemmas of Sub-Gaussian Distribution

For a more detailed explanation and well-known results of the Sub-Gaussian we used, please refer to Vershynin [2010, 2018]. We show below that the truncated Gaussian distribution utilized in our synthetic data experiments is a Sub-Gaussian distribution.

The following lemmas serve distinct purposes in the overall paper: Lemma I.1 is employed to verify the sub-Gaussianity of Data 2 in Expr. I, II and III Lemma I.2 establishes a key property that is instrumental throughout the entirety of our proof. Lemmas I.3 and I.4 provide the necessary extensions of the results by Ba et al. [2022] and Moniri et al. [2024] to settings involving non-centered sub-Gaussian distributions.

**Lemma I.1** (Sub-Gaussian Property of Truncated Gaussian). *The truncated Gaussian distribution has support on $(a, b)$ s.t. $a, b \in (-\infty, \infty)$ is Sub-Gaussian.*

*Proof.* Denote $\mathcal{N}_{(a,b)}(0, \sigma^2)$ is Truncated Gaussian distribution which have support on $(a, b)$ s.t. $a, b \in (-\infty, \infty)$. We utilize sufficient conditions for Sub-Gaussian distributions based on their tail structure. support $(\mathcal{N}_{(a,b)}(0, \sigma^2)) \subset \mathbb{R}^d$. Therefore, $\mathbb{P}(|X| \geq t)$ s.t. $X \sim \mathcal{N}_{(a,b)}(0, \sigma^2)$ has the same tail behavior as Gaussian, and Gaussian is Sub-Gaussian. $\square$

## I.1 Generalization of centered Sub-Gaussian results toward non-centered

We verify below that the results on centered Sub-Gaussian distributions from Vershynin [2018] can be extended to non-centered Sub-Gaussian distributions.

**Lemma I.2** (Sub-Gaussian Property of Sum of non-centered Sub-Gaussian). *The sum of non-centered Sub-Gaussian random variables is Sub-Gaussian.*

*Proof.* If the Orlicz 2 norm is bounded $\|X\|_{\psi_2} < \infty$, then X is Sub-Gaussian. Also, $\|\mathbb{E}X\|_{\psi_2} \leq C\|X\|_{\psi_2}$, and the sum of the centered Sub-Gaussian random variable is Sub-Gaussian. We show $\|\sum X_i\|_{\psi_2} < \infty$, s.t. X is non-centered Sub-Gaussian.

$$\|\sum X_i\|_{\psi_2} \leq \|\sum (X_i - \mathbb{E}X_i)\|_{\psi_2} + \|\sum \mathbb{E}X_i\|_{\psi_2}$$
$$\leq \|\sum (X_i - \mathbb{E}X_i)\|_{\psi_2} + \sum \|\mathbb{E}X_i\|_{\psi_2} \tag{95}$$
$$\leq \|\sum (X_i - \mathbb{E}X_i)\|_{\psi_2} + C \sum \|X_i\|_{\psi_2} < \infty$$

$\square$

**Lemma I.3** (Operator norm bound for non-centered Sub-Gaussian matrix, generalization of 4.4.5 in Vershynin [2018]). *let $A \in \mathbb{R}^{m \times n}$, $A[i][j]$ is independent, non-centered Sub-Gaussian. $\forall t > 0$,*

$$\|A\| \leq CK(\sqrt{m} + \sqrt{n} + t) \text{ w.p. } 1 - \exp(-t^2)$$
$$\text{Alternatively, } \|A\| \leq CK(\sqrt{m + n} + t) \text{ w.p. } 1 - \exp(-t^2) \tag{96}$$

$K = \max_{i,j} \|A[i][j]\|_{\psi_2}$

**Lemma I.4** (Expectation of operator norm for non-centered Sub-Gaussian matrix generalization of 4.4.6 in Vershynin [2018]).

$$\mathbb{E}\|A\| \leq CK(\sqrt{m} + \sqrt{n})$$
$$\text{Alternatively, } \mathbb{E}\|A\| \leq CK(\sqrt{m + n}), \quad \text{and, } \mathbb{E}\|A\|^2 \leq C(m + n) \tag{97}$$

*Proof of Lemma I.3 and Lemma I.4.* Based on the result of Lemma I.2, one can follow the same proof process of Vershynin [2018] $\square$

# J Supplementary Lemmas

**Lemma J.1** (Taylor expansion of Hermite polynomials from Lemma C.2 Moniri et al. [2024]). *For any $k_1, k_2 \in \{0, 1, \cdots, \}$ and $x, y \in \mathbb{R}$,*

$$H_k(x + y) = \sum_{j=0}^{k} \binom{k}{j} x^j H_{k-j}(y). \tag{98}$$

## J.1 Lemmas for Norm Bounds

The proofs of the lemmas that require justification are provided in Section J.2.

*Fact* J.2 (Norm Bound for Hadamard Product from Ba et al. [2022]). For $m \in \mathbb{R}^{l_1}, n \in \mathbb{R}^{l_2}, M \in \mathbb{R}^{l_1 \times l_2}$,

$$mn^\top \odot M = \text{diag}(m)M\text{diag}(n)$$

$$\|mn^\top \odot M\| \le \|\text{diag}(m)\| \, \|M\| \, \|\text{diag}(n)\| = \|m\|_\infty \|M\| \|n\|_\infty \tag{99}$$

Using the sufficient conditions for Sub-Gaussianity via the Orlicz norm and bound assumptions on activation, the following fact holds:

*Fact* J.3 (Sub-Gaussian Property of Random Variables after Activation). Let a Sub-Gaussian random variable $v$ s.t. $\|v\|_{\psi_2} \le k$, and bounded function $\sigma$, then $\sigma(v)$ is Sub-Gaussian, i.e.

$$\|\sigma(v)\|_{\psi_2} \le \|\lambda\|_{\psi_2} < \infty. \tag{100}$$

**Lemma J.4** (Norm Bounds of Data, Gradient and Parameters). *The following facts will be used in subsequent proofs. Remark spike direction $\beta_{ij} \triangleq \frac{1}{n}X_{ij}^\top y$ in Definition 3.2.*

A. $\|X_{ij}\| = O_\mathbb{P}(\sqrt{\mathbf{n}})$, $\|y\| = O_\mathbb{P}(\sqrt{\mathbf{n}})$, $\|\beta_{ij}\| = O_\mathbb{P}(1)$

B. $\|X_{ij}\beta_{ij}a_{ij}\| = \|X\beta_{ij}\|_2\|a_{ij}\|_2 = O_\mathbb{P}(\sqrt{\mathbf{n}})$

C. $\|W_0\| = O_\mathbb{P}(1)$, $\|W\| = \|W_0 + G\| \le \|W_0\| + \|G\| = O_\mathbb{P}(1)$

D. $\|X_{ij}G\| = O_\mathbb{P}(\sqrt{\mathbf{n}})$

E. $M_a \triangleq \|a_{ij}\|_\infty = \max_{1 \le k \le \mathbf{N}}|a_{ij}[k]| \le \frac{C\log^{1/2}\mathbf{n}}{\sqrt{\mathbf{n}}}$ *w.p* $1 - 2ne^{-c\log\mathbf{n}}$

F. $M_b \triangleq \|X\beta\|_\infty = \max_{1 \le k \le \mathbf{n}}| < X[k], \beta > | \le C\log^{1/2}\mathbf{n}$, *w.p.* $1 - 2\mathbf{n}e^{-c\log\mathbf{n}}$

G. $M_{W_0} \triangleq \sup_{k \ge 1}\|(W_0W_0^\top)^{\circ k}\| \le C$ *w.p.* $1 - o(1)$

H. $\|A^{\circ k}\| \le \|A\|^k$

*Fact* J.5 (Bounds of Norms of Vectors and Matrices). For any vector $u, v$ and any matrix $A, B$

A. $\|uv^\top\|_{op} = \|u\|_2\|v\|_2$

B. $\|u\|_\infty \le \|u\|_2 \le \sqrt{\mathbf{n}}\|u\|_\infty$

C. $\|u^{\circ k}\| \le \|u\|^k$

D. $\|u^{\circ k}\|_2 \le \sqrt{\mathbf{n}}\|u^{\circ k}\|_\infty \le \sqrt{\mathbf{n}}\max_i(|u_i^k|) = \sqrt{\mathbf{n}}(\max_i|u_i|)^k = \sqrt{\mathbf{n}}\|u\|_\infty^k$

E. Schur Product Theorem

$$\|A \circ B\|_{op} = \sup_{\|x\|=1} tr(A^\top\text{diag}(x)B\text{diag}(x)) \le \|A\|_{op} \cdot \|B\|_{op} \tag{101}$$

**Lemma J.6** (Probabilistic Bound on the Inf Norm of Sub-Gaussian Random Vector). *For Sub-Gaussian R.V. $a$,*

$$\mathbb{P}(\|a\|_\infty \le t/\sqrt{\mathbf{N}}) \ge 1 - 2\mathbf{N}e^{-ct^2} \tag{102}$$

**Lemma J.7** (Operator Norm Bound of Random Matrices). *Given random matrix A, Following statement holds,*

$$\mathbb{P}(\|A\|_{op} \ge t) \le \mathbb{P}(\|\frac{1}{\mathbf{n}}AA^\top - EAA^\top\|_{op} \ge \frac{t^2}{\mathbf{n}} - \|EAA^\top\|_{op}) \tag{103}$$

## J.2 Proofs of Lemma J.4, J.6, and J.7

*proof of Lemma J.4.* It is evident from Lemma I.3, Equation (24) in the proportional regime that *A, B, C,* and *D* hold. We employ proof techniques adapted from Moniri et al. [2024] to prove *E, F,* and *G.* For *E*, by Lemma J.6, with $t = \log^{\frac{1}{2}}\mathbf{n}$, $M_a \le \frac{C\log^{\frac{1}{2}}\mathbf{n}}{\sqrt{\mathbf{n}}}$, w.p. $1 - o(1)$.

For *F*,
$$\mathbb{P}(C|x^\top\beta| \geq t) = \mathbb{P}(C|x^\top\beta - Ex^\top\beta + Ex^\top\beta| \geq t)$$
$$\leq \mathbb{P}(C|x^\top\beta - Ex^\top\beta| \geq t - C|Ex^\top\beta|) \leq 2\exp(-ct^2). \tag{104}$$

Then, $\mathbb{P}(|x^\top\beta| \geq t) \leq 2\exp(-c(t - Ex^\top\beta)^2) \leq 2\exp(-ct^2)$.

Therefore, $M_b \leq C\log^{\frac{1}{2}} n$, w.p. $1 - o(1)$ with $t = \log^{\frac{1}{2}}\mathbf{n}$.

For *G*, refer Moniri et al. [2024]. For *H*, refer Bai and Silverstein [2010] Corollary A.21. $\qquad\square$

*proof of Lemma J.6.* We use the Hoeffding inequality such that

$$\mathbb{P}(\|a\|_\infty \geq \frac{t}{\sqrt{\mathbf{N}}}) = \mathbb{P}\left(\max_i |a_i| \geq \frac{t}{\sqrt{\mathbf{N}}}\right) \leq \mathbb{P}\left(\bigcup_i \{|a_i| \geq \frac{t}{\sqrt{\mathbf{N}}}\}\right) \leq \sum_i \mathbb{P}\left(|a_i| \geq \frac{t}{\sqrt{\mathbf{N}}}\right)$$
$$\overset{\text{i.i.d.}}{=} \mathbf{N}\mathbb{P}\left(|a_i| \geq \frac{t}{\sqrt{\mathbf{N}}}\right) = \mathbb{P}(|\sqrt{\mathbf{N}}a_i| \geq t) \leq 2\mathbf{N}\exp(-ct^2) \tag{105}$$
$\square$

*Proof of Lemma J.7.* We use the properties of the norm.

$$\mathbb{P}(\|A\|_{op} \geq t) = \mathbb{P}(\|A\|_{op}^2 \geq t^2) = \mathbb{P}(\|\frac{1}{\mathbf{n}}AA^\top\|_{op} \geq \frac{t^2}{\mathbf{n}})$$
$$= \mathbb{P}(\|\frac{1}{\mathbf{n}}AA^\top - \mathbb{E}AA^\top + \mathbb{E}AA^\top\|_{op} \geq \frac{t^2}{\mathbf{n}})$$
$$\leq \mathbb{P}(\|\frac{1}{\mathbf{n}}AA^\top - \mathbb{E}AA^\top\|_{op} + \|\mathbb{E}AA^\top\|_{op} \geq \frac{t^2}{\mathbf{n}}) \tag{106}$$
$$= \mathbb{P}(\|\frac{1}{\mathbf{n}}AA^\top - \mathbb{E}(AA^\top)\|_{op} \geq \frac{t^2}{\mathbf{n}} - \mathbb{E}\|AA^\top\|_{op})$$
$\square$

## J.3 Expectation of the Polynomial form of Two Gaussian Random Variables

We first introduce Isserlis' theorem. This theorem allows the expectation of the product of centered Gaussian random variables to be expressed as a product of covariances, making the computation feasible.

**Theorem J.8** (Isserlis' Theorem [Isserlis, 1918, Vignat, 2011]). *Let* $X = (X_1, \cdots, X_d)$ *Gaussian random vector* s.t. $\mathbb{E}[X] = 0$ *, and let* $A = \{\alpha_1, \cdots, \alpha_N\}$ *be set of integers* s.t. $1 \leq \alpha_i \leq d$, $\forall i$. *Denote* $X_A = \prod_{\alpha_i \in A} X_{\alpha_i}$, *and* $X_\emptyset = 1$. *Let* $\prod(A)$ *denote partitions of A into disjoint pairs, and* $\sigma \in \prod(A)$ *is a pair.*
$$\mathbb{E}[X_A] = \sum_{\sigma \in \prod(A)} \prod_{(i,j) \in \sigma} \mathbb{E}[X_{\alpha_i} X_{\alpha_j}]\mathbf{1}_{\text{d is even}}. \tag{107}$$

**Lemma J.9** (Moment of Centered Bivariate Gaussian Variables). *For* $d \in \mathbb{N}$, $u, v \in \mathbb{R}^d$, $g \sim \mathcal{N}(0, I_d)$, $\bar{Z}_1 = \langle u, g \rangle$, $\bar{Z}_2 = \langle v, g \rangle$.
$$\begin{pmatrix} \bar{Z}_1 \\ \bar{Z}_2 \end{pmatrix} \sim n\left( \begin{pmatrix} 0 \\ 0 \end{pmatrix}, \begin{pmatrix} \|u\|^2 & \langle u, v \rangle \\ \langle v, u \rangle & \|v\|^2 \end{pmatrix} \right) \tag{108}$$

$X_{\alpha_i}$ *is defined at Thm. J.8*

$$\mathbb{E}_{\bar{Z}_1, \bar{Z}_2}[H_j(\bar{Z}_1)\bar{Z}_2^k] = j! \sum_{m=0}^{\lfloor \frac{j}{2} \rfloor} \frac{(-1)^m}{m!(j-2m)!2^m} \sum_{\sigma \in \prod(\{\{\bar{Z}_1\} \times j - 2m\} \cup \{\{\bar{Z}_2\} \times k\})} \prod_{(p,q) \in \sigma} \mathbb{E}[X_{\alpha_p} X_{\alpha_q}]\mathbf{1}_{\text{j+k-2m is even}}$$

$$\mathbb{E}_{\bar{Z}_1, \bar{Z}_2}[\bar{Z}_1^j \bar{Z}_2^k] = \sum_{\sigma \in \prod(\{\{\bar{Z}_1\} \times j\} \cup \{\{\bar{Z}_2\} \times k\})} \prod_{(p,q) \in \sigma} \mathbb{E}[X_{\alpha_p} X_{\alpha_q}]\mathbf{1}_{\text{j+k is even}}$$
$$\tag{109}$$

*Proof.* By the explicit formula of Hermite polynomials

$$\mathbb{E}_{\bar{Z}_1,\bar{Z}_2}[H_j(\bar{Z}_1)(\bar{Z}_2)^k] = j! \sum_{m=0}^{\lfloor \frac{j}{2} \rfloor} \frac{(-1)^m}{m!(j-2m)!2^m} \mathbb{E}_{\bar{Z}_1,\bar{Z}_2}[\bar{Z}_1^{j-2m}\bar{Z}_2^k] \tag{110}$$

Therefore, we need to figure out $\mathbb{E}_{\bar{Z}_1,\bar{Z}_2}[\bar{Z}_1^p \bar{Z}_2^q]$. We know $\bar{Z}_1, \bar{Z}_2$ is a mean-zero Gaussian. Thus, we can apply Theorem J.8 with $A = \{\{\bar{Z}_1\} \times p\} \cup \{\{\bar{Z}_2\} \times q\}\}$, $\mathbb{E}[\bar{Z}_1^p \bar{Z}_2^q] = \sum_{\sigma \in \prod(A)} \prod_{(\tau,\upsilon) \in \sigma} \mathbb{E}[X_{\alpha_\tau} X_{\alpha_\upsilon}].\mathbf{1}_{\text{p+q is even}}$

$\square$

**Corollary J.10** (Corollary of Lemma J.9, Moment of Centered Univariate Gaussian Variables)**.** *Remark $\bar{Z}_1 \sim \mathcal{N}(0, \|u\|^2)$ For the case $k = 0$,*

$$\mathbb{E}_{\bar{Z}_1}[\bar{Z}_1^j] = \|u\|^j (j-1)!! \mathbf{1}_{j \text{ is even}} \tag{111}$$

*Proof.*

$$\begin{aligned}
\mathbb{E}_{\bar{Z}_1,\bar{Z}_2}[\bar{Z}_1^j \bar{Z}_2^k] = \mathbb{E}_{\bar{Z}_1}[\bar{Z}_1^j] &= \sum_{\sigma \in \prod(\{\bar{Z}_1\} \times j)} \prod_{(p,q) \in \sigma} \mathbb{E}[X_{\alpha_p} X_{\alpha_q}] \mathbf{1}_{j \text{ is even}} \\
&= \sum_{\sigma \in \prod(\{\bar{Z}_1\} \times j)} \prod_{(p,q) \in \sigma} \|u\|^2 \mathbf{1}_{j \text{ is even}} = \sum_{\sigma \in \prod(\{\bar{Z}_1\} \times j)} \|u\|^j \mathbf{1}_{j \text{ is even}} = (j-1)!! \|u\|^j \mathbf{1}_{j \text{ is even}}
\end{aligned} \tag{112}$$

$\square$

The following Corollary, which calculates the Expectation of the Power of a Gaussian Random Variable, can be derived using the binomial expansion with the reparametrization technique and Corollary J.10. It corresponds to the case $k = 0$ in Lemma J.9.

**Corollary J.11** (Corollary of Lemma J.9, Moments of Gaussian Variables)**.** *Given $\omega \in \mathbb{R}^d$, let Gaussian Random Variable $Z \sim \mathcal{N}(\mu^\top \omega, \|\omega\|^2)$, then*

$$\begin{aligned}
\mathbb{E}_Z(Z)^k &= \sum_{t=0}^k \binom{k}{t} (\mu^\top \omega)^{k-t} \mathbb{E}_{\bar{Z} \sim \mathcal{N}(0, \|\omega\|^2)}[\bar{Z}^\top] \\
&= \sum_{t=0}^k \binom{k}{t} (\mu^\top \omega)^{k-t} (t-1)!! \cdot \|\omega\|^\top \mathbf{1}_{t \text{ is even}}.
\end{aligned} \tag{113}$$

The following Corollary, which computes the Gaussian expectation of Hermite polynomials, is derived from the explicit form of Hermite polynomials and Corollary J.10. It corresponds to the case $k = 0$ in Theorem H.7.

**Corollary J.12** (Corollary of Theorem H.7 Gaussian Expectation of Hermite Polynomials)**.** *Given $\omega \in \mathbb{S}^{d-1}$, let Gaussian Random Variable $Z \sim \mathcal{N}(\mu^\top \omega, 1)$, then*

$$\begin{aligned}
\mathbb{E}_x[H_k(\omega^\top x)] &= \mathbb{E}_{\xi \sim \mathcal{N}(0,1)}[H_k(\omega^\top \mu + \xi)] \\
&= \sum_{j=0}^k \binom{k}{j} (\omega^\top \mu)^{\circ j} E[H_k(\xi) H_0(\xi)] = (\omega^\top \mu)^k
\end{aligned} \tag{114}$$

## K    Empirical Insights into High-Dimensional Asymptotics

In asymptotic analysis, $\mathbf{n}, \mathbf{d}, \mathbf{N} \rightarrow \infty$ is crucial for observing the result. Please see Figure 9, Figure 10 for the *Cohesion* and *Separability* in $\mathbb{R}^{2000}, \mathbb{R}^{20000}, \mathbb{R}^{320000}$, respectively. As the dimension increases, the range where *Cohesion* and *Separability* align with our expectations.

For component analysis, please refer to Figure 11, Figure 12, and Figure 13 for *Cohesion*, which demonstrates progressively wider ranges, and Figure 14, Figure 15, and Figure 16 for *Separability*, which exhibits a same ranges. We compare *Cohesion* and *Separability* from the predominant $s_L$ term (Equation (6) and Equation (7)), with the suppressed contributions from other terms of *Cohesion* and *Separability* of $F_L$ (Equation (67) and Equation (68)).

## L    Additional Observation of Multi Classes Feature Analysis

We conduct experiments using new input data points $x_1$ and $x_2$, and *spike directions* $\beta_2$ and $\beta_3$, which are not orthogonal to $x_1$ and $x_2$, as well as $\beta_4$, which is orthogonal to both in the main text. Furthermore, we extended the experiment by incorporating the midpoint direction $\beta_1 \triangleq \frac{x_1+x_2}{2}$. See Figure 17. Consistent with our analysis, we observed that $\beta_1$ also does not contribute significantly to feature formation.

To isolate the source of this behavior, we decomposed the network function $F_L$ into $F_{0,L}$ and $s_L$, and conducted additional experiments. See Figure 18. This decomposition confirms that the observed phenomenon primarily originates from $s_L$.

We also provide the original (unprocessed) data used before *expressibility* computation. See Figure 19, Figure 20, Figure 21, and Figure 22. *Expressibility* is the maximum feature distance achievable when two data points are rotated to maximize their angular separation. This corresponds to the slope of the original plot before scaling.

We summarize these slope values in Figure 23 to offer an alternative perspective. The results demonstrate that including either midpoint or orthogonal directions during training leads to negligible changes in slope.

These observations imply that training data aligned with the midpoint or orthogonal directions of new input data do not contribute meaningfully to feature extraction.

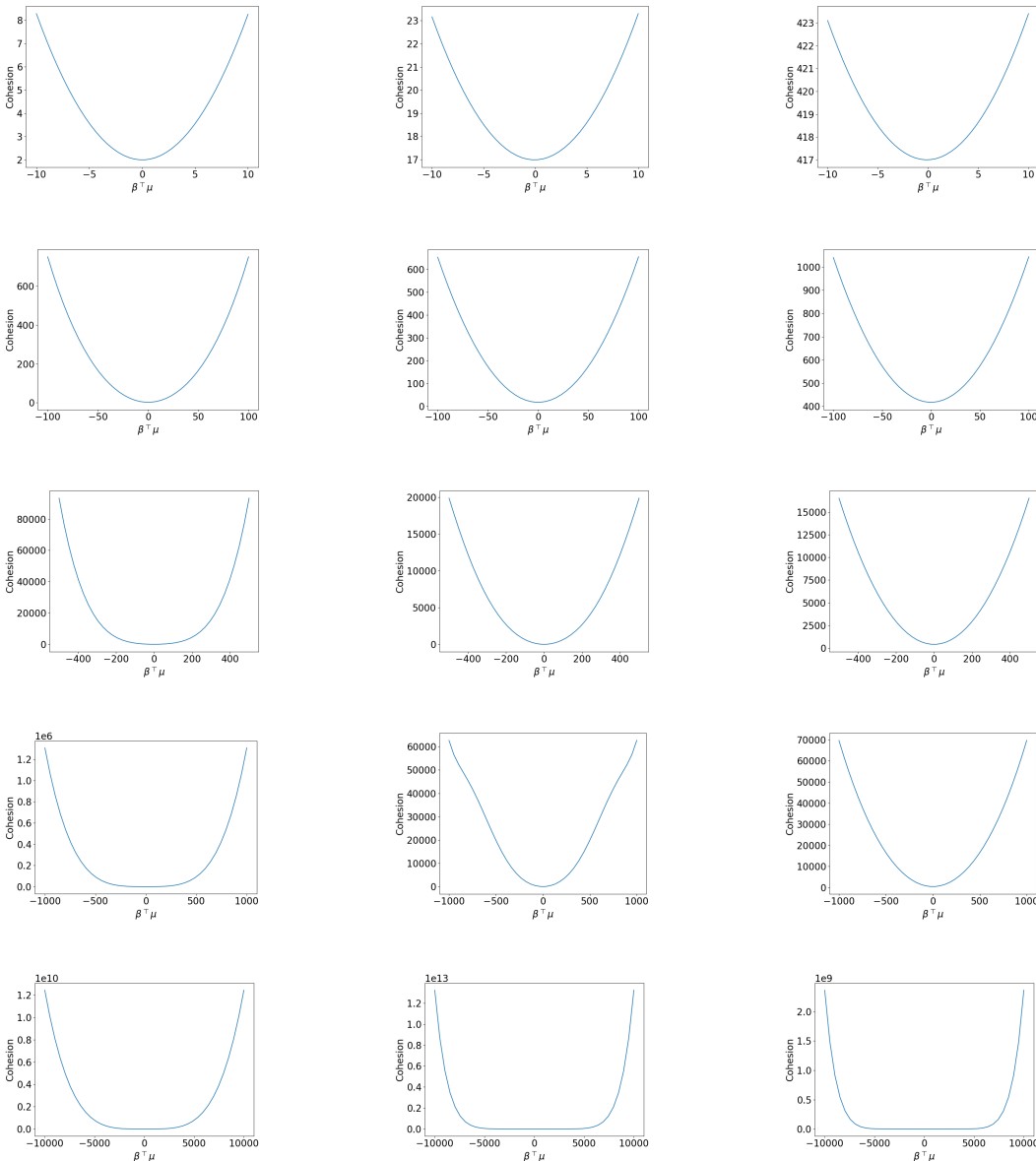

Figure 9: Cohesion in $\mathbb{R}^{2000}$, $\mathbb{R}^{20000}$, $\mathbb{R}^{320000}$ (left to right), with the computed range expanding from top to bottom.

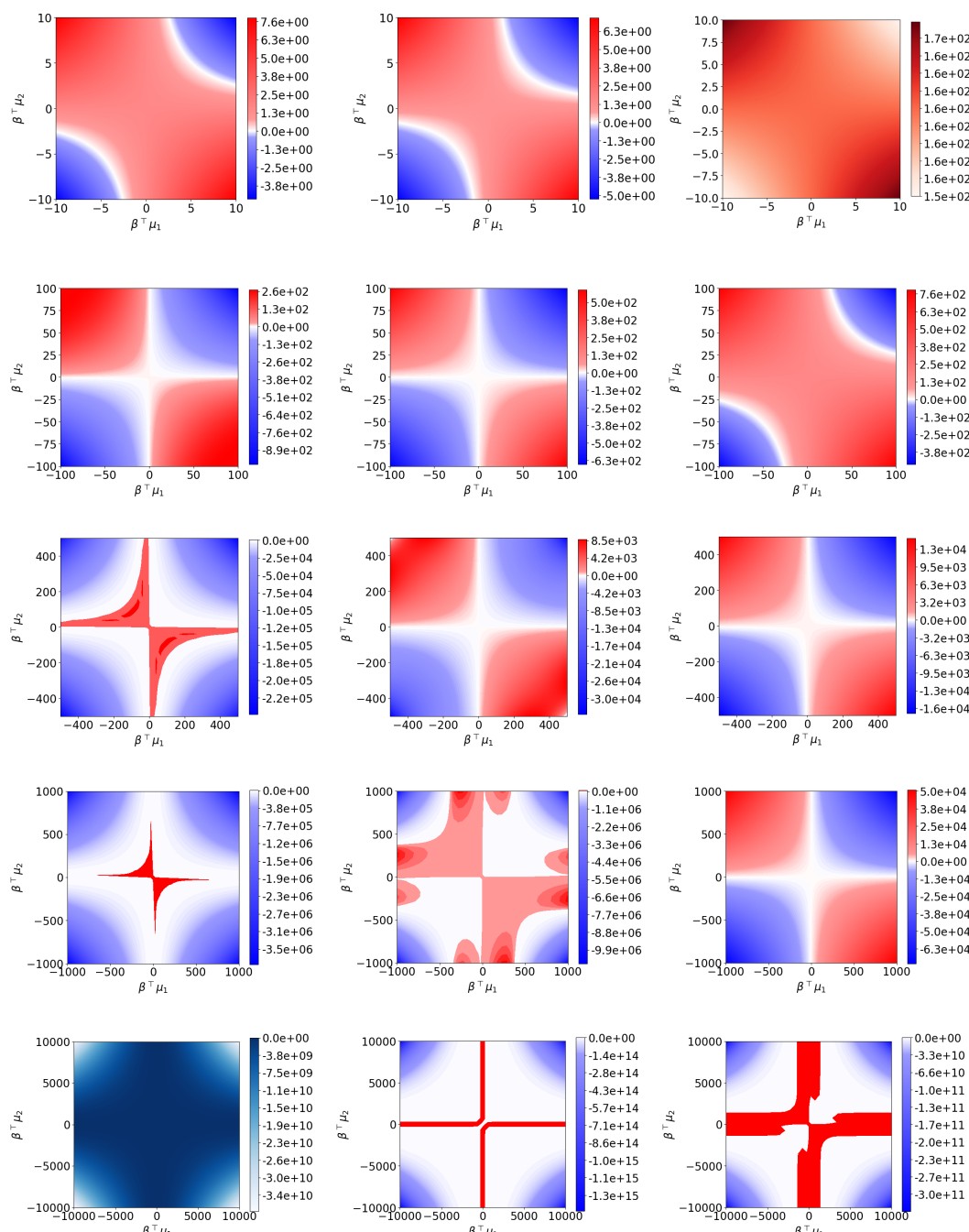

Figure 10: Separability in $\mathbb{R}^{2000}, \mathbb{R}^{20000}, \mathbb{R}^{320000}$ (left to right), with the computed range expanding from top to bottom.

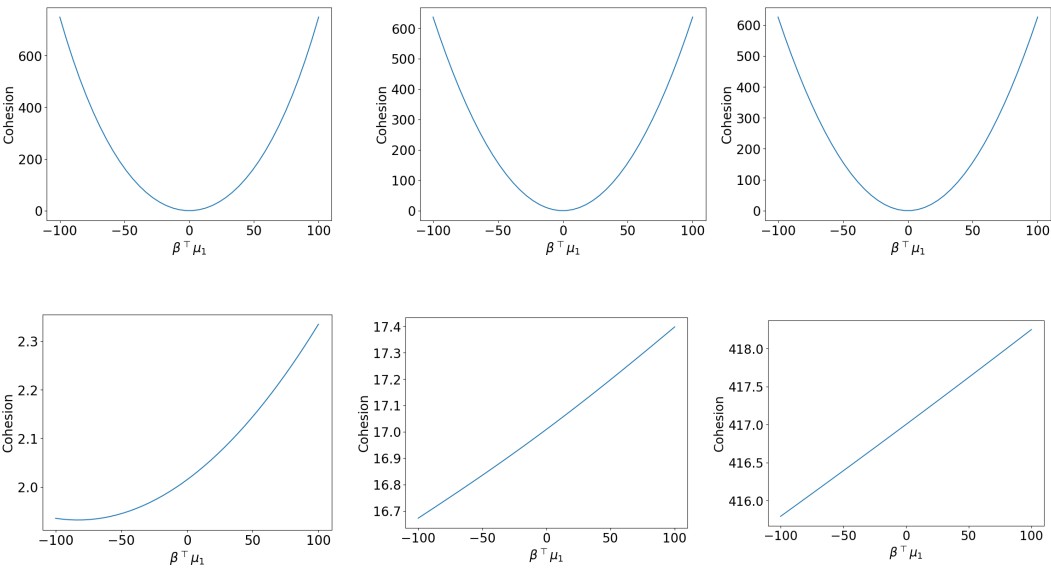

Figure 11: Component analysis of *Cohesion* in $\mathbb{R}^{2000}, \mathbb{R}^{20000}, \mathbb{R}^{320000}$ (left to right) in the range $[-100, 100]$. Top: *Cohesion* of the dominant component $s_L$ i.e. Equation (6). Bottom: sum of the other terms in Equation (67), which make only suppressed contributions.

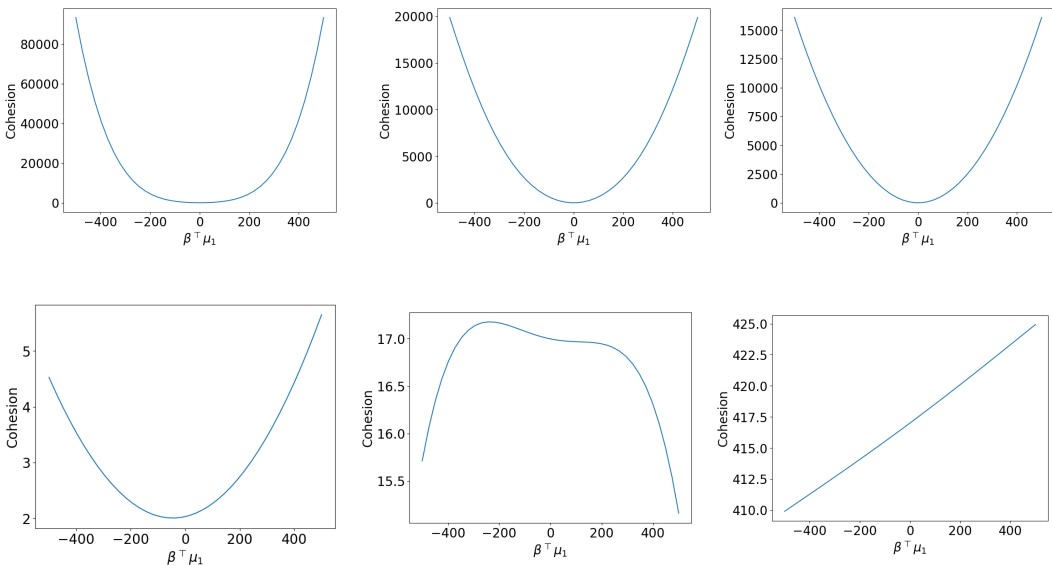

Figure 12: Component analysis of *Cohesion* in $\mathbb{R}^{2000}, \mathbb{R}^{20000}, \mathbb{R}^{320000}$ (left to right) in the range $[-500, 500]$. Top: *Cohesion* of the dominant component $s_L$ i.e. Equation (6). Bottom: sum of the other terms in Equation (67), which make only suppressed contributions.

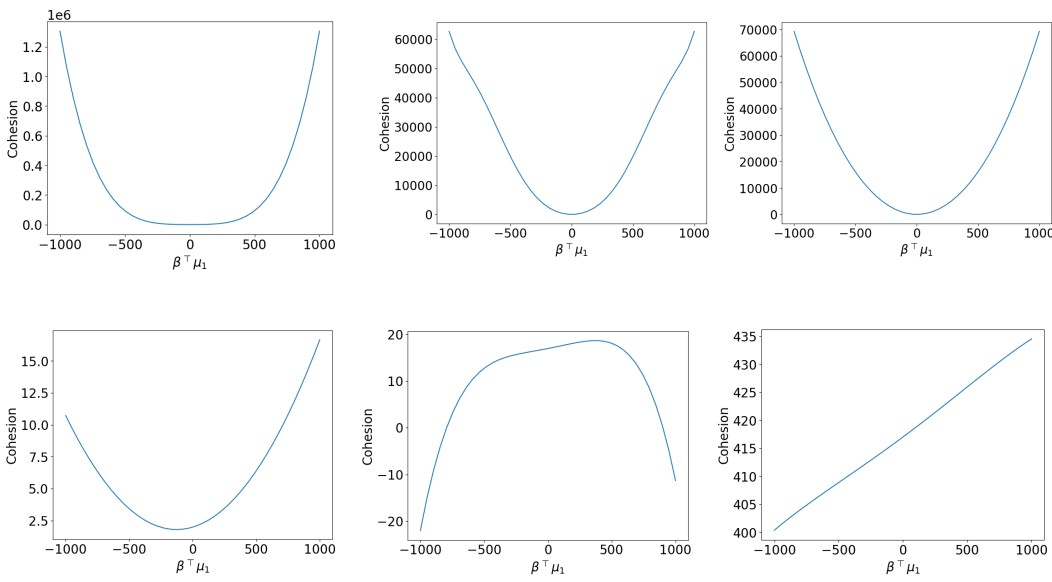

Figure 13: Component analysis of *Cohesion* in $\mathbb{R}^{2000}, \mathbb{R}^{20000}, \mathbb{R}^{320000}$ (left to right) in the range $[-1000, 1000]$. Top: *Cohesion* of the dominant component $s_L$ i.e. Equation (6). Bottom: sum of the other terms in Equation (67), which make only suppressed contributions.

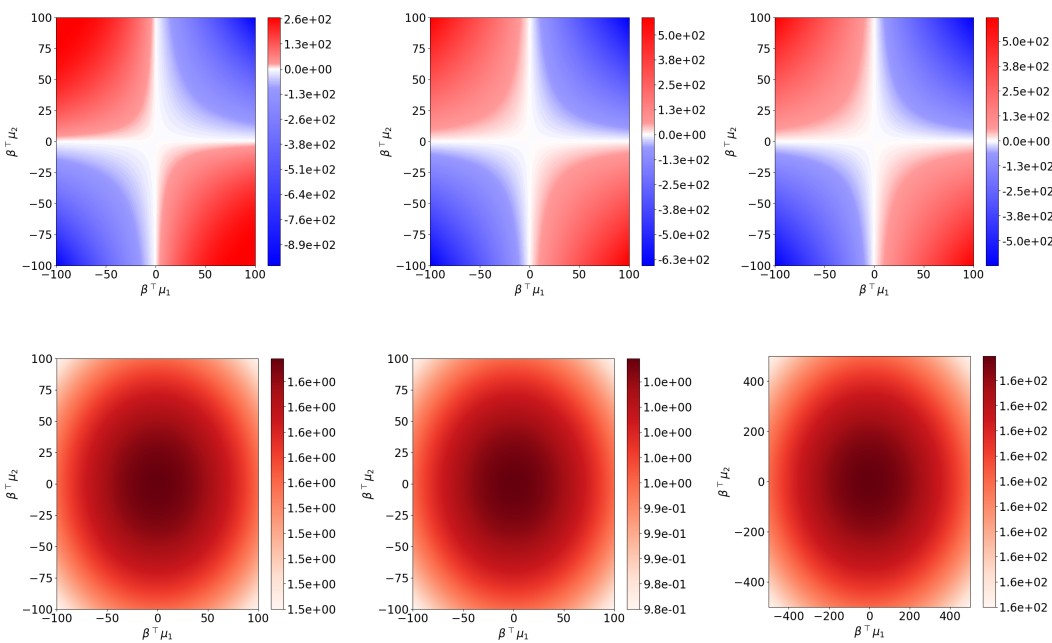

Figure 14: Component analysis of *Separability* in $\mathbb{R}^{2000}, \mathbb{R}^{20000}, \mathbb{R}^{320000}$ (left to right) in the range $[-500, 500]$. Top: *Separability* of the dominant component $s_L$ i.e. Equation (7). Bottom: sum of the other terms in Equation (68), which make only suppressed contributions.

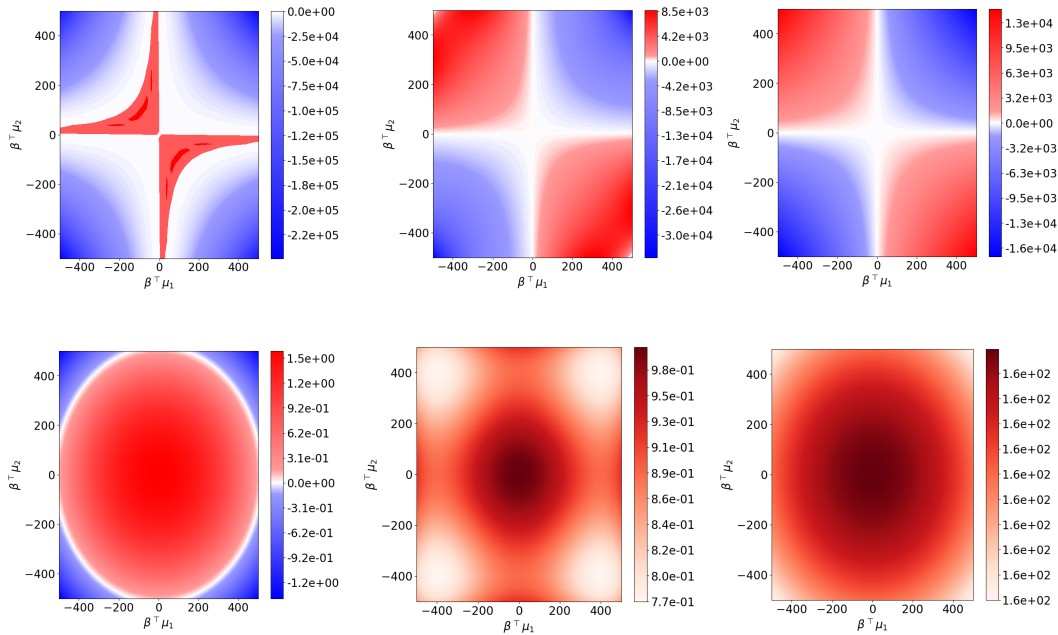

Figure 15: Component analysis of *Separability* in $\mathbb{R}^{2000}, \mathbb{R}^{20000}, \mathbb{R}^{320000}$ (left to right) in the range $[-500, 500]$. Top: *Separability* of the dominant component $s_L$ i.e. Equation (7). Bottom: sum of the other terms in Equation (68), which make only suppressed contributions.

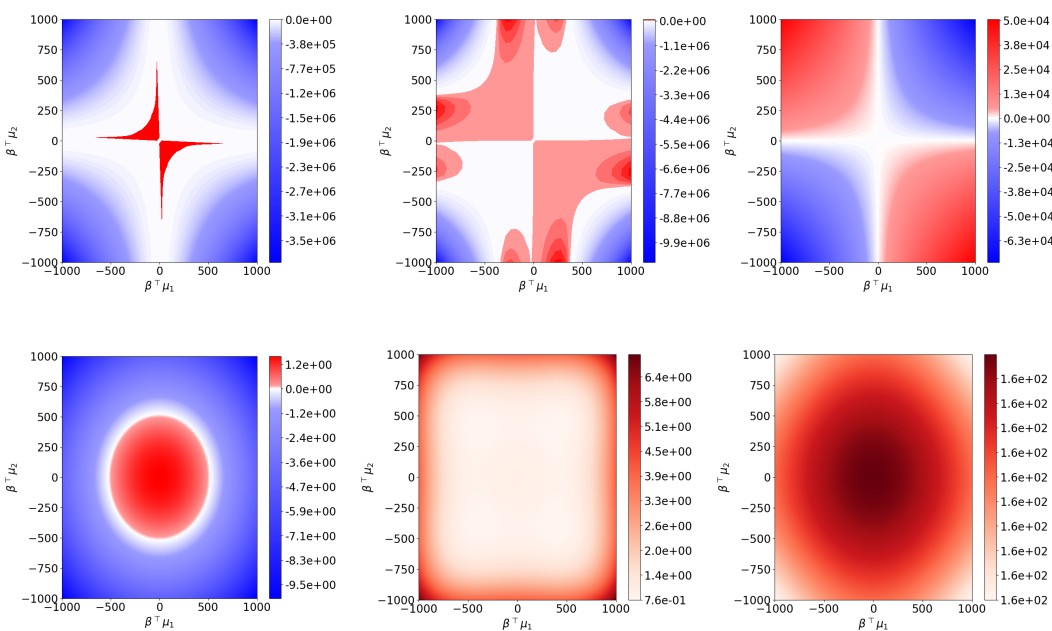

Figure 16: Component analysis of *Separability* in $\mathbb{R}^{2000}, \mathbb{R}^{20000}, \mathbb{R}^{320000}$ (left to right) in the range $[-1000, 1000]$. Top: *Separability* of the dominant component $s_L$ i.e. Equation (7). Bottom: sum of the other terms in Equation (68), which make only suppressed contributions.

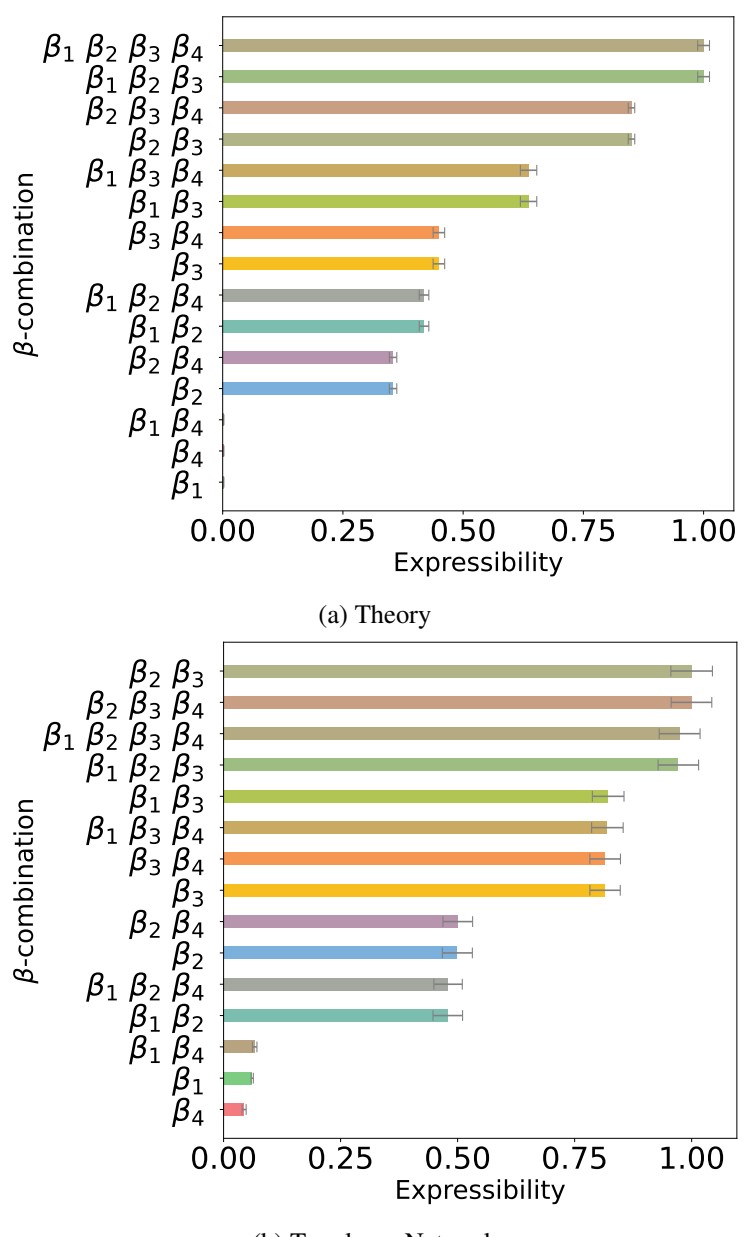

(a) Theory

(b) Two-layer Network

Figure 17: *Expressibility* measurement with $F, F_L$ of $\sum_k \binom{4}{k} - 1$ combinations of $\beta_i$s. All cases are strongly influenced only by $\beta_2$ and $\beta_3$ directions. Thus when using only the $\beta_1$ or $\beta_4$ directions, the two features are always mapped to the nearly same position.

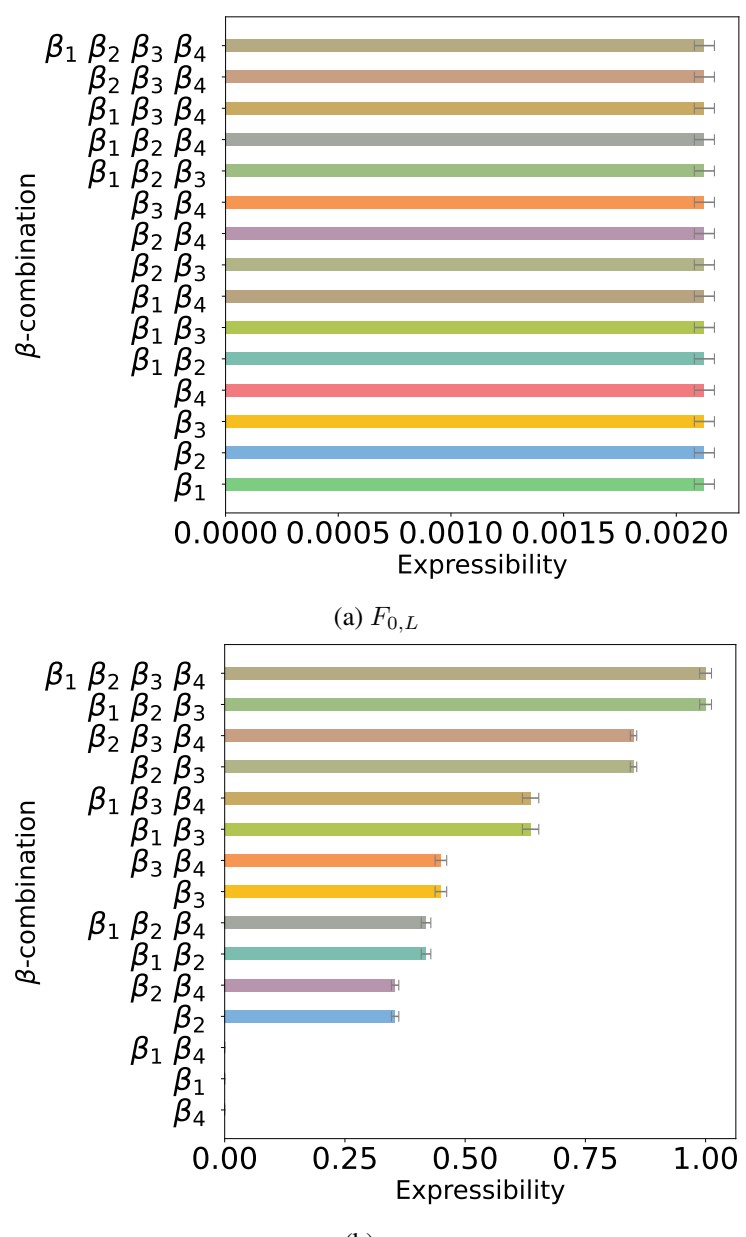

(a) $F_{0,L}$

(b) $s_L$

Figure 18: *Expressibility* measurement with $F_{0,L}, s_L$ of $\sum_k \binom{4}{k} - 1$ combinations of $\beta_i$s. $F_{0,L}$ is not influenced by trainind data and generates random features in all cases. $s_L$ is influenced only by $\beta_2$ and $\beta_3$. Thus when using only the $\beta_1$ or $\beta_4$, the two features are always mapped to the same position.

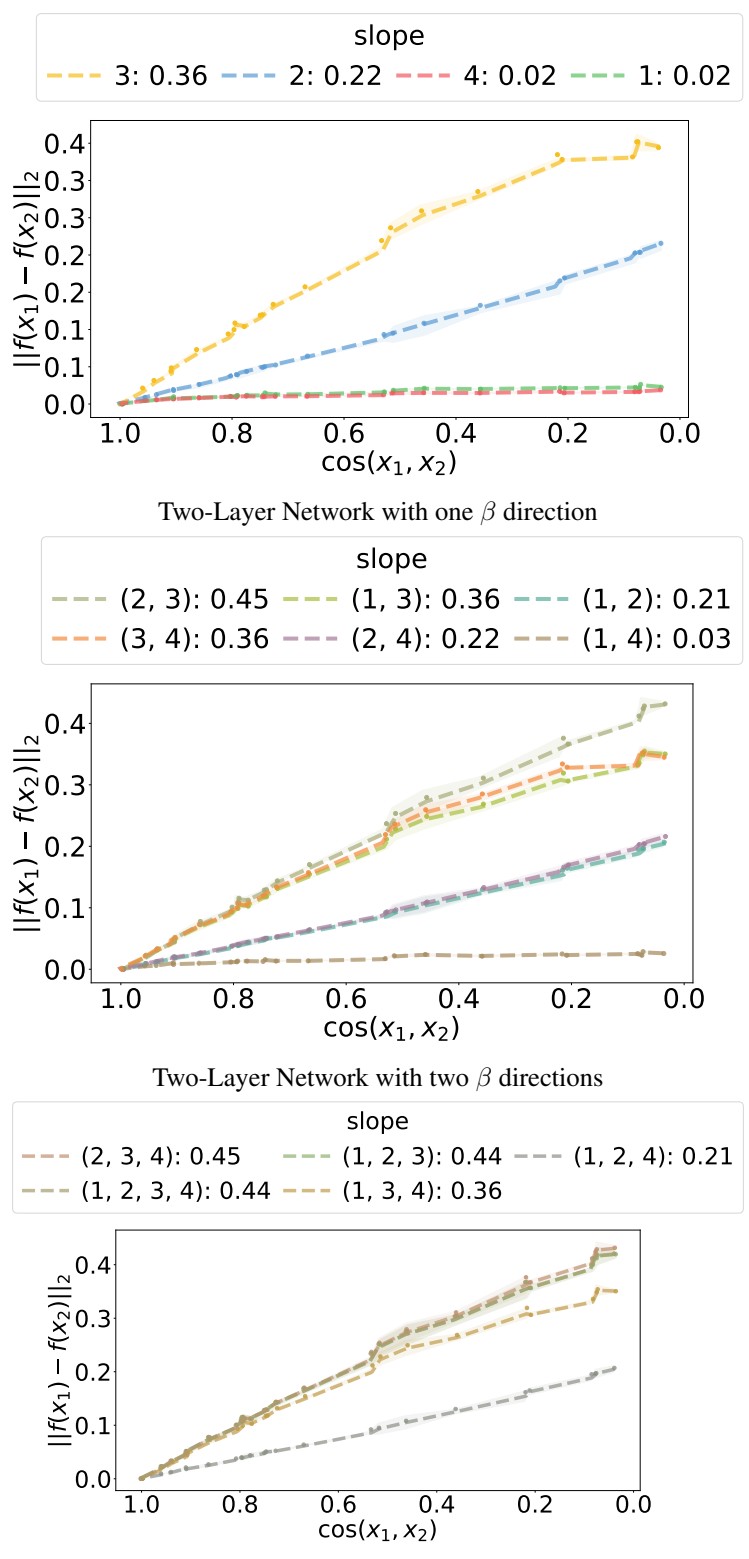

Figure 19: Experiments for $\beta_i$ combinations. We vary angle between $x_1, x_2$ and measure $L_2$ distance of two-layer network feature $F$. Scaled y-axis by dividing by $10^{-2}\sqrt{\mathbf{N}}$.

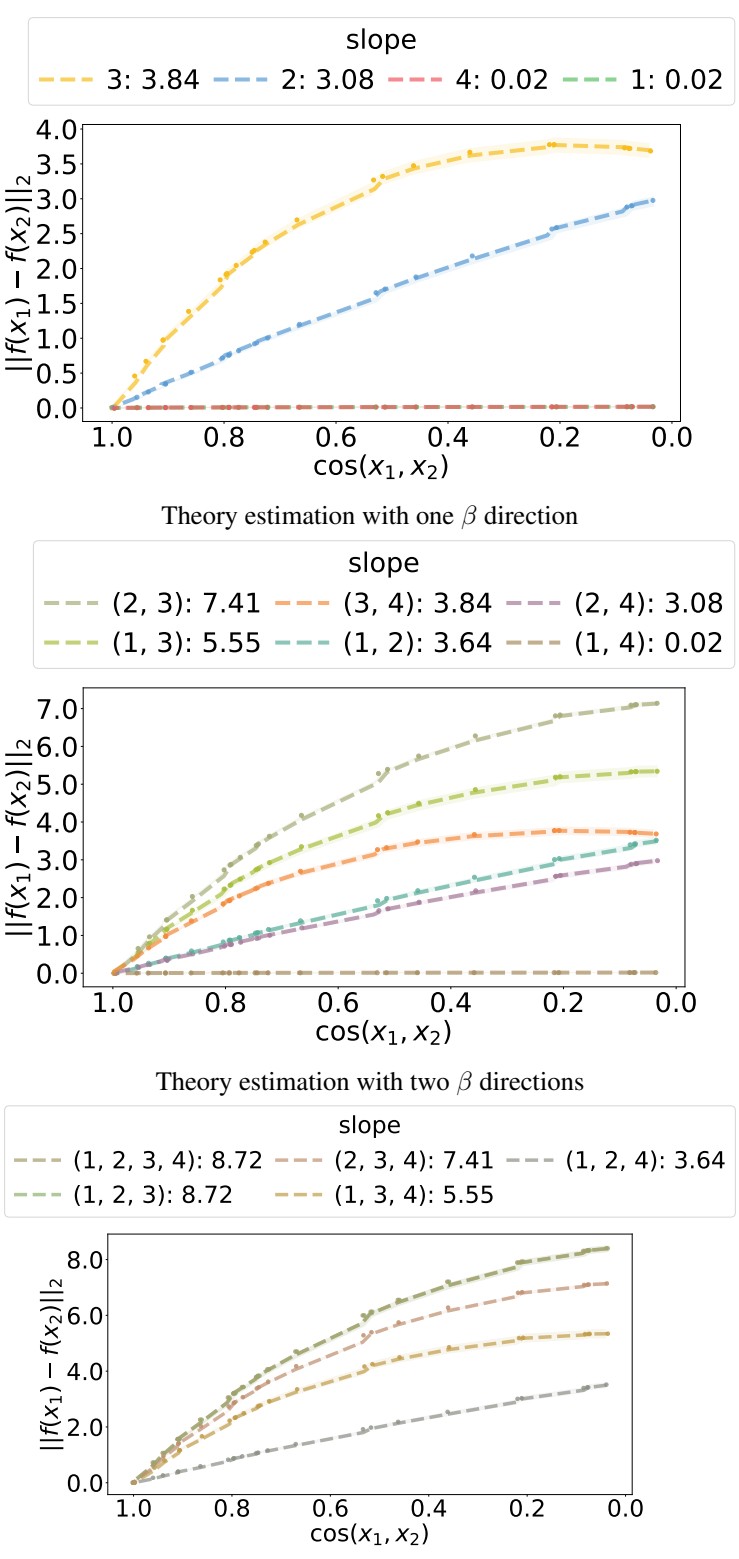

Figure 20: Experiments for $\beta_i$ combinations. We vary angle between $x_1, x_2$ and measure $L_2$ distance of approximated feature $F_L$. Scaled y-axis by dividing by $10^{-2}\sqrt{\mathbf{N}}$.

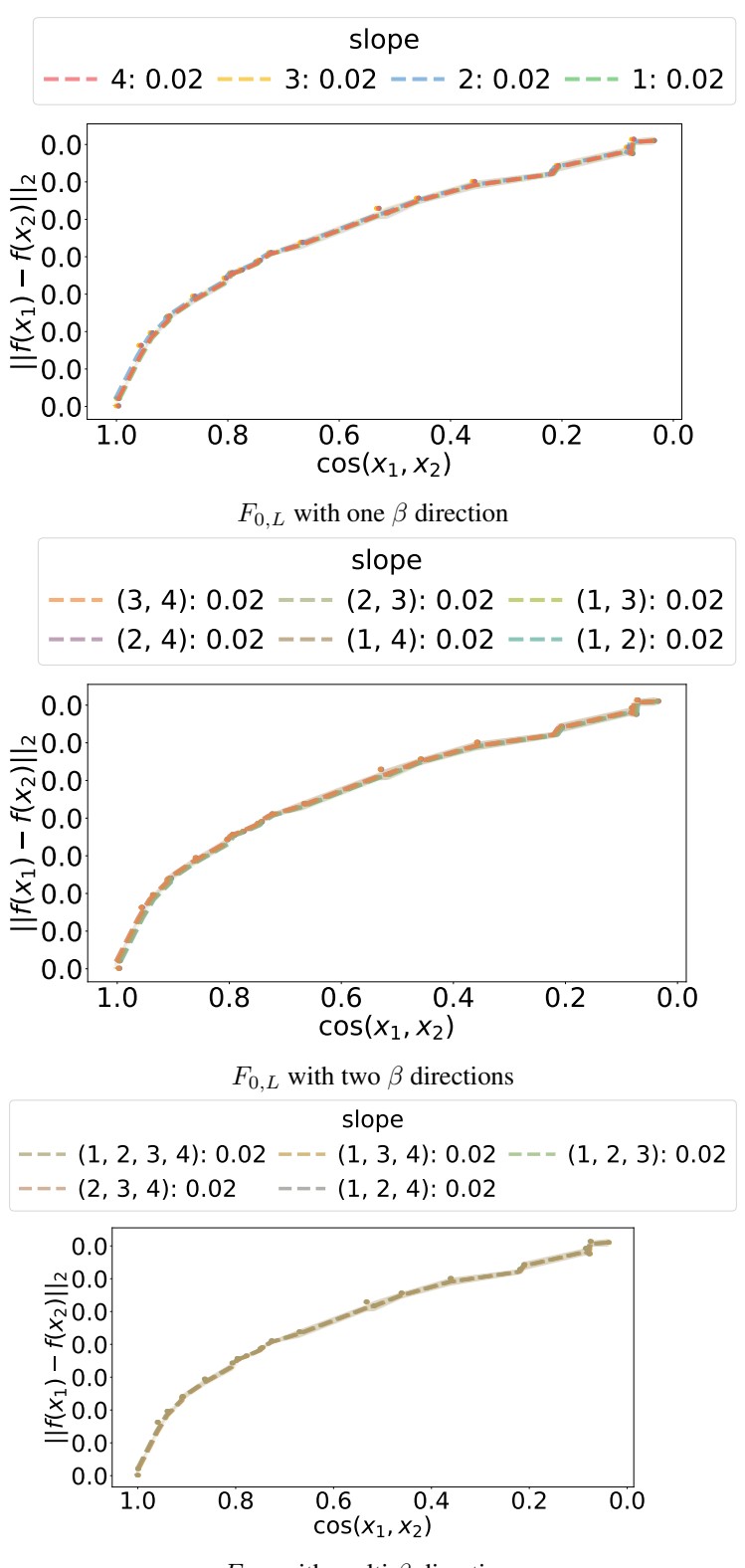

Figure 21: Experiments for $\beta_i$ combinations. We vary angle between $x_1, x_2$ and measure $L_2$ distance of approximated initialized feature $F_{0,L}$. Scaled y-axis by dividing by $10^{-2}\sqrt{\mathbf{N}}$.

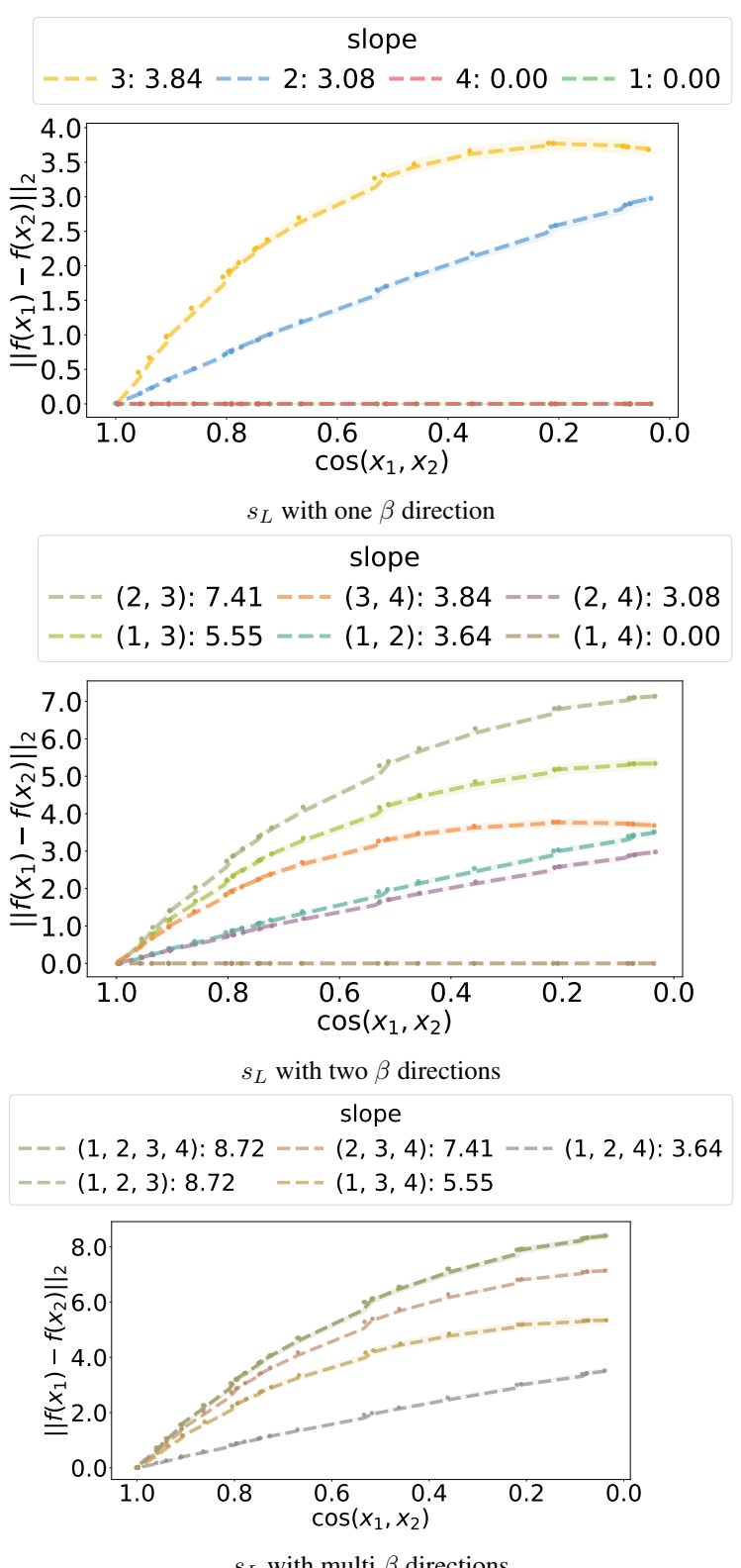

Figure 22: Experiments for $\beta_i$ combinations. We vary angle between $x_1, x_2$ and measure $L_2$ distance of approximated initialized feature $s_L$. Scaled y-axis by dividing by $10^{-2}\sqrt{\mathbf{N}}$.

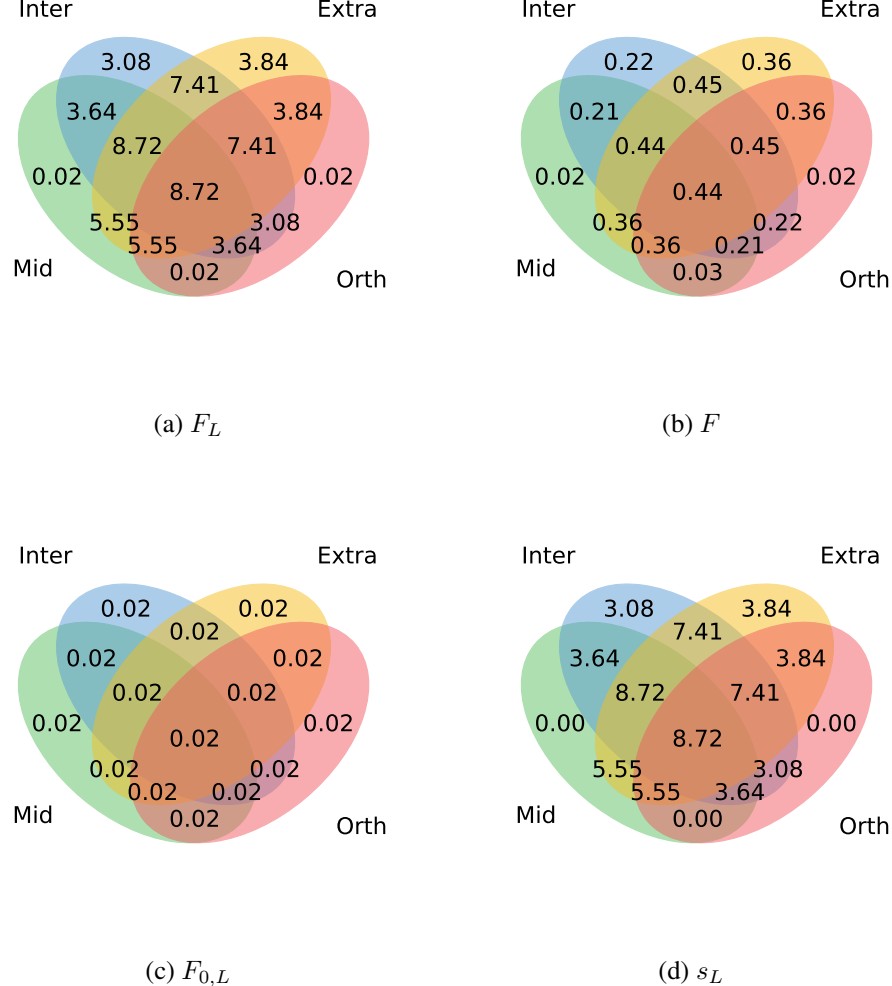

(a) $F_L$

(b) $F$

(c) $F_{0,L}$

(d) $s_L$

Figure 23: Slope measurements summarized in the Venn diagram from Figure 19, Figure 20, Figure 21, and Figure 22. Each intersection represents training with the corresponding combination of $\beta_i$ directions. $F_{0,L}$ is not influenced by trainind data and generates random features in all cases. $F, F_L$ and $s_L$ is influenced strongly by $\beta_2$ and $\beta_3$, so when using only the $\beta_1$ or $\beta_4$, the two features are always mapped to the nearly same position.

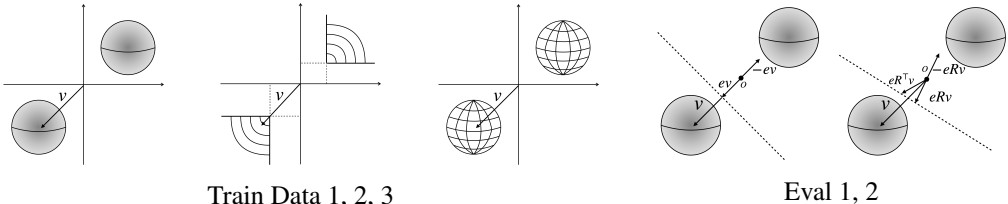

Train Data 1, 2, 3                    Eval 1, 2

Figure 24: Examples of training datasets (Data 1, 2, 3) and evaluation data Eval 1, 2.

## M   Additional Observations of two-classes Experiments

### M.1   Additional setup for Experiments I, II, III

We set $\mathbf{d} = \mathbf{n} = \mathbf{N} = 2^{11}$ and use Shifted ReLU.

**Training Datasets**   For illustration, see Figure 24. (Data 1) two uniform distributions over a radius-$\sqrt{\mathbf{d}}$ ball, (Data 2) two multi-dimensional element-wise truncated Gaussian distributions, and (Data 3) two uniform distributions over a radius-$\sqrt{\mathbf{d}}$ sphere, symmetric about the origin [4]. The two means of training class are denoted as $v$ and $-v$, respectively. For Data 1, 3, we set $v \triangleq 2r \cdot \mathbf{u}$, with $\mathbf{u} \sim \text{Unif}\left(\mathbb{S}^{\mathbf{d}-1}\right)$. For Data 2, one class has support on $[1, \infty)$ across all dimensions, while the other class has support on $(-\infty, -1]$.

**Evaluation Datasets**   Eval 1 and 2 use the projected Gaussian distribution, which is projected onto the mean direction of one training data $v$, as defined in Note M.1. For Eval 1, we translate the mean of the projected Gaussian distribution with $e$; for Eval 2, we rotate the mean of the projected Gaussian distribution with $R \in \mathcal{R}$ and fixed $e$. Note that we generate distinct rotation matrices $\mathcal{R}$ using the process in Appendix P.

*Note* M.1 (Sampling projected Gaussian distribution). For Eval 1, let $\nu \triangleq ev$, $c = 1$ and for Eval 2, let $\nu \triangleq Rev$, $c = 10^{-1}$ with $e = 0.01$ for Data 2 experiment and $e = 0.008$ for Data 1 and 3 experiments. Note $R \in SO(d)$. The projected Gaussian distribution is sampled as follows,

$$z - \frac{z^{\top}\nu\nu}{\|\nu\|^4} + \nu, \quad \text{where} \quad z \sim \mathcal{N}(0, cI). \tag{115}$$

### M.2   Comprehensive Results of All Experiments

The results of Expr. I are presented in linear scale in Figure 25 and logarithmic scale in Figure 26. Additionally, as demonstrated in Figure 4, experiments for Expr. II and III settings are shown in linear scale in Figure 27, with results for *Cohesion*, *Separability*, and *Recall@1* (IP). Furthermore, *Recall@1* (cos) results are presented in linear scale in Figure 28. *Recall@1* in both cosine and inner product follows a similar trend to *Separability*. All observed results align with the theoretical predictions.

### M.3   Empirical Observations in Multi-Step and Multi-Layer Settings

To further examine the potential validity of our theory in multi-step and multi-layer settings, we empirically observe that increasing $\beta^{\top}\mu$ generally leads to improvements in both *Cohesion* and *Separability*, consistent with our theoretical predictions.

#### M.3.1   Setups

To avoid gradient explosion or vanishing issues, we set the learning rate to 0.001 for multi-step experiments, and to 0.1 for multi-layer experiments. For the multi-step experiments, we performed gradient descent on all parameters, including $W$ and $a_{ij}$. For the multi-layer experiments, we

---

[4]The Sub-Gaussian property is proven for Data 1 and 3 in Vershynin [2018], and for Data 2 in Lemma I.1.

constructed deep networks by repeatedly applying the feature structure defined in Section 2. For example, in a three-layer setting, the network is defined as

$$F(x) \triangleq \sigma\big(W_2^\top \sigma(W_1^\top x)\big).$$

### M.3.2 Results

Please refer Figures 29, 30, 31, and 32 for results. For *Cohesion*, we observe that as training progresses or the network depth increases, *Cohesion* also increases with larger values of $\beta^\top \mu$. For *Separability*, although it generally tends to increase with larger $\beta^\top \mu$ as training proceeds or the network deepens, we note two exceptions: when training is extensive, *Separability* decrease at small $\beta^\top \mu$, and when the network is deeper, *Separability* decline at high $\beta^\top \mu$. These behaviors are likely due to nonlinear learning dynamics and the nonlinearity of the model. This suggests an interesting direction for future work: identifying a potential critical threshold of alignment strength in deep models that balances directional consistency and proximity.

### M.3.3 Insights

We discuss potential phenomena that may arise in multi-step training and deeper models, and consider the possible extensions of our theoretical framework.

**Multi-step training.** Our findings suggest that extending the analysis beyond a single-step update may provide additional insights. In particular, one possible hypothesis is that multi-step training progressively refines the spike component by amplifying directions that enhance *Cohesion* and *Separability* while suppressing others. Such a process could also facilitate more efficient information compression for classification and improved transferability. The analytical framework of Dandi et al. [2023], which characterizes the "staircase property" in multi-step dynamics, may offer a promising starting point for developing such an extension.

**Deeper models.** Although our two-layer model captures key aspects of feature learning, extending the framework to deeper architectures could shed further light on how transferable features emerge. Previous studies Fan and Wang [2020], Nichani et al. [2023], Wang et al. [2023] have already generalized conjugate kernel analysis to multi-layer settings, suggesting that a similar extension of our approach might be feasible. Interestingly, our preliminary multi-layer experiments indicate that when the depth increases, particularly in 3- and 4-layer models, improvements in *Separability* are not always sustained. Exploring this potential threshold effect represents an intriguing direction for future investigation.

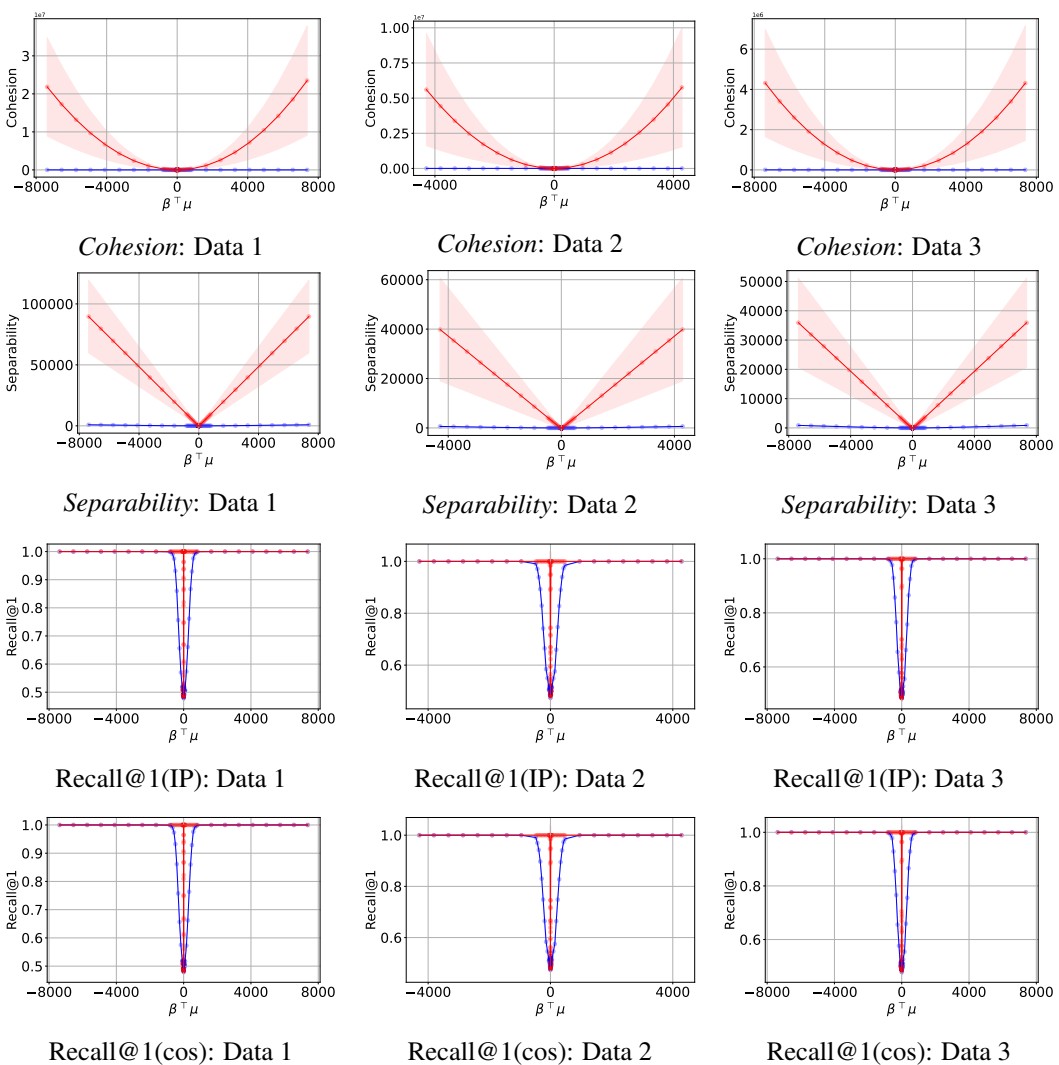

Figure 25: Expr. I: translation ($e$) variation case (linear scale). — is after one step training. — is from initialization. As the *train-unseen similarity* increases, both Cohesion and Separability become larger. *Recall@1* follows in a similar trend to *Separability*.

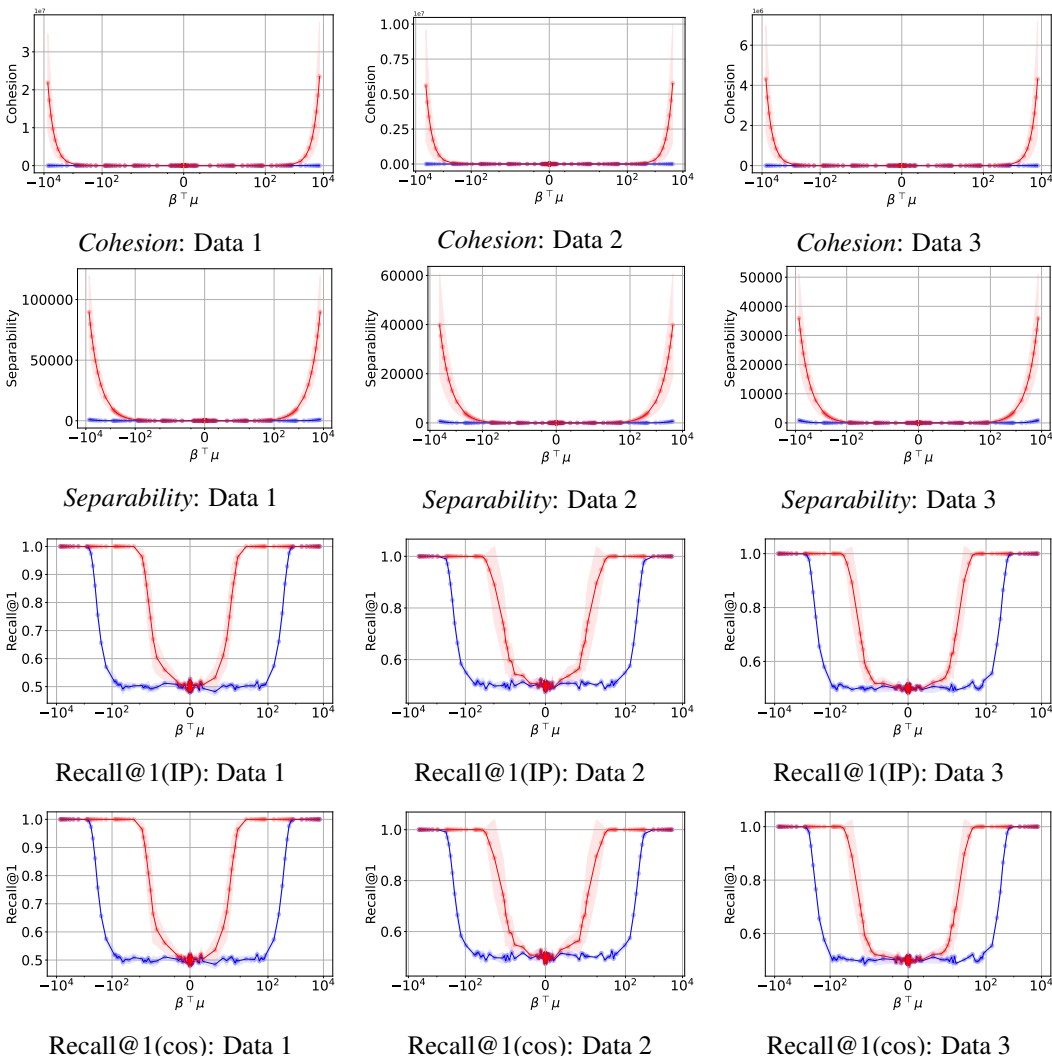

Figure 26: Expr. I: translation ($e$) variation (log scale). —— is after one step training. —— is from initialization. As the *train-unseen similarity* increases, both *Cohesion* and *Separability* become larger. *Recall@1* follows in a similar trend to *Separability*.

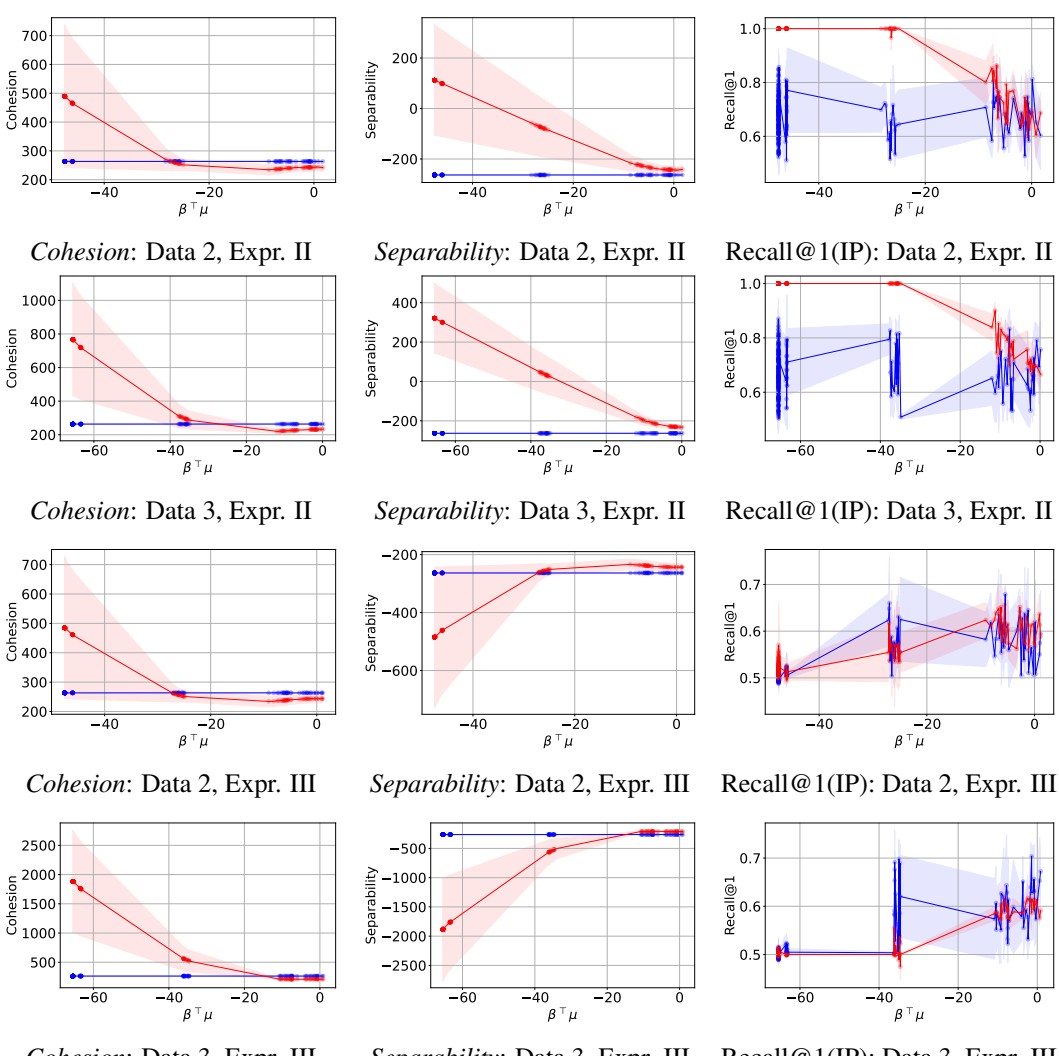

Figure 27: Expr. II and Expr. III; rotation($R$) variation (linear scale). —— is after one step training. —— is from initialization. In Expr. II, the two unseen classes are *assigned* to different classes, leading to increased *Cohesion* and *Separability* as the *train-unseen similarity* increases, consistent with our analysis. In contrast, in Expr. III, the two unseen classes are *assigned* to the same class, resulting in increased *Cohesion* and decreased *Separability* as the *train-unseen similarity* increases, which is in line with our analysis. *Recall@1* follows in a similar trend to *Separability*.

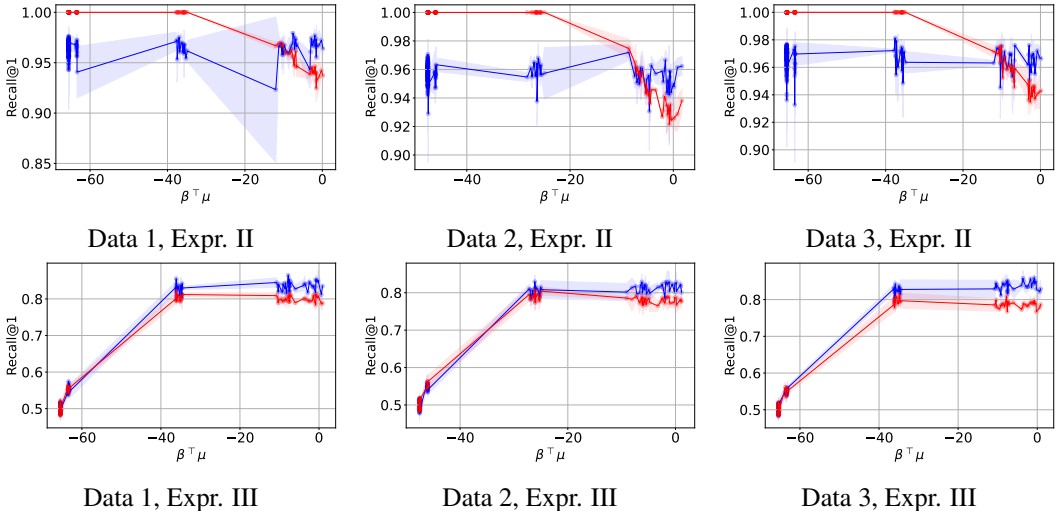

Figure 28: Recall@1 with cosine similarity of Expr. II and Expr. III; rotation($R$) variation (linear scale). —— is after one step training. —— is from initialization. *Recall@1* follows in a similar trend to corresponding *Separability*.

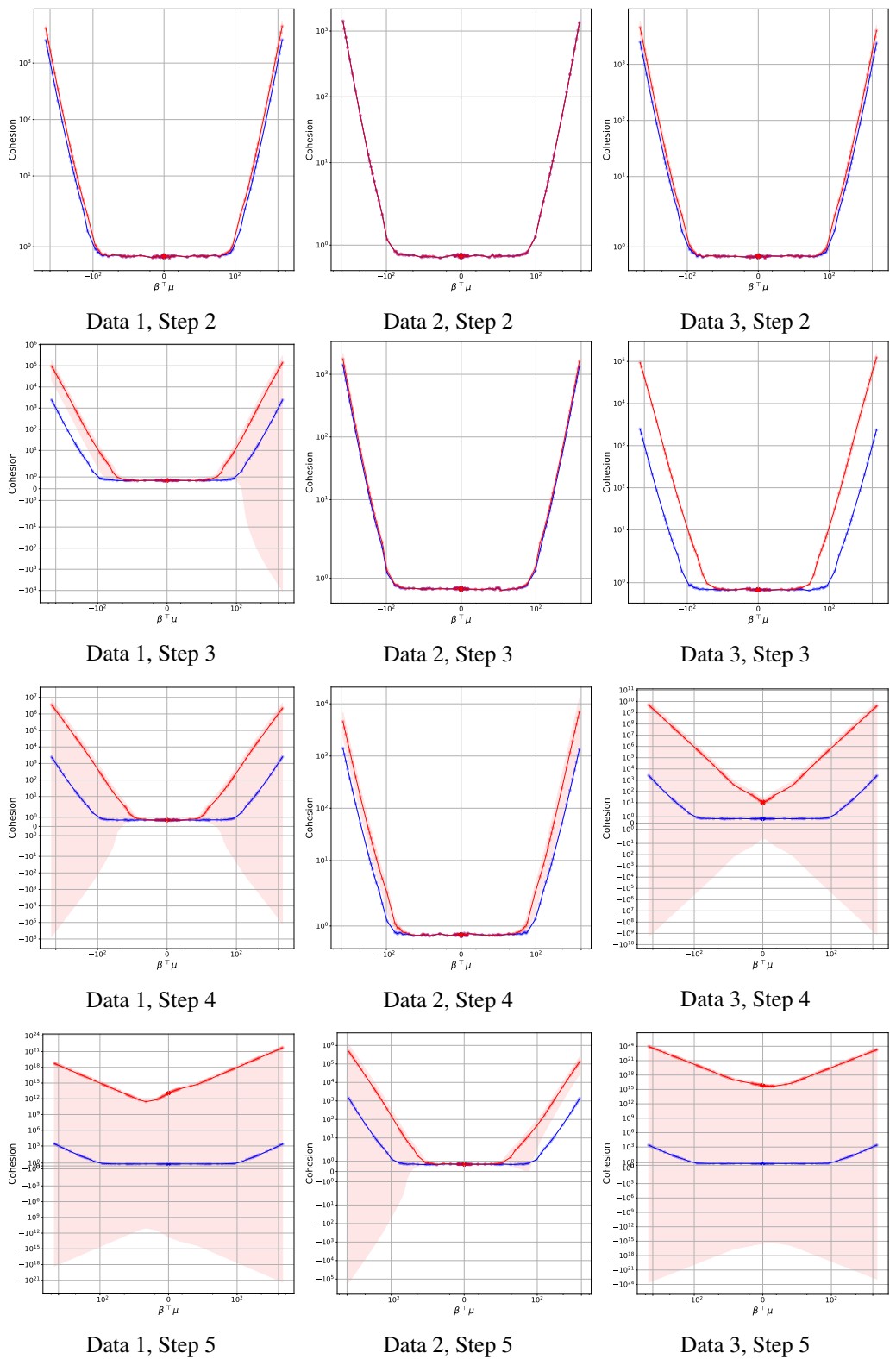

Figure 29: Evolution of *Cohesion* in a two-layer network over multiple steps: as training progresses, *Cohesion* gradually strengthens.

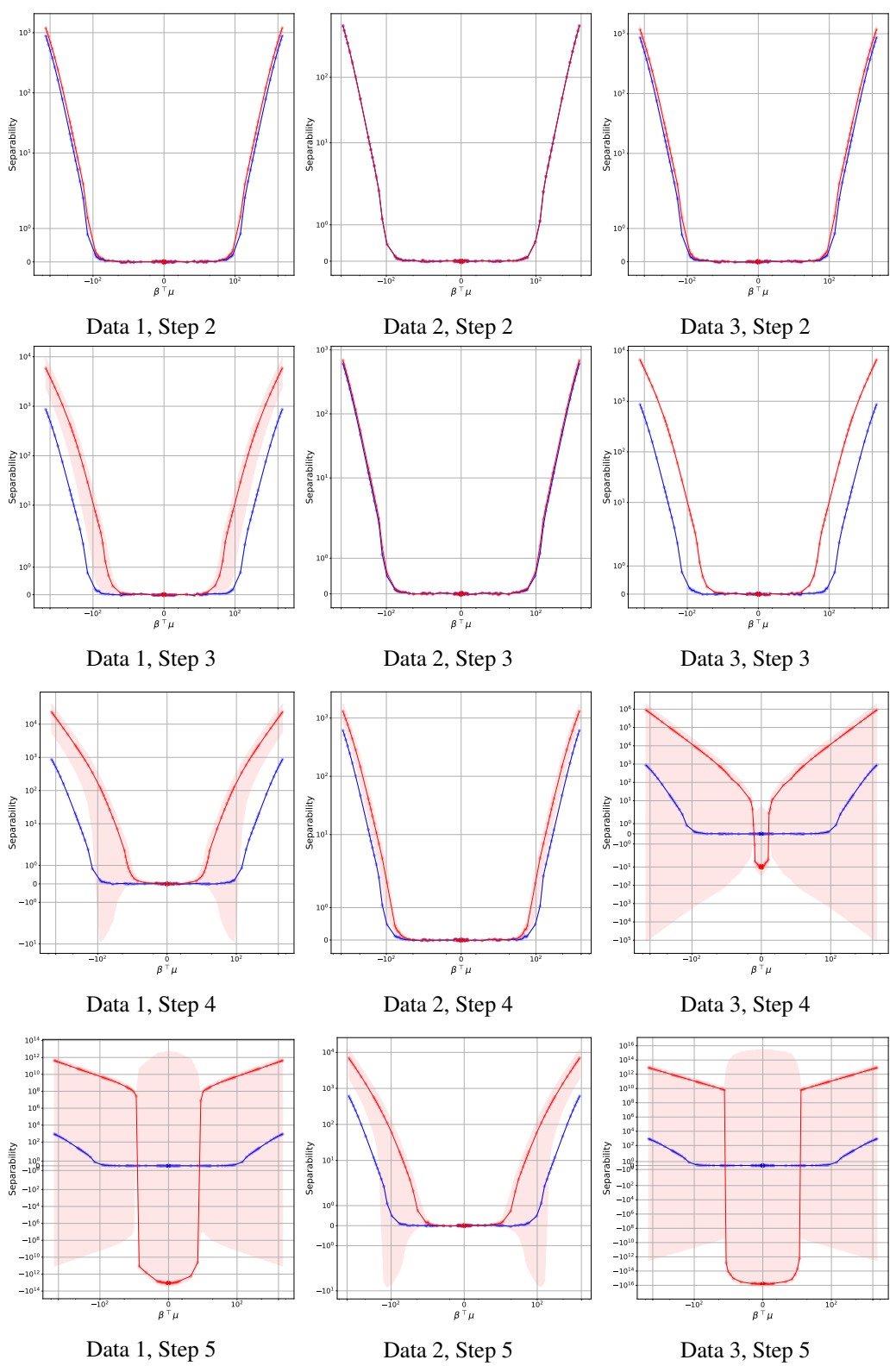

Figure 30: Evolution of *Separability* in a two-layer network over multiple steps: as training progresses, *Separability* gradually strengthens. Exceptionally, when a network is trained for many steps, there are cases where *Separability* decreases if the *train-unseen similarity* is very low.

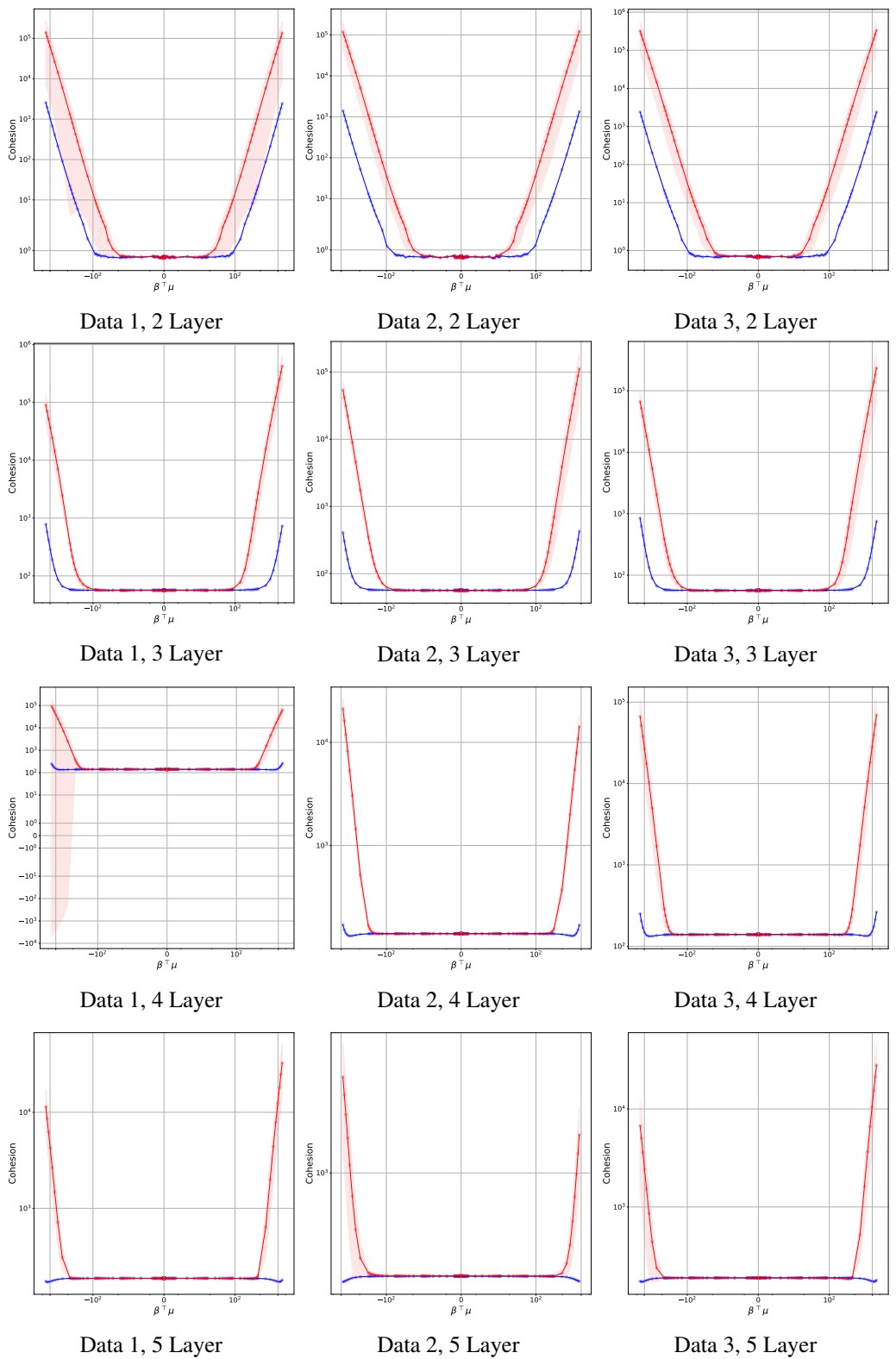

Figure 31: Evolution of *Cohesion* in a multi-layer network over single steps: as training progresses, *Cohesion* gradually strengthens.

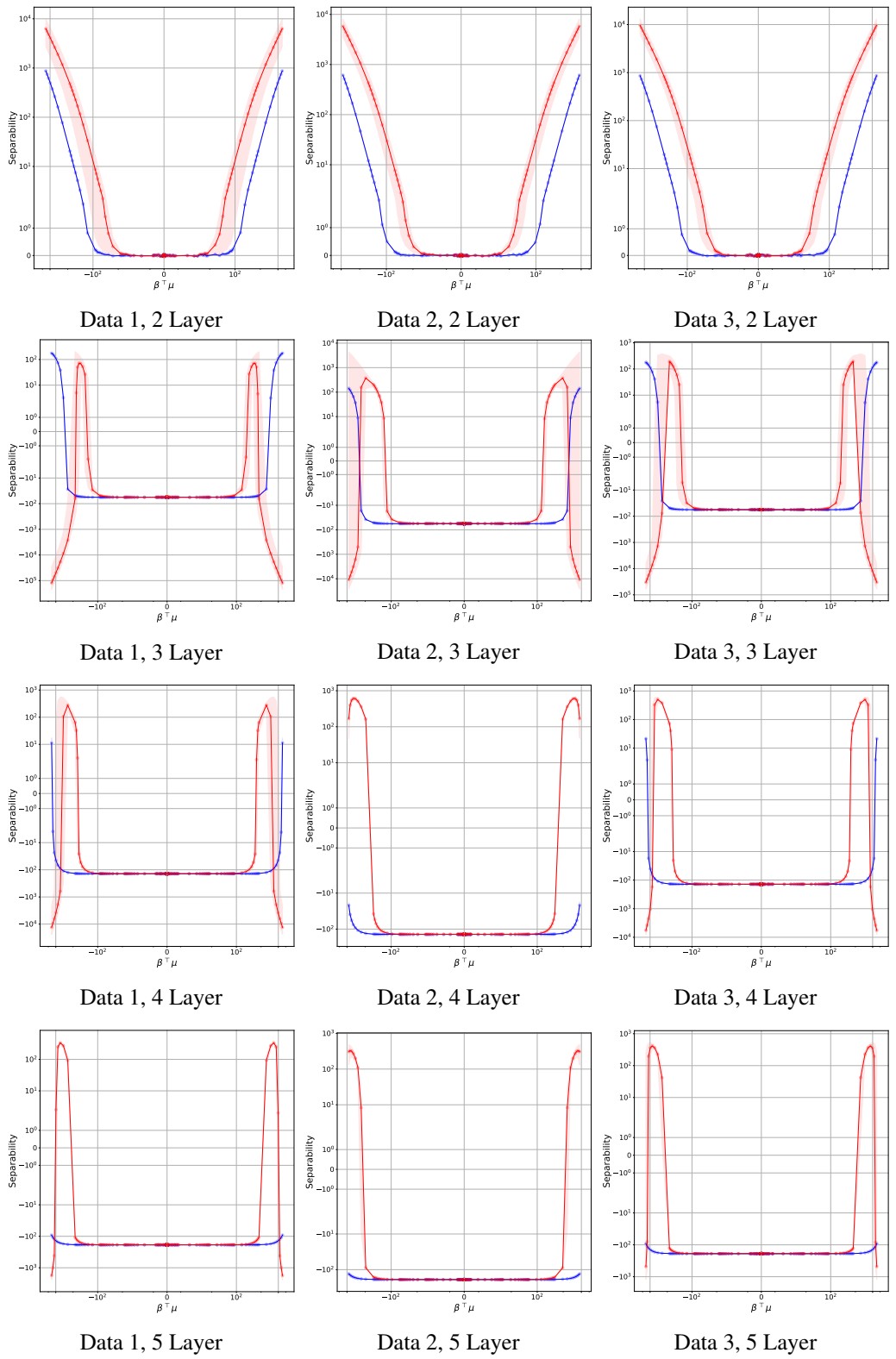

Figure 32: Evolution of *Separability* in a multi-layer network over single steps: as training progresses, *Separability* gradually strengthens. Exceptionally, for a deeper network, there are cases where *Separability* decreases if the *train-unseen similarity* is very high.

# N Additional Observations of Real-world dataset Experiments

## N.1 Additional Results

Additional Results for Expr. IV is in Figure 33 and Figure 34. Figure 33 presents experimental results using pretrained models with *Domain* datasets (CAR, CUB, SOP, ISC). Figure 34 shows results using the *Domain+sub In1k* dataset across both initialized and pretrained models. In both cases, consistent with the results in Figure 6 of the main text, models trained on a specific domain exhibit superior performance on test data from the same domain.

Additional Results for Expr. V is in Figure 35. Figure 35 includes experimental results for initialized ResNet18, pretrained ResNet18, and pretrained ResNet50, which are not discussed in the main text. Across all settings, *Domain+subsampled whole In1k* does not outperform *Domain* or *Domain+sub In1k*, despite having access to a larger number of classes (as shown in Figure 9).

Additional Results for Expr. VI is in Figure 36. We include a high-resolution version of Figure 7b and additional results using pretrained models. In most cases, increasing the number of classes leads to improved performance. We note that Table 10 shows the number of classes per step for each dataset.

Additional Results for Expr. VII is in Appendix N.2, Table 7, and Table 8.

Table 3, Table 4, Table 6, contain the original data prior to plotting.

### N.1.1 Relationship between train-unseen similarity and semantic similarity

In real world validation of our feature transfer theory, we surrogate the theoretical quantity (*train-unseen similarity*) using a semantic similarity of the dataset domain. This analogy is justified, as datapoints in each dataset (CUB, CAR, SOP, ISC) exhibit consistent visual characteristics within the dataset. Nevertheless, to strengthen our claim, we conducted the following additional experiment.

For each dataset (CUB, CAR, SOP, ISC) with three seeds, we trained a model and extracted the class-wise mean embeddings from the training set as a surrogate for $\beta$. Similarly, we computed class-wise mean embeddings from the test sets as a surrogate for $\mu$. Then, for each test class, we computed the maximum cosine similarity with any of the train class means as Listing 1.

As shown in Figures 37, 38, 39 and 40 for estimated kernel density, and Table 2 for statistics, the maximum cosine similarity between unseen and train class means tends to be highest when the train and test domains match. This supports our interpretation that semantic similarity across domains serves as a valid proxy for theoretical train-unseen alignment.

### N.1.2 Relationship between clustering criteria and Recall@1

To further validate the use of *Recall@1* as a surrogate for clustering criteria, we evaluated *Separability* as defined in Definition Definition 4.3 for Expr. IV. Following standard practice [Zhai and Wu, 2019], all models were trained with unit-normalized features, and thus we report only *Separability*. Please refer Table 5. The results indicate that models achieve the best *Separability* of test data when trained on the corresponding domain (e.g., CAR test dataset trained on CAR training dataset), consistent with the trends observed in *Recall@1*, thereby supporting the validity of *Recall@1* as a surrogate.

## N.2 Expr. VII: Removing Duplicately *Assigned* Eval Classes

In **Expr. VII**, as suggested by the theoretical results on *Separability*, we validated whether eliminating duplicate in the *assignments* improves performance. To clarify, we will provide an example of duplicate *assignment* at Note N.1.

*Note* N.1 (Example of duplicate *assignment*). For two train classes $c_1^{(train)}, c_2^{(train)}$ and two test classes $c_1^{(test)}, c_2^{(test)}$, if most instances of $c_1^{(test)}$ and $c_2^{(test)}$ are classified as $c_1^{(train)}$, both test classes are assigned to $c_1^{(train)}$, resulting in duplication. Conversely, if $c_1^{(test)}$ is classified as $c_2^{(train)}$ and $c_2^{(test)}$ as $c_1^{(train)}$, they are assigned without duplication.

To validate, we introduce treatment and control groups. For the treatment group, we eliminate duplicate in the *assignments* for the train classes, i.e., for each unseen class, the most frequently

classified training class is aggregated, and the classes are randomly removed to ensure that the selected training classes become unique (refer Algorithm Algorithm 2). For the control group, we randomly selected the same number of classes as the treatment group (refer to Algorithm Algorithm 1). These two groups are evaluated using *Recall@1*. This process is repeated five times, and the average is reported. For the dataset, we used the following categories corresponding to each domain: *Domain*, *sub In1k*, *Domain+sub In1k*, and *subsampled whole In1k* for Vehicle, Bird, Product, and Clothing. In the case of the *subsampled whole In1k* dataset for this Experiment, we performed sampling by selecting 100 instances per class. The experimental results are presented in Table 7 and 8. A total of 64 experiments are conducted, of which 51 demonstrated performance improvements; the estimated success rate is 79%. There is a $1.73\% \pm 2.87$ average percentage point improvement in *Recall@1*, with a maximum improvement of 13.65 percentage point and a minimum decrease of -3.28 percentage point. These results suggest that the duplicate reduction treatment group outperforms the randomly removed group with a binomial test p-value of $9.40 \times 10^{-7}$.

Listing 1: Observing train–unseen vs. semantic similarity

```python
# same procedure for mean_embeddings_train
for test_dataset in [cub, car, sop, isc]:
    embeddings, labels = extract_feature(model, test_dataset)
    mean_embeddings_test = torch.zeros(num_unseen_classes, dim)
    for uc in range(num_unseen_classes):
        mean_embeddings_test[uc] =\
            embeddings[labels == uc].mean(dim=0).normalize()
        # get statistics, sim: (num_unseen, num_seen)
        sim = mean_embeddings_test @ mean_embeddings_train.T
        max_values = sim.max(dim=1) # (num_unseen)
        median(max_values), mean(max_values), std(max_values)
```

---

**Algorithm 1** Random Sampling

---

**Input:** Number if unseen classes $u$, number of classes $|L|$
**Output:** Sampled class set $S_{\text{random}}$
Set $S_{\text{random}} \leftarrow \text{random.sample}(\{0, 1, \ldots, u - 1\}, |L|)$
**return** $S_{\text{random}}$

---

**Algorithm 2** Duplicated *assignment* reduction sampling

---

**Input:** Model $f$, unseen data loader $\mathcal{D}$, number of train classes $C_{\text{train}}$, number of unseen classes $C_{\text{unseen}}$
**Output:** Sampled class set $S_{\text{nondup}}$
Initialize counter matrix `counter` $\leftarrow \mathbf{0}^{C_{\text{unseen}} \times C_{\text{train}}}$
**for** (`img`, `label`) in $\mathcal{D}$ **do**
   `pred` $\leftarrow f(\text{img})$                                         *Predicted class indices*
   Update counter: `counter[label, pred]` $+= 1$
**end for**
`top1_index` $\leftarrow \text{argsort}(\text{counter}, \dim = 1, \text{descending} = \text{True})[..., 0]$
`unique_label` $\leftarrow \text{unique}(\text{top1\_index})$
Initialize $S_{\text{nondup}} \leftarrow \emptyset$
**for** each label $\ell$ in `unique_label` **do**
   $I_\ell \leftarrow \{i \mid \text{top1\_index}[i] = \ell\}$             *Indices corresponding to label $\ell$*
   $i_{\text{sample}} \leftarrow \text{random.sample}(I_\ell, 1)$           *Select one random index*
   $S_{\text{nondup}} \leftarrow S_{\text{nondup}} \cup \{i_{\text{sample}}\}$
**end for**
**return** $S_{\text{nondup}}$

---

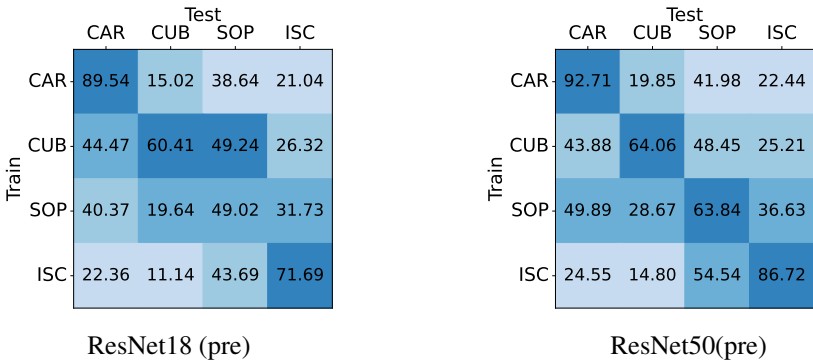

Figure 33: Expr. IV on pretrained models with *Domain* datasets (CAR, CUB, SOP, ISC)

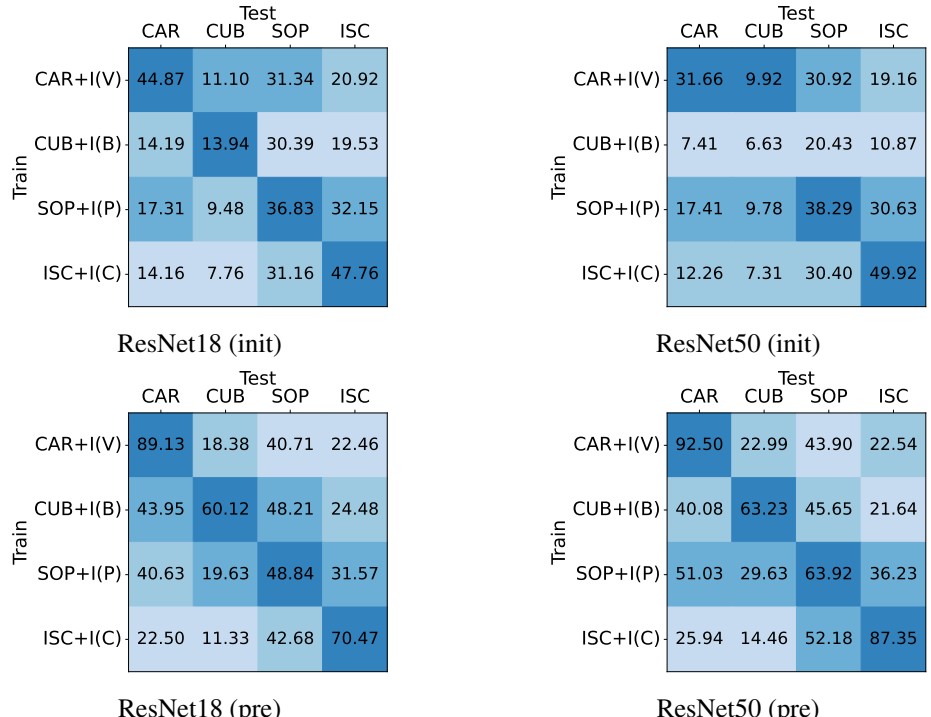

Figure 34: Expr. IV on ResNet18, ResNet50 with Domain + In(S)  e.g.  CAR+I(V), CUB+I(B), SOP+I(P), ISC+I(C)

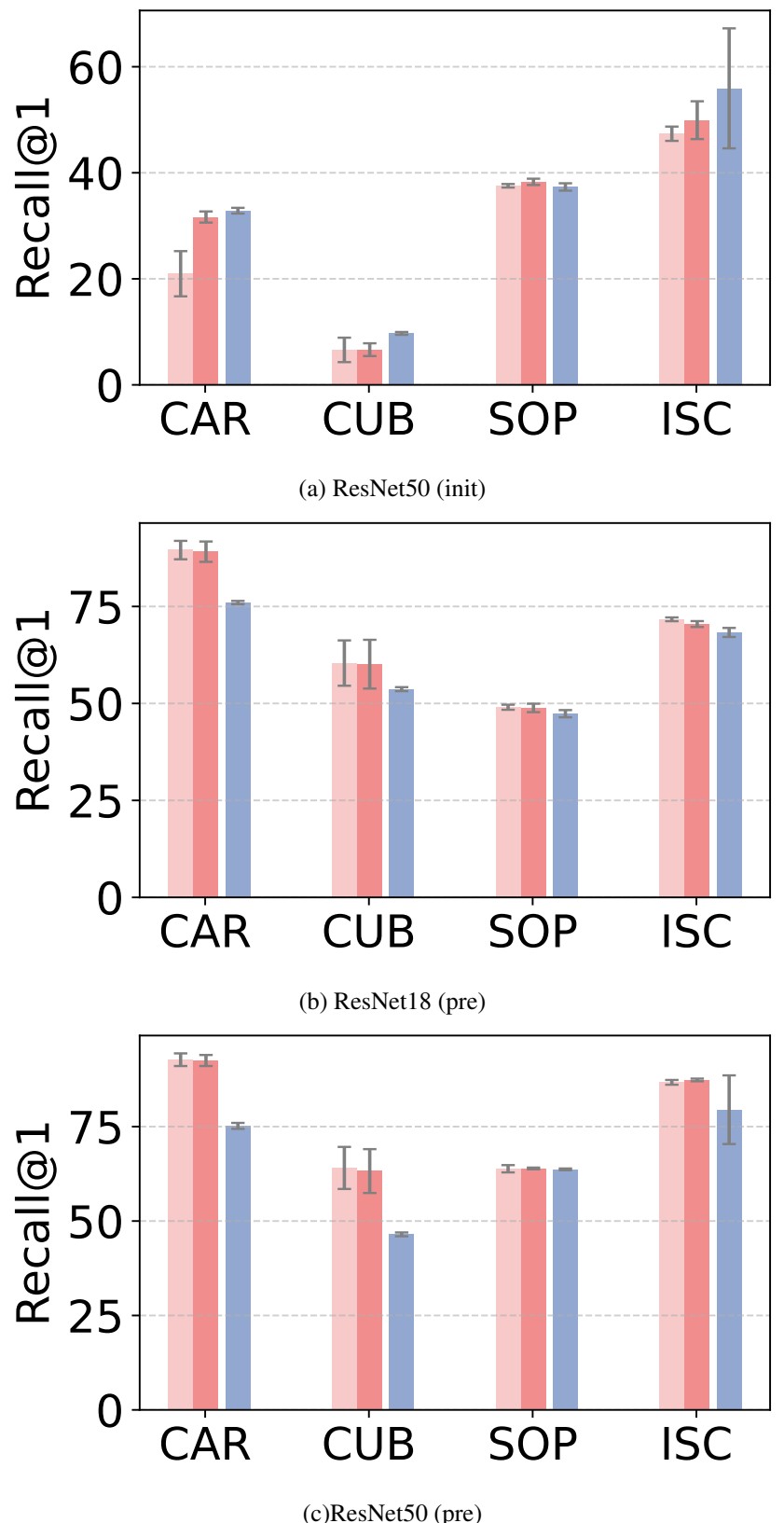

(a) ResNet50 (init)

(b) ResNet18 (pre)

(c)ResNet50 (pre)

Figure 35: Expr. V, additional results, it is represented as follows ▬ Domain ▬ Domain + Related Subset of In1k ▬ Domain + Whole In1k subsampled, Adding unrelated classes for training does not significantly affect the performance.

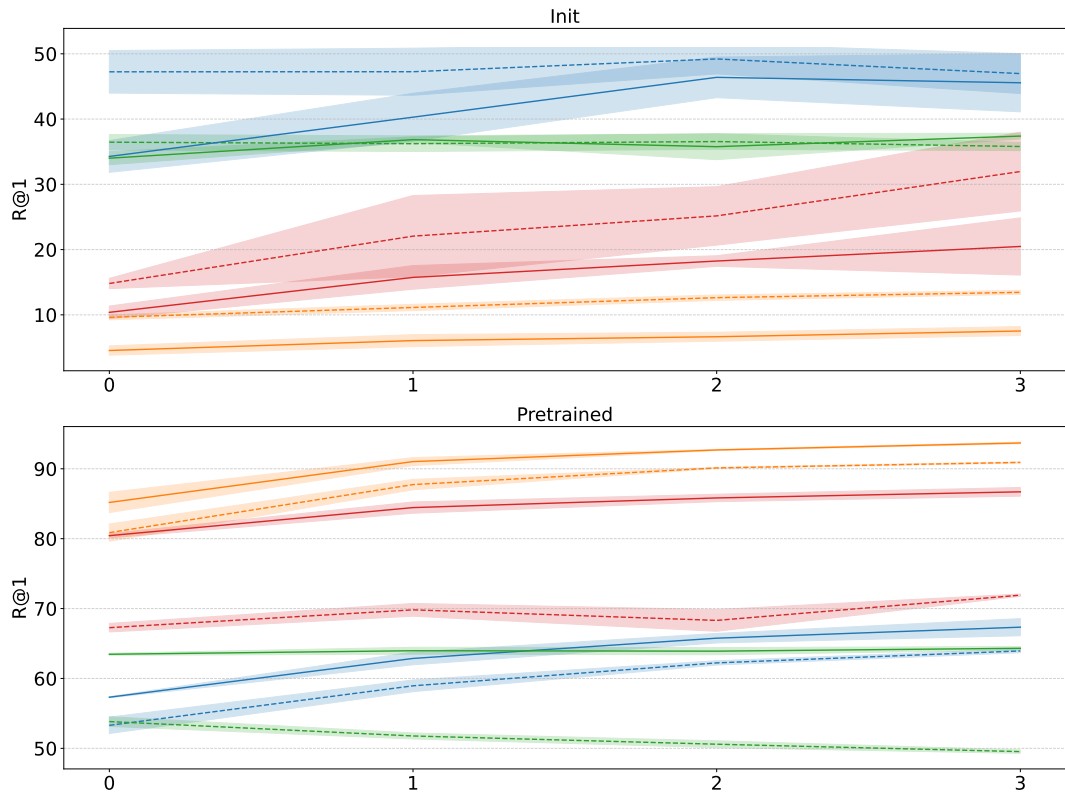

Figure 36: Expr VI, it is represented as follows: ResNet18 - - , ResNet50 —, Dataset car, cub, sop, isc. As the steps increased and related classes were added, performance generally improved consistently.

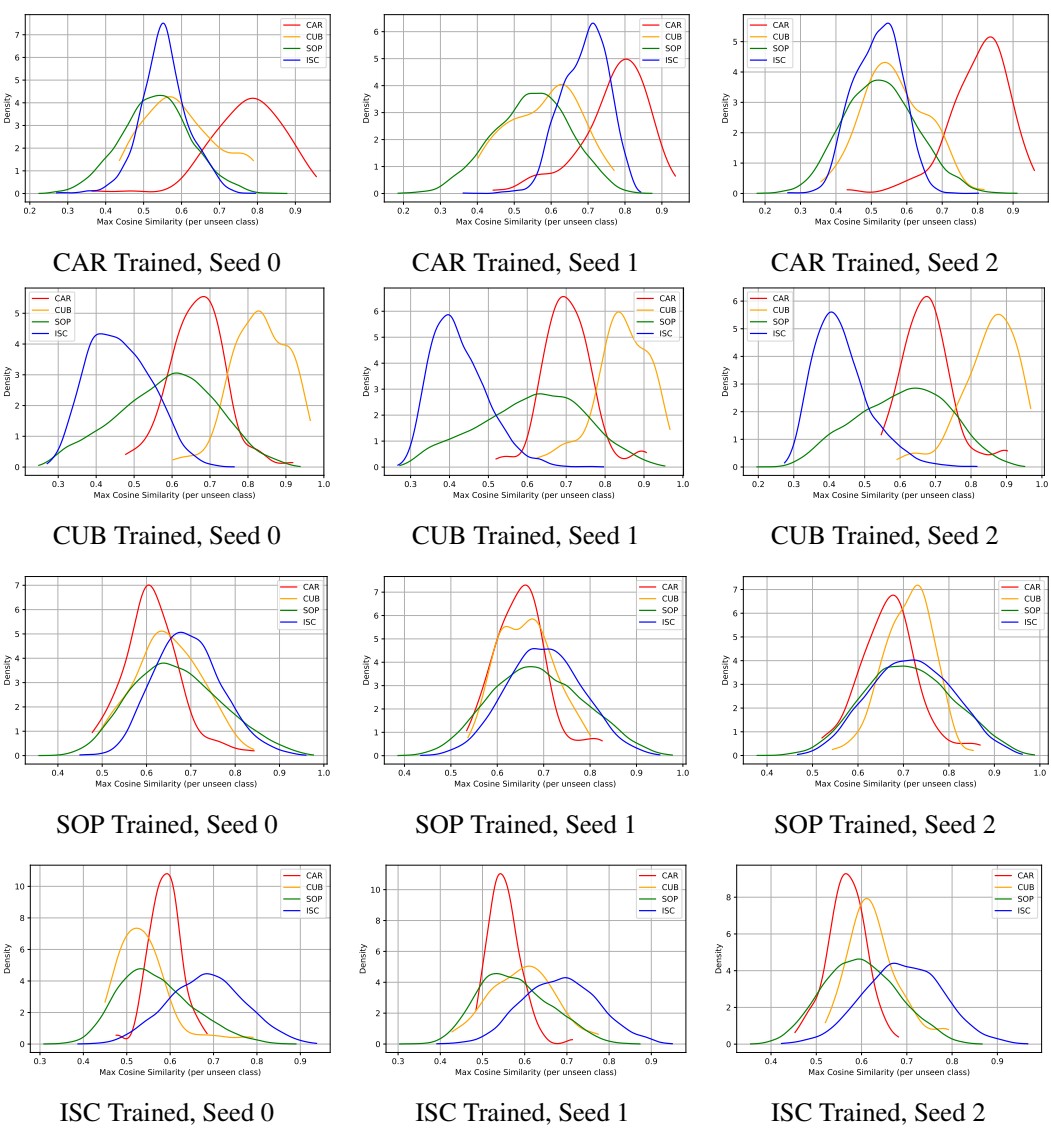

Figure 37: ResNet18 (Initialized): In most cases, the maximum cosine similarity is higher when the test dataset belongs to the same domain as the training dataset.

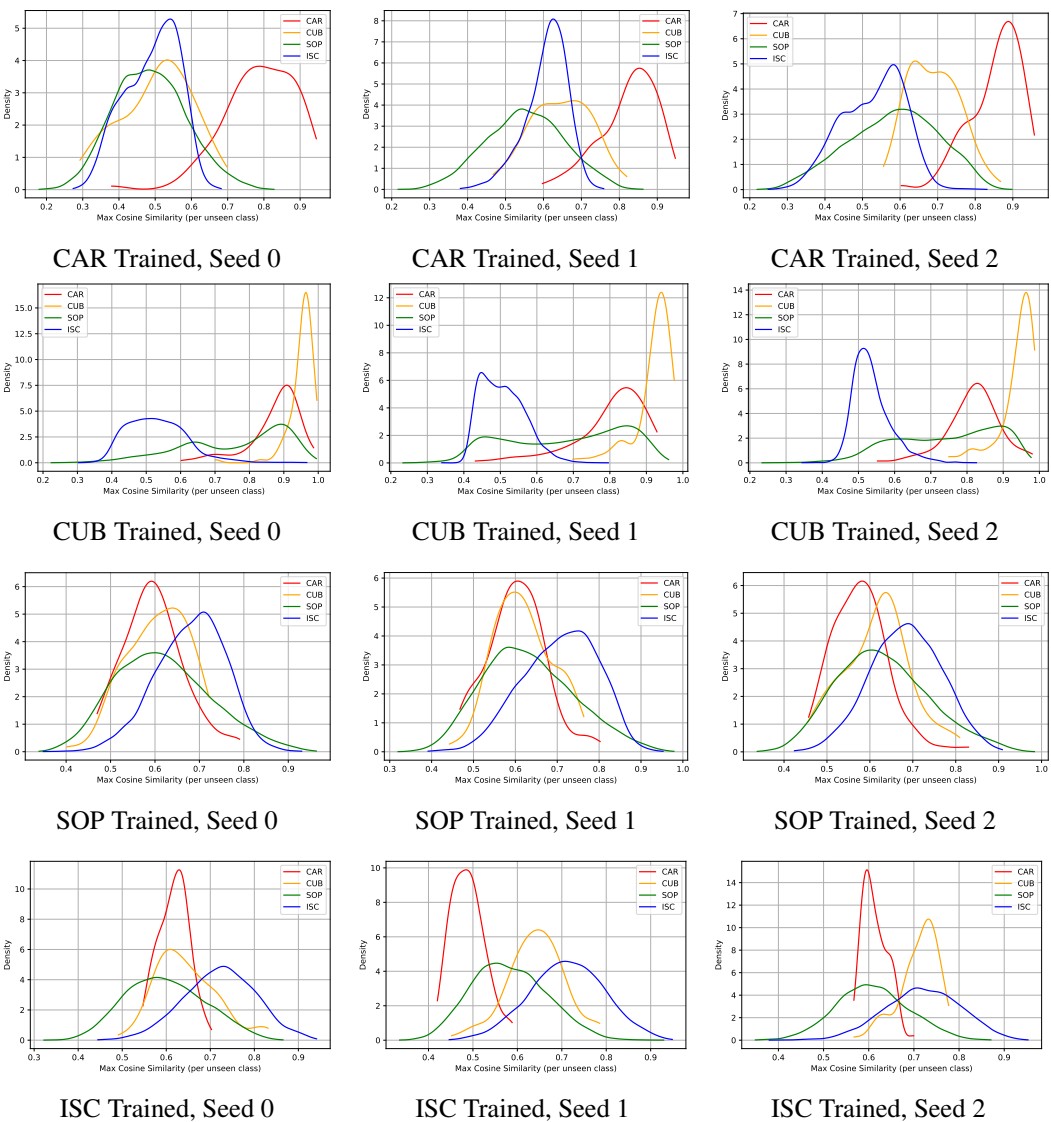

Figure 38: ResNet50 (Initialized): In most cases, the maximum cosine similarity is higher when the test dataset belongs to the same domain as the training dataset.

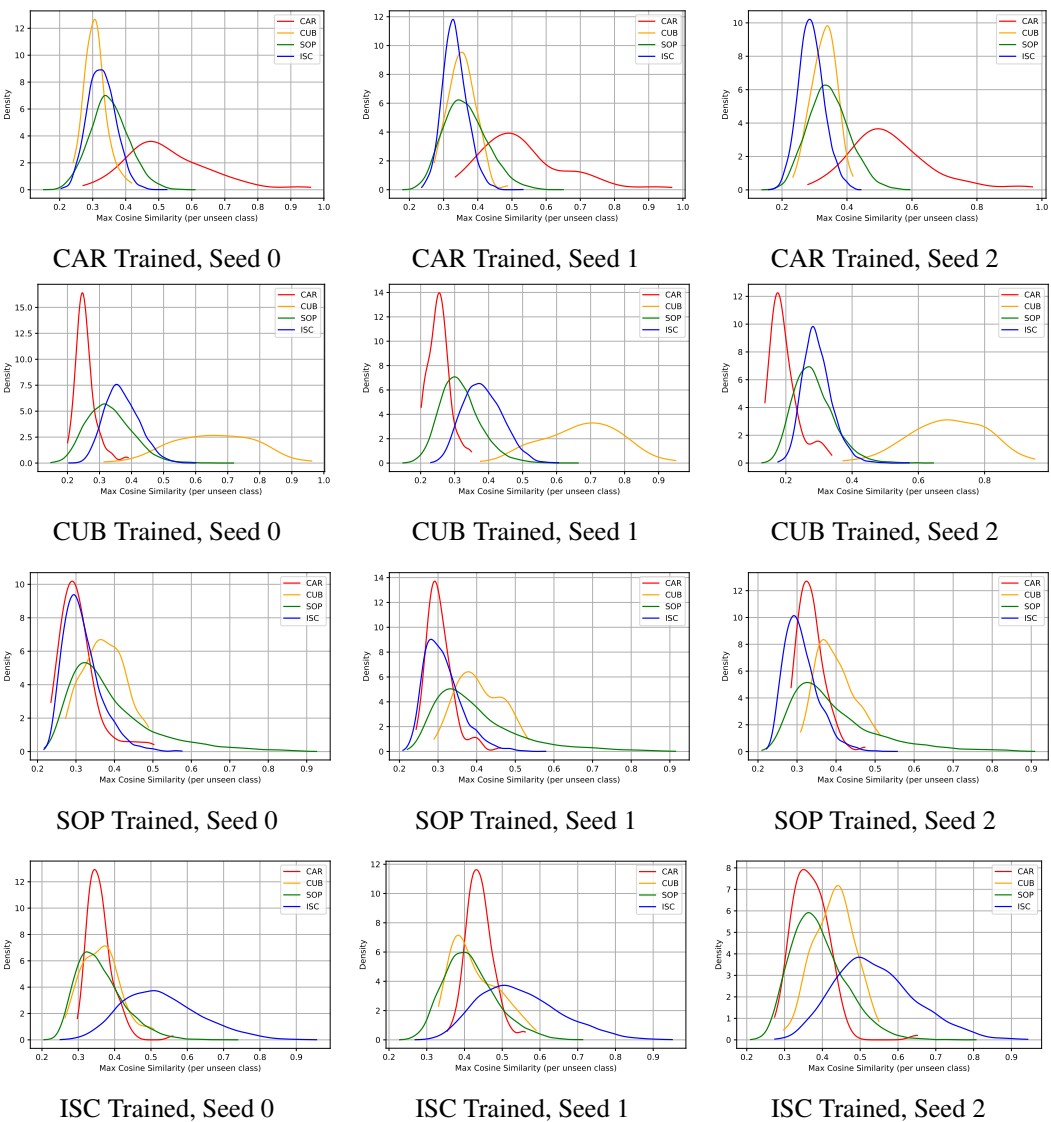

Figure 39: ResNet18 (Pretrained): In most cases, the maximum cosine similarity is higher when the test dataset belongs to the same domain as the training dataset.

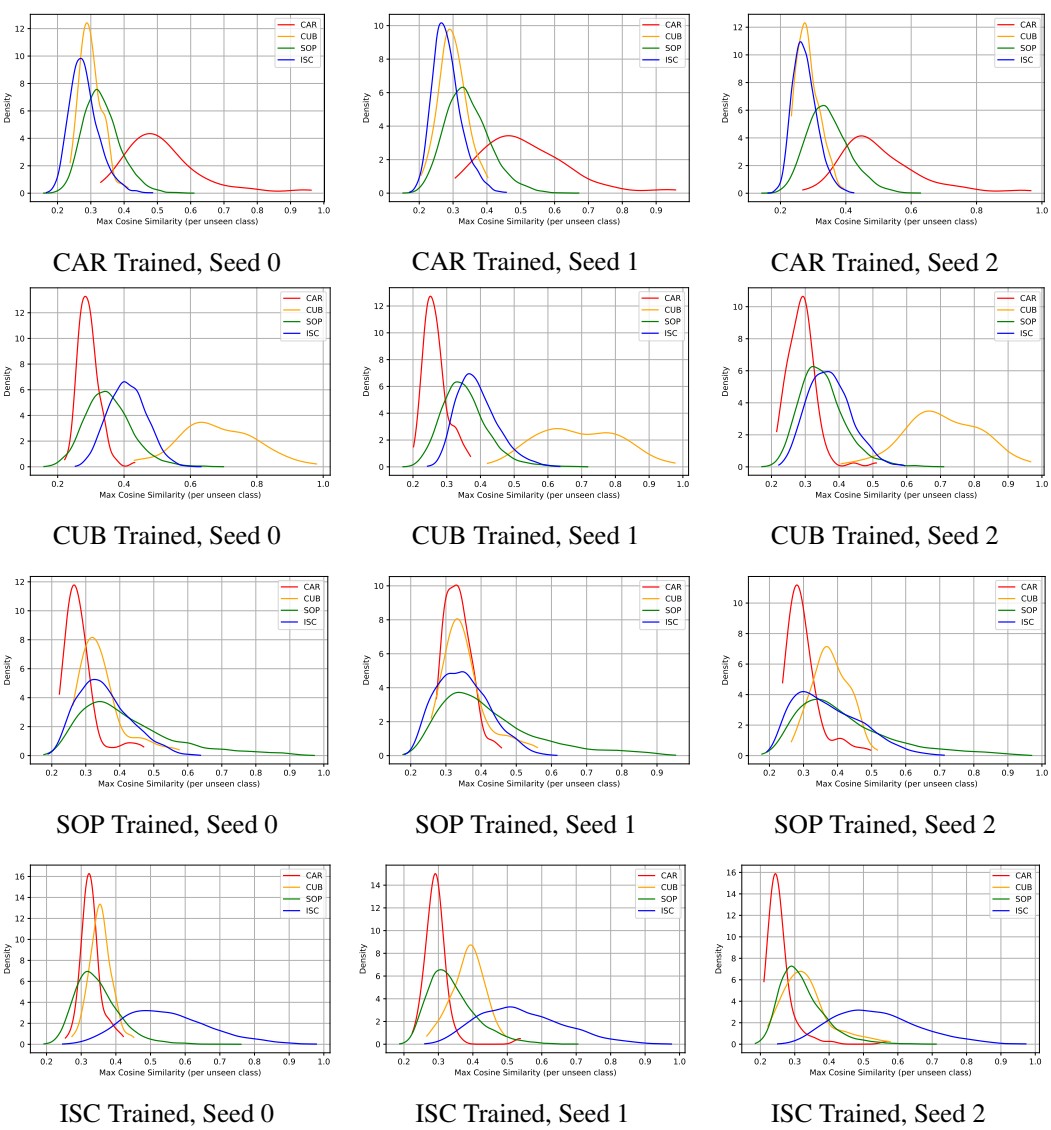

Figure 40: ResNet50 (Pretrained): In most cases, the maximum cosine similarity is higher when the test dataset belongs to the same domain as the training dataset.

Table 2: Relationship between *train–unseen similarity* and semantic similarity, with the maximum value highlighted in each row. In a model trained on a single domain, the *train–unseen similarity* is mostly highest for the test data that is most semantically similar.

### ResNet18 (Randomly Initialized)

| Train↓\Test → | CAR | | CUB | | SOP | | ISC | |
|---|---|---|---|---|---|---|---|---|
| | median | mean±std | median | mean±std | median | mean±std | median | mean±std |
| CAR | **0.80** | **0.78±0.09** | 0.58 | 0.58±0.09 | 0.53 | 0.53±0.10 | 0.59 | 0.59±0.06 |
| CUB | 0.68 | 0.68±0.07 | **0.85** | **0.84±0.07** | 0.61 | 0.60±0.13 | 0.43 | 0.44±0.08 |
| SOP | 0.64 | 0.64±0.06 | 0.67 | 0.67±0.06 | *0.68* | *0.69±0.10* | **0.70** | **0.70±0.08** |
| ISC | 0.57 | 0.57±0.04 | 0.58 | 0.59±0.07 | 0.57 | 0.58±0.08 | **0.69** | **0.69±0.09** |

### ResNet50 (Randomly Initialized)

| Train↓\Test → | CAR | | CUB | | SOP | | ISC | |
|---|---|---|---|---|---|---|---|---|
| | median | mean±std | median | mean±std | median | mean±std | median | mean±std |
| CAR | **0.83** | **0.82±0.08** | 0.62 | 0.61±0.08 | 0.54 | 0.54±0.11 | 0.55 | 0.54±0.07 |
| CUB | 0.85 | 0.83±0.08 | **0.95** | **0.94±0.05** | 0.76 | 0.73±0.15 | 0.52 | 0.52±0.07 |
| SOP | 0.59 | 0.59±0.07 | *0.62* | 0.61±0.07 | *0.62* | *0.63±0.11* | **0.70** | **0.69±0.08** |
| ISC | 0.57 | 0.57±0.03 | 0.67 | 0.67±0.06 | 0.59 | 0.60±0.08 | **0.72** | **0.72±0.08** |

### ResNet18 (ImageNet 1K Pretrained)

| Train↓\Test → | CAR | | CUB | | SOP | | ISC | |
|---|---|---|---|---|---|---|---|---|
| | median | mean±std | median | mean±std | median | mean±std | median | mean±std |
| CAR | **0.51** | **0.53±0.13** | 0.33 | 0.33±0.04 | 0.35 | 0.35±0.06 | 0.32 | 0.32±0.04 |
| CUB | 0.23 | 0.23±0.04 | **0.67** | **0.67±0.12** | 0.30 | 0.31±0.06 | 0.34 | 0.35±0.05 |
| SOP | 0.31 | 0.32±0.04 | **0.39** | **0.39±0.05** | *0.36* | **0.39±0.11** | 0.31 | 0.31±0.05 |
| ISC | 0.38 | 0.39±0.04 | 0.40 | 0.41±0.06 | 0.38 | 0.39±0.07 | **0.53** | **0.54±0.11** |

### ResNet50 (ImageNet 1K Pretrained)

| Train↓\Test → | CAR | | CUB | | SOP | | ISC | |
|---|---|---|---|---|---|---|---|---|
| | median | mean±std | median | mean±std | median | mean±std | median | mean±std |
| CAR | **0.48** | **0.51±0.12** | 0.29 | 0.29±0.04 | 0.33 | 0.34±0.06 | 0.27 | 0.28±0.04 |
| CUB | 0.28 | 0.28±0.04 | **0.68** | **0.69±0.12** | 0.35 | 0.35±0.07 | 0.39 | 0.39±0.06 |
| SOP | 0.30 | 0.31±0.05 | 0.35 | 0.36±0.06 | **0.39** | **0.42±0.14** | 0.35 | 0.36±0.08 |
| ISC | 0.29 | 0.29±0.04 | 0.35 | 0.36±0.05 | 0.32 | 0.34±0.07 | **0.53** | **0.54±0.12** |

Table 3: Table results for Expr. IV. In all cases, training with the corresponding domain data resulted in the best test performance.

ResNet18 (Randomly Initialized)

| Train↓\Test → | CAR | CUB | SOP | ISC |
|---|---|---|---|---|
| CAR | 26.9709±2.89 | 8.9242±1.85 | 27.6459±0.18 | 19.7552±0.17 |
| CAR+I(V) | **44.8653±5.00** | 11.1017±2.28 | 31.3428±0.09 | 20.9156±0.67 |
| CUB | 14.1270±0.17 | 12.5366±1.45 | 29.9312±1.62 | 18.6831±1.18 |
| CUB+I(B) | 14.1885±0.62 | **13.9433±1.76** | 30.3902±0.68 | 19.5329±1.00 |
| SOP | 17.5050±1.49 | 9.8132±0.93 | **36.6247±1.22** | 30.8448±2.27 |
| SOP+I(P) | 17.3082±1.02 | 9.4812±1.73 | 36.8302±1.28 | 32.1444±1.79 |
| ISC | 14.1680±1.43 | 8.3446±1.89 | 31.3990±2.66 | **47.8817±3.09** |
| ISC+I(C) | 14.1557±1.20 | 7.7538±1.43 | 31.1632±1.60 | 47.7550±3.13 |

ResNet50 (Randomly Initialized)

| Train↓\Test → | CAR | CUB | SOP | ISC |
|---|---|---|---|---|
| CAR | 20.9609±4.26 | 6.7466±1.63 | 25.9485±1.99 | 14.8615±3.33 |
| CAR+I(V) | **31.6566±1.05** | 9.9201±1.11 | 30.9257±1.21 | 19.1651±1.04 |
| CUB | 7.3341±1.14 | 6.5946±2.31 | 19.2065±0.64 | 10.0050±0.93 |
| CUB+I(B) | 7.4120±0.12 | 6.6340±1.21 | 20.4285±2.26 | 10.8672±2.83 |
| SOP | 18.1732±1.03 | **9.8019±2.07** | 37.5448±0.33 | 29.7677±1.96 |
| SOP+I(P) | 17.4107±0.69 | 9.7738±1.38 | **38.2880±0.59** | 30.6237±1.23 |
| ISC | 11.7739±1.36 | 6.7578±1.50 | 28.6343±1.50 | 47.3624±1.34 |
| ISC+I(C) | 12.2576±1.20 | 7.3093±1.61 | 30.3984±0.97 | **49.9217±3.57** |

Table 4: Table results for Expr. IV. In the most cases, training with the corresponding domain data resulted in the best test performance.

ResNet18 (ImageNet 1K Pretrained)

| Train↓\Test → | CAR | CUB | SOP | ISC |
|---|---|---|---|---|
| CAR | **89.5421±2.37** | 15.0180±4.31 | 38.6362±0.96 | 21.0411±1.55 |
| CAR+I(V) | 89.1280±2.62 | 18.3772±6.12 | 40.7116±0.70 | 22.4612±1.42 |
| CUB | 44.4636±8.42 | **60.4096±5.84** | 49.2369±0.46 | 26.3188±1.09 |
| CUB+I(B) | 43.9511±8.04 | 60.1227±6.29 | 48.2089±0.66 | 24.4776±2.55 |
| SOP | 40.3640±2.06 | 19.6433±4.03 | 49.0204±0.65 | 31.7294±0.78 |
| SOP+I(P) | 40.6346±0.82 | 19.6264±3.60 | 48.8414±1.09 | 31.5741±2.91 |
| ISC | 22.3589±0.96 | 11.1467±2.37 | 43.6862±1.69 | **71.6921±0.49** |
| ISC+I(C) | 22.5065±2.90 | 11.3324±1.42 | 42.6812±3.22 | 70.4721±0.75 |

ResNet50 (ImageNet 1K Pretrained)

| Train↓\Test → | CAR | CUB | SOP | ISC |
|---|---|---|---|---|
| CAR | **92.7069±1.68** | 19.8458±8.36 | 41.9793±2.20 | 22.4363±4.95 |
| CAR+I(V) | 92.5019±1.46 | 22.9856±8.93 | 43.8988±1.74 | 22.5407±4.13 |
| CUB | 43.8814±11.40 | **64.0558±5.57** | 48.4447±0.66 | 25.2143±2.87 |
| CUB+I(B) | 40.0853±10.21 | 63.2287±5.81 | 45.6481±0.15 | 21.6412±2.57 |
| SOP | 49.8955±2.81 | 28.6743±5.80 | 63.8398±0.97 | 36.6294±0.78 |
| SOP+I(P) | 51.0269±3.89 | 29.6309±6.06 | **63.9213±0.23** | 36.2244±0.76 |
| ISC | 24.5480±2.59 | 14.8042±1.82 | 54.5414±0.71 | 86.7164±0.63 |
| ISC+I(C) | 25.9419±2.90 | 14.4609±2.10 | 52.1762±1.38 | **87.3562±0.35** |

Table 5: Table of Expr. IV Results with *Separability* Measurements. In all cases, training with the corresponding domain data resulted in the best *Separability*.

ResNet18 (Randomly Initialized)

| Train↓\Test → | CAR | CUB | SOP | ISC |
|---|---|---|---|---|
| CAR | **-0.25±0.12** | -0.63±0.12 | -0.29±0.10 | -0.82±0.07 |
| CUB | -0.75±0.12 | **-0.28±0.19** | -0.14±0.11 | -0.81±0.06 |
| SOP | -0.69±0.13 | -0.42±0.13 | **-0.01±0.01** | -0.47±0.09 |
| ISC | -0.86±0.08 | -0.79±0.08 | -0.28±0.10 | **-0.01±0.01** |

ResNet50 (Randomly Initialized)

| Train↓\Test → | CAR | CUB | SOP | ISC |
|---|---|---|---|---|
| CAR | **-0.28±0.12** | -0.72±0.13 | -0.29±0.10 | -0.81±0.06 |
| CUB | -0.71±0.15 | **-0.36±0.27** | -0.10±0.11 | -0.88±0.06 |
| SOP | -0.67±0.12 | -0.37±0.12 | **-0.01±0.01** | -0.46±0.09 |
| ISC | -0.90±0.05 | -0.83±0.07 | -0.30±0.10 | **-0.01±0.01** |

ResNet18 (ImageNet 1K Pretrained)

| Train↓\Test → | CAR | CUB | SOP | ISC |
|---|---|---|---|---|
| CAR | **-0.09±0.05** | -0.85±0.04 | -0.36±0.06 | -0.76±0.05 |
| CUB | -0.84±0.04 | **-0.21±0.13** | -0.37±0.05 | -0.67±0.06 |
| SOP | -0.64±0.08 | -0.47±0.08 | **-0.01±0.01** | -0.28±0.06 |
| ISC | -0.82±0.09 | -0.75±0.07 | -0.22±0.08 | **-0.01±0.01** |

ResNet50 (ImageNet 1K Pretrained)

| Train↓\Test → | CAR | CUB | SOP | ISC |
|---|---|---|---|---|
| CAR | **-0.10±0.05** | -0.83±0.04 | -0.34±0.06 | -0.68±0.05 |
| CUB | -0.85±0.04 | **-0.28±0.12** | -0.39±0.06 | -0.73±0.05 |
| SOP | -0.71±0.09 | -0.54±0.08 | **-0.01±0.00** | -0.40±0.08 |
| ISC | -0.86±0.08 | -0.77±0.05 | -0.25±0.07 | **-0.01±0.00** |

Table 6: Table results of performance for Expr. V. The performance of D and D+I(Sub) is comparable to that of D+I, despite D+I having a larger number of classes.

ResNet18 (Randomly Initialized)

| Train↓\Test → | CAR | CUB | SOP | ISC |
|---|---|---|---|---|
| D | 26.97±2.89 | 12.54±1.45 | 36.62±1.22 | **47.88±3.09** |
| D+I(Sub) | **44.87±5.00** | 13.94±1.76 | **36.83±1.28** | 47.76±3.13 |
| D+I | 31.61±1.62 | **14.11±0.10** | 36.62±0.88 | 44.82±0.55 |

ResNet50 (Randomly Initialized)

| Train↓\Test → | CAR | CUB | SOP | ISC |
|---|---|---|---|---|
| D | 20.96±4.26 | 6.59±2.31 | 37.54±0.33 | 47.36±1.34 |
| D+I(Sub) | **31.66±1.05** | 6.63±1.21 | **38.29±0.59** | 49.92±3.57 |
| D+I | 32.85±0.53 | **9.70±0.26** | 37.32±0.69 | **55.92±11.32** |

ResNet18 (ImageNet 1K Pretrained)

| Train↓\Test → | CAR | CUB | SOP | ISC |
|---|---|---|---|---|
| D | **89.54±2.37** | **60.41±5.84** | **49.02±0.65** | **71.69±0.48** |
| D+I(Sub) | 89.13±2.62 | 60.12±6.29 | 48.84±1.09 | 70.47±0.75 |
| D+I | 76.02±0.42 | 53.69±0.52 | 47.35±0.93 | 68.30±1.17 |

ResNet50 (ImageNet 1K Pretrained)

| Train↓\Test → | CAR | CUB | SOP | ISC |
|---|---|---|---|---|
| D | **92.71±1.68** | **64.06±5.57** | 63.84±0.97 | 86.72±0.63 |
| D+I(Sub) | 92.50±1.46 | 63.23±5.81 | **63.92±0.23** | **87.36±0.35** |
| D+I | 75.19±0.79 | 46.48±0.51 | 63.68±0.23 | 79.47±9.09 |

Table 7: Expr. VII from (Randomly Initialized). Column Original indicates the Recall@1 performance of the original trained model when neither treatment nor random removal is applied. Column $\Delta$ indicates the difference between Treatement and Random Recall@1 score.

ResNet18 (Randomly Initialized)

| Test | Train | Treatment | Random | $\Delta$ | Original |
|------|-------|-----------|--------|----------|----------|
| CAR Test | CAR | $42.57 \pm 1.59$ | $40.42 \pm 2.23$ | $2.14 \pm 2.74$ | 23.83 |
| | I(V) | $32.96 \pm 1.23$ | $27.93 \pm 3.98$ | $5.02 \pm 4.17$ | 11.17 |
| | CAR+I(V) | $56.16 \pm 1.60$ | $56.67 \pm 1.21$ | $-0.51 \pm 2.00$ | 39.22 |
| | In | $41.43 \pm 0.88$ | $38.65 \pm 2.42$ | $2.77 \pm 2.57$ | 25.70 |
| CUB Test | CUB | $24.56 \pm 1.22$ | $21.46 \pm 2.57$ | $3.10 \pm 2.85$ | 10.89 |
| | I(B) | $21.56 \pm 2.07$ | $18.68 \pm 1.33$ | $2.88 \pm 2.46$ | 6.40 |
| | CUB+I(B) | $19.97 \pm 1.33$ | $18.93 \pm 2.09$ | $1.04 \pm 2.48$ | 12.05 |
| | In | $37.66 \pm 0.87$ | $32.92 \pm 1.53$ | $4.74 \pm 1.76$ | 21.49 |
| SOP Test | SOP | $41.63 \pm 0.22$ | $41.49 \pm 0.21$ | $0.15 \pm 0.30$ | 48.87 |
| | I(P) | $50.70 \pm 2.50$ | $49.30 \pm 2.25$ | $1.40 \pm 3.37$ | 18.23 |
| | SOP+I(P) | $44.59 \pm 0.19$ | $44.57 \pm 0.46$ | $0.02 \pm 0.50$ | 48.70 |
| | In | $52.24 \pm 0.24$ | $52.71 \pm 0.76$ | $-0.48 \pm 0.80$ | 24.75 |
| ISC Test | ISC | $63.51 \pm 0.47$ | $62.37 \pm 0.47$ | $1.14 \pm 0.67$ | 37.90 |
| | I(C) | $61.43 \pm 3.65$ | $52.54 \pm 1.84$ | $8.89 \pm 4.09$ | 31.29 |
| | ISC+I(C) | $56.81 \pm 0.59$ | $56.10 \pm 0.38$ | $0.71 \pm 0.70$ | 37.20 |
| | In | $45.25 \pm 1.59$ | $45.43 \pm 1.56$ | $-0.17 \pm 2.23$ | 38.82 |
| Average Improvement | | | | 2.05 | |
| Success Rate | | | | 0.8125 | |

ResNet50 (Randomly Initialized)

| Test | Train | Treatment | Random | $\Delta$ | Original |
|------|-------|-----------|--------|----------|----------|
| CAR Test | CAR | $33.62 \pm 1.89$ | $32.66 \pm 1.01$ | $0.97 \pm 2.15$ | 20.67 |
| | I(V) | $24.07 \pm 1.29$ | $1.30 \pm 0.97$ | $2.77 \pm 1.61$ | 10.48 |
| | CAR+I(V) | $48.11 \pm 1.36$ | $46.02 \pm 1.96$ | $2.09 \pm 2.39$ | 32.80 |
| | In | $53.52 \pm 0.98$ | $46.75 \pm 2.41$ | $6.77 \pm 2.60$ | 30.06 |
| CUB Test | CUB | $25.10 \pm 1.90$ | $20.28 \pm 2.52$ | $4.82 \pm 3.16$ | 3.93 |
| | I(B) | $26.96 \pm 1.40$ | $22.31 \pm 2.20$ | $4.64 \pm 2.61$ | 3.58 |
| | CUB+I(B) | $17.43 \pm 1.22$ | $14.76 \pm 1.60$ | $2.67 \pm 2.02$ | 5.27 |
| | In | $47.89 \pm 1.15$ | $39.48 \pm 1.28$ | $8.42 \pm 1.72$ | 28.06 |
| SOP Test | SOP | $45.53 \pm 0.42$ | $45.70 \pm 0.51$ | $-0.17 \pm 0.66$ | 45.81 |
| | I(P) | $45.02 \pm 2.24$ | $45.55 \pm 2.15$ | $-0.53 \pm 3.10$ | 14.46 |
| | SOP+I(P) | $44.33 \pm 0.36$ | $43.96 \pm 0.50$ | $0.37 \pm 0.61$ | 53.18 |
| | In | $59.42 \pm 0.85$ | $58.11 \pm 0.76$ | $1.31 \pm 1.14$ | 22.85 |
| ISC Test | ISC | $64.90 \pm 0.46$ | $62.92 \pm 0.57$ | $1.98 \pm 0.73$ | 37.50 |
| | I(C) | $55.21 \pm 1.52$ | $50.42 \pm 4.59$ | $4.78 \pm 4.84$ | 27.16 |
| | ISC+I(C) | $65.22 \pm 0.19$ | $64.72 \pm 0.66$ | $0.50 \pm 0.69$ | 38.12 |
| | In | $44.00 \pm 0.91$ | $43.56 \pm 1.12$ | $-0.44 \pm 1.44$ | 42.93 |
| Average Improvement | | | | 2.56 | |
| Success Rate | | | | 0.8125 | |

Table 8: Expr. VII (ImageNet 1K Pretrained). Column Original indicates the Recall@1 performance of the original trained model when neither treatment nor random removal is applied. Column $\Delta$ indicates the difference between Treatement and Random Recall@1 score.

ResNet18 (ImageNet 1K Pretrained)

| Test | Train | Treatment | Random | $\Delta$ | Original |
|---|---|---|---|---|---|
| CAR Test | CAR | 90.30 ± 0.72 | 90.10 ± 0.55 | 0.20 ± 0.90 | 86.80 |
| | I(V) | 64.40 ± 3.23 | 65.51 ± 2.93 | -1.12 ± 4.37 | 42.10 |
| | CAR+I(V) | 91.37 ± 0.52 | 90.38 ± 0.63 | 1.00 ± 0.82 | 86.10 |
| | In | 74.96 ± 5.14 | 66.56 ± 4.02 | 8.40 ± 6.52 | 26.00 |
| CUB Test | CUB | 62.86 ± 1.53 | 63.36 ± 2.56 | -0.50 ± 2.98 | 53.66 |
| | I(B) | 61.21 ± 2.40 | 54.89 ± 1.32 | 6.32 ± 2.74 | 34.00 |
| | CUB+I(B) | 71.03 ± 0.86 | 67.02 ± 1.08 | 4.01 ± 1.38 | 52.89 |
| | In | 46.29 ± 1.05 | 40.73 ± 3.47 | 5.57 ± 3.63 | 30.32 |
| SOP Test | SOP | 51.65 ± 0.17 | 50.88 ± 0.43 | 0.77 ± 0.46 | 71.15 |
| | I(P) | 69.91 ± 0.79 | 68.17 ± 0.93 | 1.74 ± 1.22 | 24.57 |
| | SOP+I(P) | 52.58 ± 0.18 | 51.69 ± 0.29 | 0.89 ± 0.34 | 70.98 |
| | In | 46.92 ± 0.63 | 45.41 ± 0.92 | 1.51 ± 1.12 | 13.85 |
| ISC Test | ISC | 76.63 ± 0.60 | 74.59 ± 0.62 | 2.04 ± 0.86 | 48.27 |
| | I(C) | 71.72 ± 5.58 | 71.28 ± 4.24 | 0.45 ± 7.01 | 48.38 |
| | ISC+I(C) | 78.21 ± 0.27 | 77.53 ± 0.49 | 0.69 ± 0.56 | 47.75 |
| | In | 30.21 ± 1.13 | 36.25 ± 4.05 | -6.04 ± 4.20 | 30.66 |
| Average Improvement | | | | 1.62 | |
| Success Rate | | | | 0.8125 | |

ResNet50 (ImageNet 1K Pretrained)

| Test | Train | Treatment | Random | $\Delta$ | Original |
|---|---|---|---|---|---|
| CAR Test | CAR | 95.32 ± 0.28 | 95.58 ± 0.70 | -0.26 ± 0.76 | 90.77 |
| | I(V) | 75.37 ± 3.98 | 72.97 ± 3.39 | 2.39 ± 5.23 | 40.13 |
| | CAR+I(V) | 94.85 ± 0.93 | 94.82 ± 0.59 | 0.03 ± 1.10 | 90.81 |
| | In | 80.45 ± 2.26 | 68.57 ± 5.35 | 11.88 ± 5.80 | 32.51 |
| CUB Test | CUB | 80.00 ± 1.45 | 76.05 ± 1.96 | 3.96 ± 2.44 | 57.78 |
| | I(B) | 60.53 ± 2.72 | 58.16 ± 1.85 | 2.37 ± 3.30 | 33.37 |
| | CUB+I(B) | 77.54 ± 0.54 | 72.83 ± 1.03 | 4.70 ± 1.16 | 56.56 |
| | In | 69.53 ± 2.10 | 56.82 ± 3.50 | 12.71 ± 4.08 | 35.53 |
| SOP Test | SOP | 70.10 ± 0.20 | 69.59 ± 0.23 | 0.51 ± 0.31 | 87.10 |
| | I(P) | 68.04 ± 0.92 | 65.75 ± 1.32 | 2.29 ± 1.61 | 24.13 |
| | SOP+I(P) | 68.61 ± 0.09 | 68.06 ± 0.37 | 0.55 ± 0.38 | 87.18 |
| | In | 55.81 ± 0.61 | 51.94 ± 1.44 | 3.87 ± 1.56 | 8.68 |
| ISC Test | ISC | 90.70 ± 0.13 | 89.85 ± 0.41 | 0.85 ± 0.43 | 62.75 |
| | I(C) | 66.42 ± 4.51 | 67.06 ± 3.79 | -0.64 ± 5.89 | 64.02 |
| | ISC+I(C) | 90.85 ± 0.35 | 89.98 ± 0.47 | 0.87 ± 0.59 | 63.66 |
| | In | 43.15 ± 3.53 | 40.84 ± 6.19 | 2.31 ± 7.12 | 28.87 |
| Average Improvement | | | | 3.02 | |
| Success Rate | | | | 0.875 | |

# O    Additional Information of ImageNet subset used in Experiments

Table 9: Configuration of Expr. V

|          | Vehicle | Bird | Product | Clothing |
|----------|---------|------|---------|----------|
| D        | 98      | 100  | 11316   | 3985     |
| D+I(Sub) | 138     | 159  | 11568   | 4031     |
| D+I      | 1098    | 1100 | 12316   | 4985     |

Table 10: Configuration of Expr. VI

|          | Step 0 | Step 1 | Step 2 | Step 3 |
|----------|--------|--------|--------|--------|
| Vehicle  | 25     | 50     | 75     | 98     |
| Bird     | 25     | 50     | 75     | 100    |
| Product  | 2829   | 5658   | 8487   | 11316  |
| Clothing | 996    | 1992   | 2989   | 3985   |

This section provides detailed information on the number of classes in the training datasets used in Expr. V and VI (9, 10), along with a list of manually selected classes that constitute the ImageNet subsets related to Vehicle, Bird, Product, and Clothing.

## O.1    I(V): The Vehicle classes chosen in ImageNet

Total 40 classes.

ambulance, cab, convertible, fire engine, forklift, freight car, garbage truck, go-kart, golfcart, half track, harvester, horse cart, jeep, jinrikisha, limousine, minibus, minivan, Model T, moped, motor scooter, mountain bike, moving van, oxcart, passenger car, pickup, police van, racer, recreational vehicle, school bus, snowmobile, snowplow, sports car, streetcar, tank, tow truck, tractor, trailer truck, tricycle, trolleybus, unicycle

## O.2    I(B): The Bird classes chosen in ImageNet

Total 59 classes.

cock, hen, ostrich, brambling, goldfinch, house finch, junco, indigo bunting, robin, bulbul, jay, magpie, chickadee, water ouzel, bald eagle, vulture, great grey owl, black grouse, ptarmigan, ruffed grouse, prairie chicken, peacock, quail, partridge, African grey, macaw, sulphur-crested cockatoo, lorikeet, coucal, bee eater, hornbill, hummingbird, jacamar, toucan, drake, red-breasted merganser, goose, black swan, tusker, white stork, black stork, spoonbill, flamingo, little blue heron, American egret, bittern, crane, limpkin, European gallinule, American coot, bustard, ruddy turnstone, red-backed sandpiper, redshank, dowitcher, oystercatcher, pelican, king penguin, albatross

## O.3    I(P): The Product classes chosen in ImageNet

Total 353 classes.

abacus, accordion, acoustic guitar, altar, analog clock, apiary, ashcan, assault rifle, backpack, balance beam, balloon, ballpoint, Band Aid, banjo, barbell, barber chair, barometer, barrel, barrow, baseball, basketball, bassinet, bassoon, bathing cap, bath towel, bathtub, beach wagon, beacon, beaker, bearskin, beer bottle, beer glass, bell cote, bib, bicycle-built-for-two, binder, binoculars, bobsled, bolo tie, bonnet, bookcase, bottlecap, bow tie, brass, breakwater, broom, bucket, buckle, bulletproof vest, caldron, candle, cannon, canoe, can opener, car mirror, carousel, carpenter's kit, carton, car wheel, cash machine, cassette, cassette player, CD player, cello, cellular telephone, chain, chain saw, chest, chiffonier, chime, china cabinet, cleaver, clog, cocktail shaker, coffee mug, coffeepot, coil, combination lock, computer keyboard, confectionery, corkscrew, cornet, cradle, crash helmet, crate, crib, Crock Pot, croquet ball, crutch, dam, desk, desktop computer, dial telephone, digital clock, digital watch, dining table, dishrag, dishwasher, disk brake, dogsled, doormat, drum, drumstick, dumbbell, Dutch oven, electric fan, electric guitar, electric locomotive, envelope, espresso maker, face powder, feather boa, file, fire screen, flagpole, flute, folding chair, football helmet, fountain pen, four-poster, French horn, frying pan, gasmask, gas pump, goblet, golf ball, gondola, gong, grand piano, grille, guillotine, hair slide, hair spray, hammer, hamper, hand blower, hand-held computer, handkerchief, hard disc, harmonica, harp, hatchet, holster, honeycomb, hook, horizontal bar, hourglass, iPod, iron, jack-o'-lantern, jigsaw puzzle, joystick, knot, ladle, lampshade, laptop, lawn mower, lens cap, letter opener, lighter, lipstick, lotion, loudspeaker, loupe, magnetic compass, mailbox, maraca, marimba, matchstick, maypole, measuring cup, medicine chest, microphone, microwave, milk can, mixing bowl, modem, monitor, mountain tent, mousetrap, muzzle, nail, neck brace, necklace, nipple, notebook,

oboe, ocarina, odometer, oil filter, organ, oscilloscope, oxygen mask, packet, paddle, paddlewheel, padlock, paintbrush, paper towel, parachute, parallel bars, park bench, parking meter, pay-phone, pedestal, pencil box, pencil sharpener, perfume, Petri dish, photocopier, pick, picket fence, piggy bank, pill bottle, pillow, ping-pong ball, plastic bag, plate rack, plow, plunger, Polaroid camera, pole, pool table, pop bottle, pot, potter's wheel, power drill, prayer rug, printer, prison, projectile, projector, puck, punching bag, purse, quill, quilt, racket, radiator, radio, radio telescope, rain barrel, reel, reflex camera, refrigerator, remote control, revolver, rifle, rocking chair, rotisserie, rubber eraser, rugby ball, rule, safe, safety pin, saltshaker, sax, scabbard, scale, scoreboard, screen, screw, screwdriver, seat belt, sewing machine, shield, shopping basket, shopping cart, shovel, shower cap, shower curtain, ski, sleeping bag, sliding door, slot, snorkel, soap dispenser, soccer ball, sock, solar dish, soup bowl, space bar, space heater, spatula, spider web, spindle, spotlight, steel drum, stethoscope, stole, stopwatch, stove, strainer, stretcher, studio couch, sunscreen, swab, switch, syringe, table lamp, tape player, teapot, teddy, television, tennis ball, theater curtain, thimble, thresher, throne, tile roof, toaster, tobacco shop, toilet seat, torch, totem pole, tray, tripod, trombone, tub, turnstile, typewriter keyboard, umbrella, vacuum, vase, vault, velvet, vending machine, violin, volleyball, waffle iron, wall clock, wallet, wardrobe, washbasin, washer, water bottle, water jug, water tower, whiskey jug, whistle, window screen, window shade, wine bottle, wing, wok, wooden spoon, comic book, crossword puzzle, street sign, traffic light, book jacket, menu, plate

### O.4   I(C): The Clothing classes chosen in ImageNet

Total 46 classes.

abaya, academic gown, apron, bikini, brassiere, breastplate, cardigan, chain mail, Christmas stocking, cloak, cowboy boot, cowboy hat, cuirass, diaper, fur coat, gown, hoopskirt, jean, jersey, kimono, knee pad, lab coat, Loafer, mailbag, mask, military uniform, miniskirt, mitten, overskirt, pajama, poncho, running shoe, sandal, sarong, ski mask, sombrero, suit, sunglass, sunglasses, sweatshirt, swimming trunks, trench coat, vestment, wig, Windsor tie, wool

## P   Rotation Matrix Generation Process of *Setup 2*

To generate a set of rotation matrices with diverse magnitudes of rotation, we constructed an algorithm that samples $k = 300$ random matrices, each formed by adding i.i.d. Gaussian noise matrix of varying variance to the identity matrix $I$. The process ensures the generation of rotation matrices with varying extents of rotation, from slight to more substantial deviations from the identity matrix.

The rotation matrices are generated as follows:

1. A matrix is initialized as $I + \epsilon \cdot M$, where $M$ is a i.i.d. standard random Gaussian matrix.

2. Using the QR decomposition, we orthogonalize this matrix to ensure it forms a valid rotation matrix.

3. Finally, if the determinant of the resulting matrix is negative, we flip the sign of the first column to maintain a determinant of $+1$, ensuring it is a valid rotation.

In summary, this method provides a collection of matrices that progressively deviate from $I$, allowing us to observe and sample rotations of increasing magnitude. Please refer to Algorithm Algorithm 3.

**Algorithm 3** Gaussian-Sampled Random Rotation Matrix Generation

---

**Input:** Number of dimensions $n$, number of matrices $k$
**Output:** Stack of random rotation matrices
Initialize empty list $\mathcal{Q}$
Set $\epsilon \leftarrow 0.5$
**for** $i \leftarrow 0$ **to** $k - 1$ **do**
   **if** $i \mod \left(\frac{k}{16}\right) = 0$ and $i \neq 0$ **then**
     $\epsilon \leftarrow \epsilon \times 0.22360679775$
   **end if**
   Generate random matrix $M$: $M \sim \mathcal{N}(0, 1)^{n \times n}$
   Compute perturbed matrix: $A \leftarrow I_n + \epsilon \times M$
   Compute QR decomposition: $Q, R \leftarrow \text{QR}(A)$
   **if** $\det(Q) < 0$ **then**
     Flip first column of $Q$: $Q[:, 0] \leftarrow -Q[:, 0]$
   **end if**
   Add $Q$ to $\mathcal{Q}$
**end for**
**return** $\mathcal{Q}$

---

