# OpenReview forum: "How Classifier Features Transfer to Downstream: An Asymptotic Analysis in a Two-Layer Model"
_NeurIPS.cc/2025/Conference — NeurIPS 2025 poster_

### Official Review · Reviewer_fCzN · 2025-06-10

**Clarity:** 2
**Significance:** 2
**Originality:** 3
**Rating:** 4
**Confidence:** 2

**Summary:**

The work establishes a theoretical framework to justify that the source representations referred to as spike direction can serve to indicate the cohesion and separability in the sense of alignment when applying downstream unseen data. The paper treats multi-classification as multiple two-class classification problems, leading to the Eq. (2) represented hidden-layer gradient in terms of MSE loss of a two-layer neural classifier as formulated in Eq. (1). Then, Eq. (4) formulates the feature undergoing update into two parts: The original one of random initialization, and the incremental part by applying a bounded proxy of gradient defined in Eq. (3), in the framework of Hermite decomposition, where the latter one is termed as Dominant Feature or Spike Component. Then, the train-unseen similarity is deifined as the alignment between the mean unseen pattern and the spike direction to allow computing the so-called Cohesion and Separability (definition 4.3), which leads to proposition 4.4 and proposition 4.5 in Eq. (6) and Eq. (7), respectively. Eq. (6) and Eq. (7) reveal that the cohesion increases with the alignment of unseen pattern to the spike direction, and the separability increases when the unseen patterns are aligned oppositely in reference to the spike direction. Figure 3(c) illustrates some cases in which the two patterns are distinguishable/indistinguishable. Numerical experiments are included.

**Questions:**

Q1: In fact, we concern manifolds more than the mean representations of two classes in measuring the separability. If Figure 3(c) illustrates the distribution of two classes, it is possible that only a few examples are indistinguishable when projecting to the spike direction, subject to the manifolds of distributions. Besides, the practical case is more complex in that the convolution or self-attention operations have modified the representations prior to feeding them to the classifier and the end-to-end learning can thus avoid the ad hoc cases shown in Figure 3(c).

Q2: I cannot understand why recall@1 can manifest the alignment of unseen pattern to the spike direction in Eq. (6) and Eq. (7), which is subject to only the mean representation.

Q3: A possible bug is: The recall@1 concerns the nearest neighbor only but omits the overall distribution of the test data.

Q4: Does the bias term omitted in Eq. (1) affect the generality of the neural approximation?

**Ethical Concerns:**

["NO or VERY MINOR ethics concerns only"]

**Final Justification:**

It is the only paper I thought publishable in my reviewing list. My final rating is accpet.

My final justification is accept. I reviewed 5 papers but the quality of the other 4 papers is not plausible, very rare case in my experience. This paper seems fine althought not as fine as I expected. Please read my “Final Justification”, which is finished much earlier. My final recommendation is accept but I cannot improve the rating.

**Limitations:**

The experiments are not solid.

**Paper Formatting Concerns:**

No.

**Quality:**

3

**Strengths And Weaknesses:**

Strength: Domain adaptation is studied from a theoretical perspective.
Weakness: The conclusion is not surprising, and not informative to indicate the possible trend.

---

> ### Author Rebuttal · Authors · 2025-07-31
>
> We appreciate your positive evaluation and constructive feedback, particularly regarding the clarity of the paper. First, we would like to take this opportunity to address some misunderstandings, especially those related to Q1 and the interpretation of Figure 3(c) and the usage of recall@1 (Q2, Q3) with additional experiments in the Expr 4 setup confirm that both **Separability and Recall@1 serve as indicators of clustering performance**, consistent with our findings in Expr 1, 2, and 3. Then, we address the concern regarding the conclusion and respond to Question 4.
>
> ## interpretation of Figure 3(c) & Question 1
>
> We believe the phrase ”*the two patterns”* in your summary refers to the Cohesion and Separability patterns, and we’d like to clarify this potential confusion. As shown in Eq. (6) and Eq. (7), the train-unseen similarity governs Cohesion and Separability, and their relationships are illustrated in Figures 1, 2, 4, and 5. In contrast, Figure 3(c) stems from Section 4.2 and aims to illustrate that, **depending on $\beta$, inputs can become distinguishable or indistinguishable in the learned feature $s_L$**. This figure presents a pair of **individual points** (rather than class means or distributions), highlighting a key insight for the experimental section and the main conclusion: *only semantically relevant classes contribute meaningfully to feature extraction*.
>
> ### Response to Question 1
>
> From a distributional viewpoint, when class **manifold distributions align with the spike direction $\beta$, they are separable; if they lie in subspaces orthogonal to $\beta$, they become indistinguishable**, regardless of manifold structure (Corollary 4.7). Your consideration of manifolds is valid, and our key contribution is identifying the precise condition—alignment with **$\beta$ that governs distinguishabilit**y.
>
> We expect that, despite the nonlinear transformations in deep architectures (e.g., convolution or attention), inputs orthogonal to the learned discriminative directions tend to persist after training, owing to the structure of matrix operations. Such inputs could be indistinguishable to the model, which underscores the importance of identifying spike directions—a central goal of our analysis in section 4.2. Moreover, discovering such directions in deeper architectures would also be crucial for understanding model behavior.
>
> ## Response to Question 2
>
> Your question may stem from a few conceptual confusions. We first address your concern directly and then provide clarifications to ensure a precise understanding of the relationship between recall@1 and the alignment of unseen patterns to the spike direction.
>
> First, **recall@1 serves as a surrogate measure of the Cohesion and Separability** of learned features (as detailed in our response to Q3). According to the interpretation given in Eq. (6), Eq. (7), and Section 4.1, this **Cohesion and Separability is governed by the train-unseen similarity,** which you referred to as the *alignment of unseen patterns to the spike direction*.
>
> Therefore, our central claim is that **recall@1 is governed by the alignment of unseen patterns to the spike direction**, which is supported by our empirical results (Experiments 1–4).
>
> We would also like to clarify the following technical points to avoid further misunderstanding:
>
> 1. Eqs. (6) and (7) involve more than mean representations.
>     - As stated on Line 35, *Cohesion and Separability represent standard clustering performance criteria* [Clémençcon, 2011; Papa et al., 2015; Liu et al., 2017; Li and Liu, 2021]. (See also Line 705 for further background.)
>     - Eqs. (6) and (7) provide approximations of the Cohesion and Separability of feature representations learned after a single gradient step (see Theorem 3.3, Definition 3.4, Line 173, and Appendix L).
>     - Hence, these equations quantify the clustering performance of the trained features for a given unseen class.
> 2. The alignment of unseen patterns to the spike direction is distinct from recall@1.
>     - *This alignment is captured by $\beta^\top \mu$, as defined in Eq. (6) and Eq. (7), and reflects train-unseen similarity.*
>     - As in our analysis (Line 282), the alignment is captured through train-unseen similarity and semantic similarity, not recall@1.
>     - Instead, this alignment determines the clustering performance, and thus indirectly affects recall@1, but the two are conceptually distinct.
>
> ## Response for Question 3: Recall@1 serves as a practical surrogate for clustering performance, cohesion, and separability.
>
> **When both Cohesion and Separability are high, the nearest neighbor is more likely to belong to the same class, resulting in a higher recall@1** measured in *inner product*. This is a standard practice in metric learning (Wang et al., [2018], Zhai and Wu [2019], An et al., [2023]) for measuring clustering performance. While recall@1 focuses on the top-1 neighbor, we claim that it is a practical and reliable surrogate for evaluating the combined effect of Cohesion and Separability. Moreover, in Experiments 1, 2, and 3, we observed that recall@1 exhibits a consistent increasing trend that aligns with the behavior of Separability, further supporting its relevance in evaluating performance (See appendix Figures 25, 26, 27, and 28).
>
> ## Additional Experiment of Separability
>
> To address concerns about Recall@1 as a surrogate for clustering criteria, we additionally evaluated **Separability in Definition 4.3 for Expr. 4**, alongside *Expr. 1–3*. Following standard practice (Zhai & Wu, 2019), we trained with unit-normalized features and, thus, report Separability only.
>
> We observed that models **best separate test data when trained on the corresponding domain (e.g., CAR to CAR), consistent with the Recall@**1, reinforcing our main argument of Expr 4. The results are provided based on the *ResNet18 with the initialization* setting. Further experiments under different setups also confirmed this pattern and will be included in the camera-ready version.
>
> Lastly, as discussed in our response to Reviewer MzGZ, we provide supporting evidence for a meaningful connection between **train-unseen similarity** and **semantic similarity**. We also test the validity of our theory in both **multi-layer** and **multi-step** settings in our response for MzGZ.
>
> | Train data | mean sep | std |
> | --- | --- | --- |
> | CAR test |  |  |
> | CAR | **-0.25** | 0.12 |
> | CUB | -0.75 | 0.12 |
> | SOP | -0.69 | 0.13 |
> | ISC | -0.86 | 0.08 |
> | CUB test |  |  |
> | CAR | -0.63 | 0.12 |
> | CUB | **-0.28** | 0.19 |
> | SOP | -0.42 | 0.13 |
> | ISC | -0.79 | 0.08 |
> | SOP test |  |  |
> | CAR | -0.29 | 0.10 |
> | CUB | -0.14 | 0.11 |
> | SOP | **-0.01** | 0.01 |
> | ISC | -0.28 | 0.10 |
> | ISC test |  |  |
> | CAR | -0.82 | 0.07 |
> | CUB | -0.81 | 0.06 |
> | SOP | -0.47 | 0.09 |
> | ISC | **-0.01** | 0.01 |
>
> ## Addressing the Weakness in the Conclusion
>
> To address the concerns raised regarding the strength of our conclusion, we propose the following improvements to articulate our contributions better and clarify the broader implications of our work:
>
> 1. **Extension of prior theory**: Our analysis generalizes existing frameworks by incorporating *non-centered sub-Gaussian input distributions* (Refer Appendix I–J) and *deterministic feature approximations for classification tasks* (Refer line 650).
> 2. **Versatile analytical framework**: Our approach opens up several promising avenues for extension, including applications to *multi-step training dynamics*, and *potential connections to Neural Collapse* under arbitrary input distributions.
> 3. **Future extension to multi-class softmax**: A natural next step is to extend the notions of *Cohesion and Separability* to the multi-class softmax setting, incorporating techniques such as *normalization* and *temperature scaling* to more closely reflect practical neural network training.
> 4. **First quantitative link between semantic alignment and clustering**: To our knowledge, this is the *first work to quantitatively connect train-unseen alignment with open-set clustering performance*, rather than relying solely on hypothesis-level arguments (line 671).
> 5. **A representation-driven view of coreset construction**: Our work suggests a potential new direction for representation-aware coreset selection—for instance, *reducing the number of training classes or training data not contributed to $\beta$ direction to lower computational complexity* while still enabling successful clustering and feature transfer.
>
> Taken together, these points clarify that our work offers not only strong theoretical insights but also **practical relevance and extensibility** across a range of future research directions.
>
> ## Response to Question 4
>
> We excluded the bias term in Eq. (1) for analytical simplicity. This follows the convention in several recent studies, such as Moniri et al. [2024] and Ba et al. [2022], where feature learning is analyzed without bias. As Ba et al. [2023] and Mousavi-Hosseini et al. [2023], incorporating the bias term is possible, but the simplified formulation in Eq. (1) is sufficient to address the research question.

---

> > ### Comment · Reviewer_fCzN · 2025-08-06
> > **Thank you for your response**
> >
> > Your endeavor is acknowledged.

---

### Official Review · Reviewer_yNMA · 2025-07-01

**Clarity:** 2
**Significance:** 2
**Originality:** 2
**Rating:** 4
**Confidence:** 4

**Summary:**

This paper theoretically investigates the feature transfer abilities of two-linear neural network classifiers.

The authors consider the setting of a two-layer neural network with specific assumptions on the activation function and initialization of parameters and uncentered sub-gaussian data distribution. In a 1-vs-1 K-class classification setting with MSE loss, one gradient descent (GD) step is asymptotically shown to be approximated by a specific rank-1 structure. This yields a feature decomposition result that essentially decomposes the feature mapping function realized by the network after one GD step into two components: random initialization features and a so-called Spike Component, which is effectively responsible for feature learning from the training data.

Given this, the authors derive closed-form expressions for the two clustering performance criteria for unseen data, Cohesion and Separability, w.r.t. the Spike Component (Propositions 4.4 and 4.5) and relate them to the so-called Train-Unseen Similarity.
Namely, *the better the alignment between training and testing data, the more effective the clustering of test data in the feature space*.

Experimental results on synthetic and image data support theory.

**Questions:**

1. Can the current results obtained for the non-conventional setting of 1-vs-1 K-class classification be transferred to the more standard K-way multi-class classification (i.e., when last layer is an $N \times K$ matrix)?
2. What if the classes are not perfectly balanced?
3. Can the requirement of $d \to \infty$ be avoided in the analysis? It seems very unnatural.
4. Line 111: if $W_0[i]$ denotes the $i$-th row of matrix $W_0 \in \mathbb{R}^{d \times N}$, how can it be initialized from the $(d-1)$-dimensional sphere? Is that a typo?
5. Can the sphere initialization be relaxed with the Gaussian initialization?
6. Line 114: $P \to K$.
7. Does Shifted ReLU actually follow Assumption 2.1? It seems that it is not bounded by some $\lambda_\sigma$ a.s.
8. How exactly are $H_k(z)$ defined?
9. Doesn't Theorem 3.1 imply the degenerate limit of $F_0(x) \to 0$ as $n \to \infty$?
10. There is a discrepancy between the definition of $\beta$ (without indices) and the way it is used in the text $\beta_{ij}$ (with indices).
11. Line 162: Definition 4.3.
12. Line 165: What is $\mathcal{n}(\mu_i, I_d)$?
13. Line 166: What is $\rho_{k,k',r,r'}$?
14. What are $C(s_L)$ and $S(s_L)$ used in Propositions 4.4, 4.5?
15. Recall@1 definition: is the data sampled from the test or train distribution?

**Ethical Concerns:**

["NO or VERY MINOR ethics concerns only"]

**Final Justification:**

I have read the authors rebuttal and raised my score, since a large portion of my concerns were addressed. Still, I consider the results of this work very specific as it follows a very narrow setting, which is far from the conventional practice.

**Limitations:**

Yes

**Quality:**

3

**Strengths And Weaknesses:**

# Strengths

This paper proposes a profound theoretical analysis of feature transferring capabilities in two-layer neural networks.
The presented results and proof techniques are novel and extend the assumptions and scope of the prior work.
Theoretical analysis is well supported by both illustrative examples (Fig. 1-3) and actual experiments on synthetic (Fig. 4-5) and real data (Fig. 6-7).

# Weaknesses

There is a significant gap between the specific setup considered in this study for theoretical derivations and real practice.
Not to mention the general questionability of applying the two-layer analysis to explain feature learning in actual deep neural networks, I hardly find the parallels between the common practice and the provided results for 1-vs-1 perfectly class-balanced K-classification in the limit of data dimension $d \to \infty$ and one gradient descent step with learning rate $\eta = 1$ under additional assumptions.
To me, the main contribution of this work &mdash; proving in a very limited context an intuitive thesis that *clustering performance in feature space depends on the degree of alignment between training and testing distributions* &mdash; is marginal and does not provide any significant novel insights.

The paper is not very clearly written, it contains a large amount of inaccuracies, undefined values, and notation abuse that hinders comprehension of the paper content.

Related Work: I am not an expert in this specific domain, but it seems that some related works on transfer learning theory (in two-layer models specifically) might be missing here.

---

> ### Author Rebuttal · Authors · 2025-07-31
>
> We appreciate the detailed reviews aimed at improving our paper. In particular, we are grateful for your recognition that our theoretical results provide novel extensions of prior work to non-centered sub-Gaussian distributions and are well supported by empirical evidence. However, you expressed concerns regarding the gap between our theoretical assumptions and practical neural networks, as well as the claim that our theoretical findings do not offer significant novel insights. We would like to first address these concerns before responding to your questions.
>
> ## On the Gap Between Theoretical Setup and Practice
>
> We acknowledge the simplified nature of our model (1. two-layer, 2. 1-vs-1 classification, 3. perfectly class-balanced, 4. proportional regime $d \rightarrow \infty$ 5. one-step gradient descent, 6. learning rate 1). However, **we support our claims with experiments on deep networks (e.g., ResNet) as in Section 5** and observe that the theoretical predictions remain consistent in practice. On top of that, we conducted **additional experiments for the rebuttal by extending our assumption**. Specifically, we performed experiments under the Expr 1 setting with **(1) 2-layer networks trained for up to 5 steps, and (2) networks with 2, 3, 4, and 5 layers.** In both cases, we observed that as the *train-unseen similarity $\beta^\top \mu$ increases, Cohesion and Separability __consistently improve__.* These results are included in the final part of our rebuttal to Reviewer MzGZ under Multi-Step and Multi-Layer Experiments.
>
> Lastly, **this setting parallels a large body of existing theoretical work** (see Table 1) and is justified for the following reasons:
>
> 1. Two Layer networks
>
>     We adopt this two layer network as it adequately enables analytical insight into feature learning and transfer, and has **proven effective and sufficient in prior work for explaining key phenomena in deep models**, such as feature learning and the superiority of neural networks (Goldt et al. [2020], Ba et al. [2022], Ba et al. [2023], Cui et al. [2024], Mousavi-Hosseini et al. [2023]).
>
> 2. 1-vs-1 classification: addressed in question 1
> 3. Perfectly class-balanced: addressed in question 2
> 4. Proportional regime: addressed in question 3
> 5. One step update
>
>     As supported by a growing body of theoretical work (Appendix Table 1), one-step updates have been successfully used to explain feature learning effects in real neural networks. This setting is not an arbitrary simplification but **a theoretically grounded framework** (Moniri et al. [2024],  Dandi et al. [2024], Demir and Dogan [2025]), especially given the *well-documented strong influence of early-phase dynamics on final model behavior (Ba et al., 2022)*. Within this framework, we show that **even a single gradient step reveals meaningful clustering** through the emergence of the $s_L$ term which sufficiently addresses our core research question. (see Appendix L).
>
> 6. Learning rate 1
>
>     We would like to point out a detail that may have been overlooked respectfully — please refer to line 119. While **our theoretical results hold for learning rates $\eta = \Theta(1)$**, we represent the learning rate as 1 purely to avoid notational complexity. This setting is equivalent to using a small learning rate (e.g., 0.01) in standard neural networks.
>
>
> ## Novelty
>
> We will explain below why our work goes far beyond merely rephrasing known intuitions.
>
> 1. A Novel Framework for Analyzing Classifier - Deterministic Spike-Based Closed-Form Decomposition:
>
>     We establish a foundational framework for analyzing the **feature behavior of classifiers on new inputs**, rather than focusing on the commonly studied regression task (Ba et al. [2022], Dandi et al. [2023], Cui et al. [2024], Dandi et al. [2024]). Notably, **our predominant feature component β admits a closed-form, deterministic expression**, unlike the stochastic feature decompositions used in recent works such as Demir and Dogan [2025] (see line 650). As a result, we are able to directly analyze the output of the learned feature map in Section 4.
>
> 2. Counterintuitive Finding – Importance of Relevant Classes, not Many Classes:
>
>     Beyond clustering, Corollary 4.7 in Section 4.2 and Experiments V–VI show that, contrary to common belief, which emphasizes the importance of more training samples, **feature expressivity depends mainly on semantically related classes.** This insight challenges the focus on quantity (Brown et al., 2020; An et al., 2023; line 337), suggesting that reducing training classes, not just samples, may improve feature transfer performance and efficiency. Moreover, our work **suggests a potential new direction for coreset selection** —for instance, *reducing the number of training classes or training data not contributed to β direction to lower computational complexity while still enabling successful clustering and feature transfer.*
>
> 3. Unifying empirical observations and prior theories - Theoretical Quantification of Feature Transfer Phenomena:
>
>     Our work offers a theoretical contribution by unifying empirical observations and prior theories, clarifying when feature transfer succeeds or fails (Section 4.1). While feature transfer is known to work better when unseen inputs resemble training data, previous metric learning studies mainly focus on performance gains without consistent explanations (See 671). In contrast, our framework precisely identifies both when feature transfer works (Propositions 4.4 and 4.5) and when it breaks down, such as when multiple unseen classes map to the same spike β, causing Separability to fail despite high similarity. **This quantitative characterization of transfer success and failure is a key strength of our work**.
>
>
> ## Response to Questions
>
> 1. To ensure compatibility with existing regression-based frameworks and to allow for a more tractable derivation, we adopted a 1-vs-1 classification setting. However, by avoiding the matrix formulation in Equation 2 and instead representing the labels y using one-hot vectors, our analysis could potentially be extended to general k-class classification problems as well.
> 2. Our proofs only require that $X$ is sampled from class-conditional distributions that are *non-centered sub-Gaussian*, so **the proof technique for $\beta = \frac{1}{n} X^\top y$ remains unchanged even in imbalanced settings**.
> 3. The assumption $d \to \infty$, commonly used in feature learning research (e.g., Ba et al. [2022]), facilitates theoretical analysis of neural networks. Ba et al. [2022] justify this regime as representative of modern neural networks with large datasets and architectures. We adopt it to isolate the key low-rank spike term (Theorems 3.1 and 3.3). **From Section 4 onward, our results assume a fixed, finite, and sufficiently large sample size $N$**, supported by **experiments demonstrating the theory’s applicability to large, finite neural networks.**
> 4. The notation $\mathcal{S}^{d-1}$ refers to the $(d-1)$-dimensional sphere embedded in $\mathbb{R}^d$. $W_0[i]$ is a $d$-dimensional vector indexed by $i \in [[N]]$. Sorry for the mistake.
> 5. To ensure consistency in the theoretical objects we analyze, we adopted the assumption used in Moniri et al. [2024]. While Ba et al. [2022] initialize parameters with Gaussian distributions, which would lead to differences in the proof techniques, we expect similar qualitative results under that alternative setup.
> 6. We apologize for the typographical error. The intended meaning is that the problem corresponds to *K* classification tasks, not *P*. Thank you for pointing this out.
> 7. This is a typographical error. The assumptions are the same as in Moniri et al. [2024], where σ′, σ′′, and σ′′′ are bounded. **The boundedness of σ itself was not used in the proof.** We apologize for the mistake and will make the necessary correction.
> 8. The Hermite polynomial $H_k$ is discussed in detail in Appendix E. For readability, we will include a reference to it in the main text.
> 9. Theorem 3.1 concerns an approximation of the gradient and is not a direct result about $F_0$. Additionally, although the term $\frac{1}{n}$ appears in $A_{ij} = \frac{c_1}{n} X_{ij}^\top y a_{ij}^\top$, as summarized in Remark K.4 and Equation 24, as $n \rightarrow \infty$, almost surely $||\beta||$ and $||a_{ij}||$ remain at constant scale, so $||A_{ij}||$ also stays at a constant scale and does not degenerate as $n$ grows.
> 10. $\beta$ refers to $\beta_{ij}$, but the distinction of i and j is not essential, so we omit the subscripts. For clarity, we will explain this in the main text. We apologize for any confusion caused.
> 11. Thank you for pointing out the typo.
> 12. It is a Gaussian distribution. We will make this explicit.
> 13. We simplified it as it does not play a significant role in our analysis. *It is a term unrelated to $\beta$ and $\mu$ that appears in the proof*, and it is defined in Appendix H. We will indicate where it is defined.
> 14. We defined cohesion and separability in Definition 4.3 as **functionals** (functions from functions to scalars) that take **any** feature extractor F (which may be $s_L$, $F_0$, or $F_L$). It can be computed using $s_L$ instead of F in the definition. Using *F* seems to confuse, so that we will replace it with a different symbol, such as *G*.
> 15. Sampling from the **test distribution**, which is standard practice in the metric learning field, will be clearly described in our work.
>
> ## Response to Concerns on Related Work
>
> Related work on transfer learning theory is discussed in *Appendix D* (line 661). These studies examine how changes in the teacher function structure affect sample complexity. **To our knowledge, theoretical analyses of feature transfer—especially in metric learning or open-set clustering without further training—remain unexplored** (see line 671). For clarity, we provide a more detailed discussion of these aspects in the main text.

---

### Official Review · Reviewer_yqtE · 2025-07-01

**Clarity:** 1
**Significance:** 2
**Originality:** 3
**Rating:** 4
**Confidence:** 2

**Summary:**

This paper presents a theoretical and empirical analysis of how features learned by a two-layer classifier generalize to unseen classes in clustering tasks. The authors use a simplified setting — one gradient descent step and Hermite polynomial expansions — to decompose features into components and study how similarity between training and unseen data affects clustering quality. Experiments on both synthetic and real-world data support their theoretical findings.

**Questions:**

- Is it possible to perform a similar analysis of deeper/multi-step models?
- Did you find any unexpected behavior that could led to novel approeaches on the topic?

**Ethical Concerns:**

["NO or VERY MINOR ethics concerns only"]

**Final Justification:**

The authors addessed my concerns, wo I will raise my score towards borderline accept.

**Limitations:**

yes

**Paper Formatting Concerns:**

No concerns

**Quality:**

3

**Strengths And Weaknesses:**

**Quality:**
- (+) The theoretical analysis is detailed and mathematically sound.
- (+) The experiments are well-designed to support the theoretical claims.
- (-) A two-layer network trained with just one step is far from how real models are used.
- (-) The main result (features transfer better when unseen inputs are similar to training data) is well-known.

**Clarity:**
- (-) The introduction section is very hard to read. I recommend a profesional proofreading to make it more accessible to other researchers.
- (-) The English writing is often overly complex. It is very hard to follow.
- (-) I suggest the authors to separate simpler explanations from hard demonstrations.

**Significance:**
- (+) Understanding when and how features generalize to unseen classes is an important question in the machine learning community.
- (+) The topic is relevant for researchers working on transfer learning, clustering, and metric learning.
- (-) The contribution is **mostly theoretical formalization** of ideas that are already intuitively known (e.g., aligned inputs transfer better).
- (-) The impact on practice is limited, as the analysis is far from current Machine Learning architectures. It would be interesting to know is the assumptions correlate to bigger networks.

**Originality:**
- (+) The authors' approach seems creative and novel.
- (-) It is not a negative per se, but it is dissappointing that the theory confirms known behavior rather than discovering surprising or counterintuitive results

---

> ### Author Rebuttal · Authors · 2025-07-31
>
> We appreciate the recognition of the theoretical soundness and experimental rigor of our work. Your comments encouraged us to clarify key contributions more precisely.
>
> Notably, we would like to respectfully highlight an aspect that may have been overlooked in the summary. Beyond clustering analysis, a key contribution of our work lies in extending the analysis of two-layer neural networks from regression to classification. To address this, we consider inputs drawn from a non-centered sub-Gaussian distribution, broadening applicability to classification tasks. **To our knowledge, this is the first such tool to enable tractable, closed-form analysis of feature learning in classification.** Moreover, this framework not only supports clustering results but more generally **characterizes how feature expressivity for unseen classes depends on the number of semantically related training classes** (see Section 4.2, Corollary 4.7).
>
> ## On the Gap Between Theoretical Setup and Practice
>
> We acknowledge the simplified nature of our model (two-layer, one-step gradient descent).
>
> However, we support our claims with **experiments in Section 5 on deep networks** (e.g., ResNet) and observe that *the theoretical predictions remain consistent in practice.* We believe this combination of theory and empirics provides a compelling case for the relevance of our simplified setting.
>
> On top of that, we conducted additional experiments for the rebuttal by extending our original 2-layer, 1-step setup. Specifically, we performed experiments under the Expr 1 setting with (1) 2-layer networks trained for up to 5 steps, and (2) **_Bigger networks_** with  3, 4, and 5 layers. In both cases, we observed that as the *train-unseen similarity $\beta^\top \mu$ increases, Cohesion and Separability consistently improve.* These results are included in the final part of *our rebuttal to Reviewer MzGZ under Multi-Step and Multi-Layer Experiments*.
>
> Lastly, this setting is consistent with a large body of works (see Appendix D) and is justified for the following reasons:
>
> 1. Our assumptions enable analytical tractability and provide clear insight into the mechanisms of feature learning and transfer. In particular, our central research question—*”Can we capture the presence of feature learning in classification and identify the conditions where features cluster effectively in new distributions?”—can be addressed using a single training step of a two-layer neural network*.
> 2. One-step update captures *early-phase dynamics that are known to be strongly influential in determining final model behavior* (Early Phase of NN Optimization Section of Ba et al., 2022).
> 3. *Shallow network* has been successfully used in prior work to explain **nontrivial phenomena observed in bigger networks**, such as the superiority of neural networks over traditional kernel methods (Ba et al. [2022]), the mechanisms of feature learning (Dandi et al. [2023], Moniri et al. [2024]), the mechanisms of learning the internal structure of data (Ba et al. [2023], Mousavi-Hosseini et al. [2023]), the representational power of random features (Louart et al. [2017], Goldt et al. [2020], Cui et al. [2024]), lottery ticket hypothesis([1]), generalization, interpolation and double descent ([2, 3, 4]), overparameterization ([5]), Catastrophic Forgetting([6])
>
> [1] Malach, Eran, et al. "Proving the lottery ticket hypothesis: Pruning is all you need." *International Conference on Machine Learning*. PMLR, 2020.
>
> [2] Suzuki, Keita, and Taiji Suzuki. "Optimal criterion for feature learning of two-layer linear neural network in high dimensional interpolation regime." *The Twelfth International Conference on Learning Representations*. 2023.
>
> [3] Abeykoon, Chathurika S. *The Double Descent Behavior in a Two Layer Neural Network for Binary Classification*. Diss. The University of Mississippi, 2023.
>
> [4] Xu, Ruichen, and Kexin Chen. "Rethinking benign overfitting in two-layer neural networks." *arXiv preprint arXiv:2502.11893*(2025).
>
> [5] Zhang, Yaoyu, et al. "Local Linear Recovery Guarantee of Deep Neural Networks at Overparameterization." *Journal of Machine Learning Research* 26.69 (2025): 1-30.
>
> [6] Li, Boqi, Youjun Wang, and Weiwei Liu. "Towards Understanding Catastrophic Forgetting in Two-layer Convolutional Neural Networks." *Forty-second International Conference on Machine Learning*.
>
> We hope that our findings, *alignment in real-world networks, the empirical validity of our theoretical predictions in multi-step and multi-layer settings, and strong connections to prior literature linking simplified assumptions to deep models* help convey the broader relevance of our work and encourage further interest in our research.
>
> ## Response to Question 1
>
> ### Deeper Networks
>
> While our current two-layer model already offers insight into how feature learning enhances clustering, building on this with deeper architectures could deepen our understanding.  Existing techniques (e.g., Fan & Wang [2020]; Nichani et al. [2023]; Wang et al. [2023]) have  analysis for multi-layer settings, suggesting that our approach may extend similarly. Such extensions could provide insights into how neural networks learn to classify by selectively amplifying or suppressing directions in input space. Ultimately, this line of inquiry helps further explain how **generally transferable features** **emerge** across layers in deep models.
>
> ### Multi-step training
>
> *Dandi et al. (2023)* proposed an analytical framework for studying multi-step training dynamics in two-layer neural networks. They analyzed the so-called *staircase property* in such networks and demonstrated that features are learned progressively from the data structure, leading to a monotonic decrease in generalization error. Given the similarity between their setting and ours, *their methods could potentially be adapted to study the evolution of classifier features across multiple training steps*, beyond the one-step clustering analysis we focus on.
>
> ## Novelty & Question 2
>
> We understand your concern that the core result—features cluster better when unseen inputs resemble training data—may seem unsurprising. However, we believe our work goes significantly beyond rephrasing known intuitions:
>
> 1. Novel Theoretical Tools for Classifier Analysis – Extending to Non-Centered Sub-Gaussian Distributions:
>
>     Our analysis extends prior work (see Table 1 in the Appendix) by generalizing feature learning theory from centered Gaussian settings to non-centered sub-Gaussian data, thus enabling application to classification tasks beyond regression. This requires new technical tools (see line 93, Appendices I–J) that allow analysis under arbitrary labels and finitely supported distributions.
>
>     In Appendix I, we extend classical results on standard Gaussian expectations of Hermite polynomial products (O'Donnell [2021], Moniri et al. [2024]) **to non-centered, non-unit-variance Gaussians**. We then **derive computable bounds of the above expectations for general sub-Gaussian inputs.** In Appendix J, we generalize key results from Vershynin [2010, 2018]—originally for centered sub-Gaussians—to *the non-centered case*.
>
>     These contributions are essential for analyzing models trained on non-Gaussian separable data and offer broadly applicable tools for future theoretical work.
>
> 2. A Novel Framework for Analyzing Classifier - Deterministic Spike-Based Closed-Form Decomposition:
>
>     We establish a **foundational framework for analyzing the feature behavior of classifiers on new inputs**, rather than focusing on the commonly studied regression task (Ba et al. [2022] and Table 1) or training data (See line 685). Notably, our predominant feature component β admits a closed-form, deterministic expression, unlike the stochastic feature decompositions used in recent works such as Demir and Dogan [2025] (see line 650). As a result, *we are able to directly analyze the output of the learned feature map* in Section 4.
>
> 3. **(Question 2)** Counterintuitive Finding – Importance of Relevant Classes, not Many Classes:
>
>     While it is widely believed that increasing the number of training samples improves feature quality (Brown et al., 2020; An et al., 2023; line 337), our results in Corollary 4.7 and Experiments V–VI suggest a more nuanced view—namely, that **only semantically aligned classes contribute meaningfully to expressivity**. This challenges the conventional emphasis on quantity and opens a promising direction aligned with recent work in coreset selection, where reducing the number of training classes rather than just the number of samples can enhance feature transfer.
>
> 4. Unifying empirical observations and prior theories - Theoretical Quantification of Feature Transfer Phenomena:
>
>     Our work offers a theoretical contribution by unifying empirical observations and prior theories, clarifying when feature transfer succeeds or fails (Section 4.1). While feature transfer is known to work better when unseen inputs resemble training data, previous metric learning studies mainly focus on performance gains without consistent explanations (Chopra et al. 2005, Liu et al. 2018, El-Nouby et al., 2021, Caron et al., 2021, Ermolov et al. 2022). In contrast, our framework precisely identifies both when feature transfer works (Propositions 4.4 and 4.5) and when it breaks down, such as when multiple unseen classes map to the same spike β, causing Separability to fail despite high similarity. **This quantitative characterization of transfer success and failure is a key strength of our work** (See 671).
>
> ## Response to Clarity
>
> Finally, we appreciate your suggestion regarding the introduction and presentation. In the revised version, we will:
>
> 1. Simplify the introduction and text by breaking down longer sentences and providing intuitive context before diving into formalism.
> 2. Separate intuition and theory throughout the paper to improve accessibility.

---

### Official Review · Reviewer_MzGZ · 2025-07-02

**Clarity:** 3
**Significance:** 3
**Originality:** 3
**Rating:** 4
**Confidence:** 3

**Summary:**

This paper theoretically analyzes how features learned by a classifier perform in clustering unseen classes. The study uses a simple model where a two-layer nonlinear network is trained with a single gradient descent step. As a result, it is shown that the learned features can be decomposed into a component derived from random initialization and a "spike component" that depends on the training data. Based on the properties of this spike component, the paper concludes that clustering Cohesion and Separability are determined by the similarity between the training data and unseen data distributions. Specifically, higher similarity improves Cohesion, while Separability changes depending on the assignment of unseen classes. It was also shown that feature expressiveness diminishes for inputs that are irrelevant to the training data. These theories are validated through experiments on both synthetic data and real-world benchmark datasets.

**Questions:**

- Your work focuses on a single gradient descent step. How do you anticipate the behavior of the spike component sL would change if multiple training steps or epochs were considered? Specifically, how might your conclusions (especially regarding Cohesion and Separability) be extended or modified as more complex features are learned?
- In Expr. IV, you interpret the theoretical "train-unseen similarity" as practical "semantic similarity." Would it be possible to perform a more quantitative analysis to bridge this gap? For instance, one could try to approximate the spike direction β from a model trained on one domain (e.g., CAR) and compute its inner product with the class means μ from another domain (e.g., CUB).
- Your analysis of Separability shows that the "assignment" of unseen classes to training classes is critical. This assignment can be interpreted as being based on the learned decision boundary. Do you have any insights into the stability of this assignment itself (e.g., its robustness to initial conditions or minor perturbations in the data)? If the assignment is unstable, it seems that predicting Separability would also become difficult.

**Ethical Concerns:**

["NO or VERY MINOR ethics concerns only"]

**Limitations:**

yes

**Quality:**

3

**Strengths And Weaknesses:**

**Strengths**
- The framework of decomposing learned features via a "spike component" and explaining clustering performance (Cohesion, Separability) with "train-unseen similarity" is novel and provides deep insights into the mechanics of feature transfer. It moves beyond the empirical rule-of-thumb that simply "training on large datasets is better."
- The paper carefully validates its theoretical claims in two stages: (1) rigorous validation on synthetic data that meets the theoretical assumptions, and (2) practical validation in real-world scenarios with deep models (ResNet). This dual approach demonstrates both the theory's validity and its practical implications, making the paper highly persuasive.

**Major Weaknesses**
- The analysis is confined to "a two-layer network trained with a single gradient descent step." Modern deep learning models are much deeper and are trained through complex dynamics over multiple epochs. How the insights from this simplified setting generalize to deeper, more complex models requires further discussion. While the experiments with ResNet on real data show consistent trends, a gap remains between the theory and practice.
- The core theoretical metric, "train-unseen similarity" (β⊤μ), is clearly defined. However, in experiments on real data (e.g., Expr. IV), it cannot be directly computed and is replaced by the intuitive concept of "semantic similarity." The relationship between these two metrics, while plausible, is not quantitatively analyzed, leaving the applicability of the theory somewhat ambiguous.

**Minor Weaknesses**
- While it is standard for major proofs to be in the appendix, some key theoretical steps (e.g., the transition from Theorem 3.3 to the definition of the Dominant Feature) feel a bit rushed in the main text. Including more intuition or a brief sketch of these steps in the main body would improve clarity for the reader.
- As noted in Appendix A, the technique of approximating the activation function with polynomials has its limits. Specifically, it can require high-dimensional spaces for computation, and in finite dimensions, a discrepancy can arise between the approximation and the actual network's behavior. A more in-depth discussion in the main text on the validity of this approximation and its potential impact on the results would further strengthen the paper's credibility.

---

> ### Author Rebuttal · Authors · 2025-07-31
>
> We sincerely thank the reviewer for their thoughtful and accurate summary of our work. In this rebuttal, we address the concerns you raised regarding the **weaknesses and questions**.
>
> ## Response to Question 1 and Major Weaknesses 1
>
> We share **intuition on multi-step training and deeper models**. In addition, we provide **supporting experiments** showing that our theoretical predictions largely hold in both multi-step and multi-layer settings.
>
> ### For multi-step training
>
> We expect that multi-step training refines the spike component by amplifying directions that **enhance Separability** while suppressing others, causing class features to collapse more tightly and become better separated. This process may lead to effective information compression for classification and improved transferability. To this end, one may build on Dandi et al. (2023), who analyze the "staircase property" in multi-step training.
>
> ### For the deeper model
>
> While our current two-layer model already offers insight into how feature learning enhances clustering, building on this with deeper architectures could deepen our understanding.  Existing techniques (e.g., Fan & Wang [2020]; Nichani et al. [2023]; Wang et al. [2023]) have already generalized conjugate kernel analysis to multi-layer settings, suggesting that our approach may extend similarly. Such extensions could help further explain how generally transferable features emerge. Notably, our multi-layer experiments showed that when $\beta^\top\mu$ becomes too large, especially in 3- and 4-layer models, Separability may not improve. **Studying this critical threshold could be an interesting direction**.
>
> ## Response to Question 2 and Major Weaknesses 2
>
> We appreciate your concern regarding the applicability of our theoretical framework to real-world networks. In real world validation of our feature transfer theory, we surrogate the theoretical quantity $\beta^\top \mu$ (train-unseen similarity) using a semantic similarity of  the dataset domain. This analogy is justified, as datapoints in each dataset (CUB, CAR, SOP, ISC) exhibit consistent visual characteristics within the dataset. Nevertheless, to address your request and strengthen our claim, **we conducted the following additional experiment**.
>
> ### Additional Experiment:
>
> As you suggested, for each dataset (CUB, CAR, SOP, ISC) with three seeds, we trained a model and *extracted the class-wise mean embeddings from the training set as a surrogate for $\beta$*. Similarly, we computed *class-wise mean embeddings from the test sets as a surrogate for $\mu$. Then, for each test class*, we computed the maximum cosine similarity with any of the train class means, as follows:
>
> ```python
> # same procedure for mean_embeddings_train
> for test_dataset in [cub, car, sop, isc]:
> 	embeddings, labels = extract_feature(model, test_dataset)
> 	mean_embeddings_test = torch.zeros(num_unseen_classes, dim)
> 	for uc in range(num_unseen_classes):
> 		mean_embeddings_test[uc] = embeddings[labels == uc].mean(dim=0).normalize()
> 		# get statistics
> 		sim = mean_embeddings_test @ mean_embeddings_train.T # (num_unseen, num_seen)
> 		max_values = sim.max(dim=1) # (num_unseen)
> 		median(max_values), mean(max_values), std(max_values)
> ```
>
> Findings:
>
> As shown in the table, **the maximum cosine similarity between unseen and train class means is consistently highest when the train and test domains match** (e.g., 0.84 for CUB → CUB, 0.78 for CAR → CAR). This supports our interpretation that semantic similarity across domains serves as a valid proxy for theoretical train-unseen alignment. We provide the results of ResNet18 trained initialization in the table below. We will include analyses of all setups and visualization in the camera-ready version.
>
> | Train data | median | mean | std |
> | --- | --- | --- | --- |
> | CAR test |  |  |  |
> | CAR | **0.80** | **0.78** | 0.09 |
> | CUB | 0.68 | 0.68 | 0.07 |
> | SOP | 0.64 | 0.64 | 0.06 |
> | ISC | 0.57 | 0.57 | 0.04 |
> | CUB test |  |  |  |
> | CAR | 0.58 | 0.58 | 0.09 |
> | CUB | **0.85** | **0.84** | 0.07 |
> | SOP | 0.67 | 0.67 | 0.06 |
> | ISC | 0.58 | 0.59 | 0.07 |
> | SOP test |  |  |  |
> | CAR | 0.53 | 0.53 | 0.10 |
> | CUB | 0.61 | 0.60 | 0.13 |
> | SOP | **0.68** | **0.69** | 0.10 |
> | ISC | 0.57 | 0.58 | 0.08 |
> | ISC test |  |  |  |
> | CAR | 0.59 | 0.59 | 0.06 |
> | CUB | 0.43 | 0.44 | 0.08 |
> | SOP | **0.70** | **0.70** | 0.08 |
> | ISC | *0.69* | *0.69* | 0.09 |
>
> ## Response to Question 3
>
> We believe there may have been a misunderstanding regarding the "assignment" in Note 4.6. The assignment we refer to is a theoretical construct defined **without training**, computed via the sign $\beta^\top \mu = \frac{1}{n} (X^\top y)^\top \mu$ in data space. It reflects a linear decision boundary **in the data space**, not one learned through neural network training. We will clarify this in the revised manuscript to avoid potential misinterpretations.
>
> However, your question raises a valid point about **how stable this assignment** is to perturbations in the training data itself, such as *reduced class separability*. In such cases, **the magnitude of $\beta$ decreases, reducing Cohesion and Separability scores in Eqs. 6 and 7 due to the $||\beta||$ term**. This aligns with the intuitive notion that *noisier training data leads to less transferable features*.
>
> To illustrate, suppose the training data consists of two classes drawn from $c_1 \sim \mathcal{N}(\mu, I)$ and $c_2 \sim \mathcal{N}(R\mu, I)$, where $R$ is a rotation matrix. Then the spike direction converges (under large $n$) to $\beta \rightarrow \frac{1}{2} (\mu - R\mu)$ by law of large numbers.
>
> $\beta = \frac{1}{n} X^\top y = \frac{1}{n} X_i y_i = \frac{1}{2} (\frac{n}{2} \sum_{c_1} X_i - \frac{n}{2} \sum_{c_2} X_j) \rightarrow \frac{1}{2} (\mu - R \mu) .$
>
> In the extreme case where $R = I$, the classes are indistinguishable and $\beta = 0$, eliminating both Cohesion and Separability. When $R = -I$, $\beta = \mu$, which yields maximal separation.
>
> ## Response to Minor Weaknesses
>
> 1. Specific key theoretical steps, especially the transition from Theorem 3.3 to Definition 3.4 (Dominant Feature), would benefit from a more intuitive explanation in the main text. We will *include a summary and interpretation to aid the reader's understanding*.
> 2. We acknowledge limitations of this approach in low-dimensional regimes. In the camera-ready version, we will discuss the need for alternative, more interpretable and efficient approximations better suited for these settings. This will clarify the scope of our approximation.
>
> ## Multi step and Multi-layer experiments
>
> To test the validity of our theory in multi-layer and multi-step settings, we applied Expr 1 and observed that **increasing $\beta^\top\mu$ generally improves both Cohesion and Separability**, consistent with the prediction in Section 4.1.
>
> In deeper networks, however, excessively large $\beta^\top\mu=5 \cdot 10^3$ leads to reduced Separability in 3- and 4-layer cases, as unseen features become aligned yet too far in magnitude. This suggests an interesting direction for future work: identifying a potential critical threshold of alignment strength in deep models that balances directional consistency and proximity.
>
> As we are unable to include figures, we provide tables instead. We report Cohesion and Separability in separate tables, where each column corresponds to a layer or step setting, and each row shows values for increasing $\beta^\top\mu$.
>
> ### Multi-layer One-step, lr 0.1, and Expr 1 of Training Data 2 (Truncated Gaussian)
>
> Cohesion
>
> | $\beta^\top\mu$ |  | 2 layer | 3 layer | 4 layer | 5 layer |
> | --- | --- | --- | --- | --- | --- |
> | 0 | $F_0$ | 0.61 | 58.02 | 144.89 | 182.08 |
> |  | $F$ | **0.61** | 53.67 | 144.89 | **185.57** |
> | $10^2$ | $F_0$ | 1.13 | 54.77 | 137.48 | 183.82 |
> |  | $F$ | **32.86** | 66.33 | 137.48 | **187.34** |
> | $10^3$ | $F_0$ | 49.02 | 65.66 | 136.58 | 181.23 |
> |  | $F$ | **5229.26** | 4260.94 | 360.86 | **200.20** |
> | $5 \cdot 10^3$ | $F_0$ | 1279.74 | 428.04 | 163.06 | 171.20 |
> |  | $F$ | **119560.01** | 107193.12 | 13343.40 | **1868.45** |
>
> Separability
>
> | $\beta^\top\mu$ |  | 2 layer | 3 layer | 4 layer | 5 layer |
> | --- | --- | --- | --- | --- | --- |
> | 0 | $F_0$ | -0.04 | -59.73 | -144.00 | -198.57 |
> |  | $F$ | **-0.04** | -59.73 | -144.00 | **-200.27** |
> | $10^2$ | $F_0$ | 0.62 | -64.89 | -149.52 | -208.04 |
> |  | $F$ | **30.99** | -41.70 | -148.67 | **-200.3** |
> | $10^3$ | $F_0$ | 56.95 | -56.56 | -141.32 | -189.67 |
> |  | $F$ | **1027.47** | 390.64 | 175.64 | **-105.12** |
> | $5 \cdot 10^3$ | $F_0$ | 582.36 | 132.975 | -23.26 | -142.38 |
> |  | $F$ | **5955.02** | ***-10772.3*** | ***148.312*** | **305.38** |
>
> ### Multi-step Two-layer, lr 0.001, and Expr 1 of Training Data 2 (Truncated Gaussian)
>
> Cohesion
>
> | $\beta^\top\mu$ |  | 1 step | 2 step | 3 step | 4 step | 5 step |
> | --- | --- | --- | --- | --- | --- | --- |
> | 0 | $F_0$ | 0.66 | 0.67 | 0.66 | 0.65 | 0.59 |
> |  | $F$ | 0.66 | 0.67 | 0.66 | 0.65 | 0.59 |
> | $10^2$ | $F_0$ | 1.12 | 1.20 | 1.16 | 1.20 | 1.14 |
> |  | $F$ | 1.12 | 1.20 | **1.26** | **2.41** | **44.59** |
> | $10^3$ | $F_0$ | 48.48 | 51.70 | 48.80 | 50.22 | 46.25 |
> |  | $F$ | 48.43 | 51.70 | **59.28** | **217.51** | **4974.99** |
> | $5 \cdot 10^3$ | $F_0$ | 1324.11 | 1311.85 | 1275.23 | 1253.07 | 1249.63 |
> |  | $F$ | 1324.11 | 1311.85 | **1617.79** | **6592.32** | **119587.06** |
>
> Separability
>
> | $\beta^\top\mu$ |  | 1 step | 2 step | 3 step | 4 step | 5 step |
> | --- | --- | --- | --- | --- | --- | --- |
> | 0 | $F_0$ | -0.05 | -0.05 | -0.05 | -0.05 | -0.05 |
> |  | $F$ | -0.05 | -0.05 | -0.05 | -0.05 | -0.05 |
> | $10^2$ | $F_0$ | 0.45 | 0.50 | 0.47 | 0.44 | 0.43 |
> |  | $F$ | 0.45 | 0.50 | **0.57** | **1.71** | **51.32** |
> | $10^3$ | $F_0$ | 42.67 | 43.02 | 45.39 | 46.56 | 43.34 |
> |  | $F$ | 42.67 | 43.02 | **51.59** | **132.29** | **1183.37** |
> | $5 \cdot 10^3$ | $F_0$ | 602.20 | 608.54 | 581.89 | 599.86 | 590.46 |
> |  | $F$ | 602.20 | 608.54 | **703.97** | **1277.33** | **6420.12** |

---

> > ### Author Response · Authors · 2025-08-08
> > **Polite Follow-up on the Review**
> >
> > Dear Reviewer MzGZ,
> >
> > We sincerely appreciate the time and care you have devoted to reviewing our paper, as well as the positive evaluation you provided.
> > Your thoughtful feedback has been immensely helpful in improving the clarity and depth of our work.
> >
> > In particular, we are grateful for your insightful suggestions regarding the multi-step and multi-layer extensions, the importance of validating the train-unseen similarity, and the need to clarify the theoretical notion of assignment. These points have guided us in strengthening both the theoretical and empirical aspects of our paper.
> >
> >
> > Following your comments, we conducted additional experiments and provided detailed responses in our rebuttal.
> > Specifically:
> > * We discussed and extended our experiments to **multi-step and multi-layer** settings, observing that the trends in **Cohesion and Separability largely align with our theoretical predictions.**
> > * We quantitatively **examined the train-unseen similarity** using cosine similarity between class-wise embeddings **across domains**, supporting the assumptions made in our experiments.
> > * We also **clarified the theoretical nature of the assignment** and analyzed its stability under variations in training data.
> >
> > We would truly appreciate it if you could let us know whether our responses have sufficiently addressed your concerns, or if there is anything further we should clarify.
> > As the discussion period is nearing its end, we’re reaching out through this official comment to kindly bring your attention back to our response—fully understanding that your time is limited and valuable.
> >
> > Your expertise and perspective are deeply appreciated.  Thank you again for your contribution to the review process.
> > Warm regards, The Authors

---

> > > ### Comment · Reviewer_MzGZ · 2025-08-09
> > >
> > > Thank you for your reply and follow-up. Overall, my concerns have been largely resolved. I will take this into consideration for my final evaluation.

---

### Note · Authors · 2025-08-12

Dear AC and Reviewers,

First, we sincerely thank all reviewers and the AC for the constructive feedback and the time they have devoted to our work. During the discussion, we resolved most concerns, as acknowledged by the reviewers (MzGZ, yNMA, fCzN). Below, we summarize the results.

### Review outcome

Our submission initially received two Weak Accept and two Weak Reject. After the rebuttal and discussions:

- yNMA increased their score, noting that their concerns were mostly addressed.
- Both Weak Accept reviewers indicated that concerns were largely resolved (MzGZ) or our responses were acknowledged (fCzN).
- yqtE was unable to join the discussion, but we made every effort to address their concerns and will reflect the changes.

### Summary of Responses to Major Concerns

1. **Concerns Regarding the Link Between Theory and Practice**
    1. Applicability to Multi-step and Multi-layer Settings: We discussed our intuition and conducted additional experiments in multi-step training and multi-layer networks, finding that the theoretical trends in *Cohesion* and *Separability* generally hold.
    2. Relationships between (Semantic Similarity, Train–Unseen Similarity) and (Recall@1, Separability): We performed an additional quantitative analysis (Experiment 4) and confirmed associations between each pair of measures.
2. **Concerns about Theoretical Assumptions**
    1. We clearly stated that our study primarily employs assumptions widely used in prior research.
    2. We remarked that our simplified setup is sufficient to address our research question of feature transfer.
3. **Concerns about Novelty**: We clarified the following novel contributions:
    1. Novel theoretical tools and a framework for classifier analysis.
    2. Provide a counterintuitive finding – Feature expressivity increases mainly with more semantically related classes, not just more classes.
    3. Unifying empirical observations and prior theories through the theoretical quantification of feature transfer.

### Commitments for the camera-ready version

We incorporated new experiments and analyses, and clarified definitions and notation. To enhance readability and support future work, we revised the introduction and conclusion. Finally, we offered clearer intuitions for key theoretical steps.

We are grateful for the constructive exchange and the opportunity to refine our work. We hope that these clarifications demonstrate how we have addressed the main concerns.

Thank you.

---

### Decision · Program_Chairs · 2025-09-17

**Decision:**

Accept (poster)

**Comment:**

This paper studies transfer learning, in particular, a theoretical analysis of how features learned by a two-layer classifier generalize to unseen classes. The analysis is conducted in a simplified setting: one gradient descent step and an asymptotic regime where the sample size, data dimension, and feature dimension all grow to infinity. Within this framework, the authors decompose features into components and study how similarity between the training and unseen data affects clustering quality. Experiments on both synthetic and real-world data support the theoretical findings.

A common concern is the restricted setting of a two-layer model, a single gradient descent step, and the asymptotic case where all dimensions and sample size go to infinity. Nevertheless, the reviewers acknowledge the solid theoretical analysis and contributions, and overall recommend acceptance.